# Sources and fate of nitrate in groundwater at agricultural operations overlying glacial sediments

Sarah A. Bourke[1,2], Mike Iwanyshyn[3], Jacqueline Kohn[4], M. Jim Hendry[1]

[1]Department of Geological Sciences, University of Saskatchewan, SK, S7N 5C9, Canada
[2]School of Earth Sciences, University of Western Australia, Crawley, WA, 6009, Australia
[3]Natural Resources Conservation Board, Calgary, AB, T2P 0R4, Canada
[4]Alberta Agriculture and Forestry, Irrigation and Farm Water Branch, Edmonton, AB, T6H 5T6, Canada

*Correspondence to*: Sarah A. Bourke (sarah.bourke@uwa.edu.au)

**Abstract.** Leaching of nitrate ($NO_3^-$) from animal waste or fertilizers at agricultural operations can result in $NO_3^-$ contamination of groundwater, lakes, and streams. Understanding the sources and fate of nitrate in groundwater systems in glacial sediments, which underlie many agricultural operations, is critical for managing impacts of human food production on the environment. Elevated $NO_3^-$ concentrations in groundwater can be naturally attenuated through mixing or denitrification. Here we use isotopic enrichment of the stable isotope values of $NO_3^-$ to quantify the amount of denitrification in groundwater at two confined feeding operations overlying glacial sediments in Alberta, Canada. Uncertainty in $\delta^{15}N_{NO3}$ and $\delta^{18}O_{NO3}$ values of the $NO_3^-$ source and denitrification enrichment factors are accounted for using a Monte Carlo approach. When denitrification could be quantified, we used these values to constrain a mixing model based on $NO_3^-$ and $Cl^-$ concentrations. Using this novel approach we were able to reconstruct the initial $NO_3$-N concentration and $NO_3$-N/$Cl^-$ ratio at the point of entry to the groundwater system. Manure filtrate had total-nitrogen (TN) of up to 1820 mg $L^{-1}$, which was predominantly organic-N and $NH_3$. Groundwater had up to 85 mg $L^{-1}$ TN, which was predominantly $NO_3^-$. The addition of $NO_3^-$ to the local groundwater system from temporary manure piles and pens equalled or exceeded $NO_3^-$ additions from earthen manure storages at these sites. On-farm management of manure waste should therefore increasingly focus on limiting manure piles in direct contact with the soil, and encourage storage in lined lagoons. Nitrate attenuation at both sites is attributed to a spatially variable combination of mixing and denitrification, but is dominated by denitrification. Where identified, denitrification reduced agriculturally-derived $NO_3^-$ concentrations by at least half and, in some wells, completely. Infiltration to groundwater systems in glacial sediments where $NO_3^-$ can be naturally attenuated is likely preferable to off-farm export via runoff or drainage networks, especially if local groundwater is not used for potable water supply.

## 1 Introduction

The contamination of soil and groundwater with nitrate from agricultural operations is a global water quality issue that has been extensively documented (Power and Schepers, 1989; Spalding and Exner, 1993; Rodvang and Simpkins, 2001; Galloway et al., 2008; Zirkle et al., 2016; Arauzo, 2017; Ascott et al., 2017). Leaching of nitrate ($NO_3^-$) from animal waste or fertilizers can result in groundwater $NO_3^-$ concentrations that exceed drinking water guidelines and pose human health risks (Fan and Steinberg, 1996; Gulis et al., 2002; Yang et al., 2007). The discharge of high-$NO_3^-$ groundwater, runoff, or drainage can contaminate streams and lakes, resulting in

eutrophication and ecosystem decline (Deutsch et al., 2006; Kaushal et al., 2011). In saturated groundwater systems with low oxygen concentrations, elevated $NO_3^-$ can be naturally attenuated by microbial denitrification (Wassenaar, 1995; Robertson et al., 1996; Smith et al., 1996; Tesoriero et al., 2000; Singleton et al., 2007). Concentrations of $NO_3^-$ will also decrease along groundwater flow paths due to attenuation via dilution by

hydrodynamic dispersion (referred to hereafter as mixing). Because of these natural attenuation mechanisms, infiltration to groundwater may be preferable to off-site drainage and runoff of nitrate-rich waters. Many agricultural operations are undertaken on fertile soils associated with glacial sediments (Spalding and Exner, 1993; Ernstsen et al., 2015; Zirkle et al., 2016). Understanding the sources and fate of agriculturally derived nitrate in groundwater systems in glacial sediments is therefore critical for managing impacts of human food production on

the environment.

Identification of the sources and fate of $NO_3^-$ at agricultural operations can be challenging because of spatial and temporal variations in sources (e.g. earthen manure storage, temporary manure piles, or fertilizer) and heterogeneity in hydrogeologic systems (Spalding and Exner, 1993; Rodvang et al., 2004; Showers et al., 2008; Kohn et al., 2016). These spatial and temporal variations can result in complex subsurface solute distributions that

are difficult to interpret using classical transect studies or numerical groundwater models (Green et al., 2010; Baily et al., 2011).

Groundwater containing significant agriculturally derived $NO_3^-$ also typically has elevated chloride ($Cl^-$) concentrations (Saffigna and Keeney, 1977; Rodvang et al., 2004; Mencló et al., 2016). Decreasing $NO_3$-N/$Cl^-$ (or $NO_3^-$/$Cl^-$) ratios have been used to define denitrification based on the assumption that $NO_3^-$ is reactive while

$Cl^-$ is non-reactive (conservative), such that denitrification results in a decrease in the $NO_3$-N/$Cl^-$ ratio (Kimble et al., 1972; Weil et al., 1990; Liu et al., 2006; McCallum et al., 2008). However, $NO_3$-N/$Cl^-$ ratios can also change in response to mixing of groundwater with different $NO_3$-N/$Cl^-$ ratios or when groundwater sampling traverses hydraulically disconnected formations (Bourke et al., 2015b). If $NO_3$-N/$Cl^-$ ratios vary among potential sources and the $NO_3$-N/$Cl^-$ ratio at the point of entry to the groundwater system can be reconstructed, this information

could be used to show that anthropogenic $NO_3^-$ at different locations within an aquifer is derived from the same or different sources.

The stable isotopes of $NO_3^-$ ($\delta^{15}N_{NO3}$ and $\delta^{18}O_{NO3}$) provide an alternative approach to characterize the source and fate of $NO_3^-$ in groundwater systems. In agricultural areas, multiple sources of $NO_3^-$ are common and could include precipitation, soil $NO_3^-$, inorganic fertilizer, manure, and septic waste (Komor and Anderson, 1993; Liu et al.,

2006; Pastén-Zapata et al., 2014; Clague et al., 2015; Xu et al., 2015). While source identification is theoretically possible using $\delta^{15}N_{NO3}$ and $\delta^{18}O_{NO3}$ (particularly with a dual-isotope approach), in practice this can be difficult due to geologic heterogeneity, overlapping source values, and the complexity of biologically mediated reactions (Aravena et al., 1993; Wassenaar, 1995; Mengis et al., 2001; Choi et al., 2003; Granger et al., 2008; Vavilin and Rytov, 2015; Xu et al., 2015).

$NO_3^-$ attenuation by denitrification in groundwater systems can be identified based on the characteristic enrichment of $\delta^{15}N_{NO3}$ and $\delta^{18}O_{NO3}$. Numerous studies have made qualitative assessments that identified denitrification in groundwater using the stable isotope approach (Böttcher et al., 1990; Wassenaar, 1995; Singleton et al., 2007; Baily et al., 2011; Clague et al., 2015; Xu et al., 2015). Recently published papers have also used stable isotopic values of $NO_3^-$ and water as the basis for mixing models in agricultural settings (Ji et al., 2017

;Lentz and Lehersch, 2019). Isotopic fractionation effects can also allow for quantitative assessment of the

proportion of substrate that has undergone a given reaction, if enrichment factors and source values are known; as in the case of evpoarative loss of water, for example (Dogramaci et al., 2012). To date, there have been very few attempts to quantify denitrification using dual-isotope enrichment, largely due to uncertainty in source values and enrichment factors (Böttcher et al., 1990, Xue et al., 2009).

The only published calculations of the fraction of $NO_3^-$ remaining after denitrification the that we are aware of assumed a constant enrichment factor and the same isotopic source values across the field site (Otero et al., 2009). However, the enrichment factor will vary across a field site in response to reaction rates (Kendall and Aravena 2000), and isotopic values of even the same type of source (e.g. manure) can vary substantially (Xue et al., 2009). If the varation in source values and enrichment factors can be characterized from measured data then these

uncertainties can be accounted for using a Monte Carlo approach (Joerin et al., 2002; Bourke et al., 2015a; Ji et al., 2017), thereby extending the application of the dual-isotope technique to allow for a robust quantitative assessment of denitrification in agricultural settings.

A synthesized analysis of stable isotopes of $NO_3^-$ with additional ionic tracers can further improve the assessment of $NO_3^-$ attenuation mechanisms and sources of $NO_3^-$ in agricultural settings (Showers et al., 2008; Vitòria et al.,

2008; Xue et al., 2009; Xu et al., 2015; Ji et al., 2017). We hypothesise that if the amount of denitrification can be quantified based on $\delta^{15}N_{NO3}$ and $\delta^{18}O_{NO3}$, then this estimate of the fraction of $NO_3$-N removed through denitrification can be used to constrain a mixing model based on $NO_3$-N and $Cl^-$ concentrations. This novel approach allows for the ratio of $NO_3$-N/$Cl^-$ at the point of entry to the groundwater system to be reconstructed from measured $NO_3^-$ and $Cl^-$ concentrations (see Section 2.3). Where the $NO_3$-N/$Cl^-$ ratio varies between sources,

this ratio can then be used to assess the source of the $NO_3^-$ in groundwater (e.g. temporary manure piles or feeding pens). These data can also then be used to estimate the initial concentrations of $NO_3^-$ and $Cl^-$ at the point of entry to the groundwater system and quantify attenuation by mixing.

In this study, we present the application of this approach at two confined feeding operations (CFOs) in Alberta, Canada, with differing lithologies and durations of operation (Fig. 1). Concentrations of $Cl^-$ and nitrogen species

(N-species) and the stable isotopes of $NO_3^-$ were measured in groundwater samples collected from monitoring wells and continuous soil cores, as well as manure filtrate at both sites. These data were interpreted to (1) assess the extent of agriculturally derived $NO_3^-$ in groundwater, (2) identify sources and initial concentrations of $NO_3^-$ at the point of entry to the groundwater system, and (3) assess mixing and denitrification as attenuation mechanisms at these sites.

**2 Materials and methods**

**2.1 Experimental sites**

This study was conducted using data from two of the five sites investigated by Alberta Agriculture and Forestry during an assessment of the impacts of livestock manure on groundwater quality (Lorenz et al., 2014). To the best of our knowledge (including discussions with farm operators) fertilizers have not been applied at either of these

sites. As such, manure waste from livestock is assumed to be the sole source of agricultural nitrogen (N) and elevated $NO_3^-$ concentrations in groundwater at these sites.

The first study site (CFO1) is located 25 km northeast of Lethbridge, Alberta (Fig. 1). Agricultural operations at this site were initiated with the construction of a dairy in 1928, which has the capacity for 150 dairy cattle. A

feedlot for beef cattle was added in 1960s along with an earthen manure storage (EMS) facility for storing liquid dairy manure (approx. 4 m deep) and a catch-basin that receives surface water runoff. This feedlot was expanded in the 1980s to the 2000 head capacity it was at the time of this study. There is also a dugout (or slough, a shallow wetland) on site that receives local runoff and an irrigation drainage canal at the southern boundary of the property.

The second study site (CFO4) is located approximately 30 km north of Red Deer, Alberta and 300 km north of CFO1. This dairy and associated EMS (approx. 6 m deep) were constructed in 1995 and the facility had 350 head of dairy cattle at the time of the study. Runoff will drain either to the small dugout in the north-west of the site, or the natural drainage features (ephemeral ponds or a creek approx. 1.5 km east).

### 2.2 Sampling and instrumentation

#### 2.2.1 Groundwater monitoring wells

Groundwater samples were collected from water table wells and piezometers (hereafter both are referred to as wells) installed at both sites (Table 1). At CFO1, groundwater samples were collected from six individual water table wells (DMW1, DMW2, DMW3, DMW4, DMW5, DMW6) and eight sets of nested wells with one well screened at the water table and one well screened 20 m below ground (BG) (DP10-2 and DP10-1, DMW10 and DP11-10b, DMW11 and DP11-11b, DMW12 and DP11-12b, DMW13 and DP11-13b, DMW14 and DP11-14b, DMW15 and DP11-15b, and DMW16 and DP11-16b). Wells DP10-2 and DP10-1 were located directly adjacent to the EMS on the hydraulically downgradient side. At CFO4, groundwater samples were collected from eight water table wells (BC1, BC2, BC3, BC4, BC5, BMW1, BMW3, BMW7) and four sets of nested wells, with wells screened across the water table and at 15 m BG. Two of these nests were located adjacent to the EMS (BMW2 and BP10-15e, BMW4 and BP10-15w) and two were hydraulically downgradient of the EMS (BMW5 and BP5-15, BMW6 and BP6-15).

Groundwater samples were collected for ion analysis (Cl$^-$ and N-species) quarterly between April 2010 and August 2015. All water samples were collected using a bailer after purging (1–3 casing volumes) and stored at $\leq$ 4 °C prior to analysis. Samples for $\delta^{15}N_{NO3}$ and $\delta^{18}O_{NO3}$ were collected from wells at CFO1 on 1 January 2013 and 1 May 2013. Samples for $\delta^{15}N_{NO3}$ and $\delta^{18}O_{NO3}$ at CFO4 were collected on 27 October 2014. Wells were purged prior to sample collection (1–3 casing volumes), and samples filtered into high-density polyethylene (HDPE) bottles in the field and frozen until analysis.

Hydraulic heads in monitoring wells were determined using manual measurements (approximately monthly, 2010-2015). Hydraulic head response tests were conducted on the majority of the wells at the sites to determine hydraulic conductivity ($K$) of the formation media surrounding the intake zone. These tests were either a slug test (water level decline after water addition), or bail test (water level recovery after water removal) depending on the location of the water table within the well at the time of testing. K was determined from hydraulic the head responses using the method of Hvorslev (1951).

#### 2.2.2 Continuous core

Continuous core was collected at CFO1 immediately adjacent to well DP11-13b on 1 May 2013 (Fig. 1). Additional core samples were collected from 1 to 5 June 2015 along a transect hydraulically downgradient of the southeastern side of the EMS at CFO1 where hydrochemistry data suggested leakage from the EMS (see Section 3). During this 2015 drilling campaign, core samples were collected at four locations (DC15-20, DC15-21,

DC15-22, DC15-23) to depths of up to 15 m below surface and distances of up to 100 m from the EMS between wells DMW3 and DP11-14.

Continuous core samples were retrieved using a hollow stem auger (1.5-m core lengths) with 0.3-m sub-samples collected at approximately 1-m intervals ensuring that visually consistent lithology could be sampled. Core samples for Cl$^-$ were stored in Ziploc$^{TM}$ bags and kept cool until analysis. Core samples for N-species analysis were stored in Ziploc bags filled with an atmosphere of argon (99.9% Ar) to minimize oxidation and kept cool until analysis. Subsamples of each core (250-300 g) were placed under 50 MPa pressure in a Carver Series NE mechanical press with a 0.5-μm filter placed at the base of the squeezing chamber, which was placed within an Ar atmosphere to minimize oxidation. A syringe was attached to the base of the apparatus and 15 mL of filtered pore water were collected for analyses within 3.5 to 6.0 h (Hendry et al., 2013).

### 2.2.3 Liquid manure storages

Samples of liquid manure slurry were collected directly from the EMS at both sites and the catch basin (containing local runoff from the feedlot) at CFO1 using a pipe and plunger apparatus to sample from approximately 0.5 m below the surface. The slurry collected was subsequently filtered (0.45 μm) to separate the liquid and solid components. The water filtered from samples collected from the EMS or catch basin is hereafter referred to as manure filtrate.

### 2.3 Laboratory analysis

Groundwater samples from wells were analysed by Alberta Agriculture and Forestry (Lethbridge, Alberta). Concentrations of Cl$^-$ were determined using potentiometric titration of H$_2$O, with a detection limit of 5.0 mg L$^{-1}$ and accuracy of 5% (APHA 4500-Cl$^-$ D). Concentrations of NH$_3$ as N (NH$_3$-N), NO$_3^-$ as N (NO$_3$-N), and NO$_2^-$ as N (NO$_2$-N) were measured by air-segmented continuous flow analysis (APHA 4500-NH3 G, APHA 4500-NO3 F). Total nitrogen (TN) was determined by high temperature catalytic combustion and chemiluminescence detection using a Shimadzu TOC-V with attached TN unit (ASTM D8083-16). Total organic nitrogen (TON) was calculated by subtracting NH$_3$-N, NO$_3$-N and NO$_2$-N from TN. Bicarbonate (HCO$_3^-$) was analysed by titration (APHA 2320 B). Dissolved organic carbon (DOC) was analysed by a combustion infrared method (APHA 5310 B) using a Shimadzu TOC-V system. Manure filtrate was analysed by ALS (Saskatoon, Saskatchewan) using similar methods for Cl$^-$ (APHA 4110 B), TN (RMMA A3769 3.3), NO$_3$+NO$_2$ as N (APHA 4500-NO3-F), NH$_3$-N (APHA 4500-NH3 D), HCO$_3^-$ (APHA 2320) and DOC (APHA 5310 B).

Pore-water samples squeezed from continuous core were analysed at the University of Saskatchewan (Saskatoon, Canada) for Cl$^-$, NO$_3$-N, and NO$_2$-N using a Dionex IC25 ion chromatograph (IC) coupled to a Dionex As50 autosampler (EPA Method 300.1, accuracy and precision of 5.0%) (Hautman and Munch, 1997). Ammonia as N (NH$_3$-N) was measured by Exova Laboratories using the automated phenate method (APHA Standard 4500-NH3 G, detection limit of 0.025 mg L$^{-1}$, accuracy of 2% of the measured concentration, and a precision of 5% of the measured concentration).

$\delta^{15}$N$_{NO3}$ and $\delta^{18}$O$_{NO3}$ in groundwater samples (from wells and pore water from continuous core) and manure filtrate were measured at the University of Calgary (Calgary, Alberta) using the denitrifier method (Sigman et al., 2001) with an accuracy and precision of 0.3‰ for $\delta^{15}$N$_{NO3}$ and 0.3‰ for $\delta^{18}$O$_{NO3}$. Groundwater samples collected for

NO$_3^-$ isotope analysis in January 2013 were also analyzed for NO$_3$-N by the University of Calgary (denitrifier technique, Delta+XL).

## 2.4 Modelling approach

### 2.4.1 Quantification of denitrification based on δ$^{15}$N$_{NO3}$ and δ$^{18}$O$_{NO3}$

Nitrate in groundwater that has undergone denitrification is commonly reported as being identified by enrichment of δ$^{15}$N$_{NO3}$ and δ$^{18}$O$_{NO3}$ with a slope of about 0.5 on a cross-plot (Clark and Fritz, 1997). However, published studies of denitrification in groundwater report slopes of up to 0.77 (Mengis et al., 1999; Fukada et al., 2003; Singleton et al., 2007). The relationship between isotopic enrichment of $^{15}$N$_{NO3}$ and $^{18}$O$_{NO3}$ and the fraction of NO$_3$-N remaining during denitrification can be described by a Rayleigh equation:

$$R = R_0 f_d^{(\frac{1}{\beta}-1)},$$    (1)

where $R_0$ is the initial isotope ratio (relative to the standard) of the NO$_3^-$ (δ$^{18}$O$_{NO3}$ or δ$^{15}$N$_{NO3}$), $R$ is the isotopic ratio when fraction $f_d$ of NO$_3^-$ remains, and $\beta$ is the kinetic fractionation factor ($> 1$) (Böttcher et al., 1990; Clark and Fritz, 1997; Otero et al., 2009; Xue et al., 2009). Kinetic fraction effects are commonly also expressed as the enrichment factor, $\varepsilon = \frac{1}{1000(\beta-1)}$ . In the case of a constant enrichment factor, $f_d$ can be calculated from measured

δ$^{15}$N$_{NO3}$ (or δ$^{18}$O$_{NO3}$), if the initial δ$^{15}$N$_{NO3}$ (δ$^{15}$N$_0$) is known;

$$f_d = exp\left(\frac{\delta^{15}N_{NO3}-\delta^{15}N_0}{\varepsilon}\right)$$    (2)

The fraction of NO$_3$-N removed from groundwater through denitrification is then given by (1-$f_d$). The concentration of NO$_3$-N that would have been measured if mixing was the only attenuation mechanism (*NO$_3$-N$_{mix}$*) can also be calculated by dividing the measured concentration by $f_d$.

A sub-set of 20 samples with isotopic values of NO$_3^-$ indicative of denitrification were identified, and for each of these samples $f_d$ (mean and standard deviation) was calculated from Eq. (2) using a Monte Carlo approach with 500 realizations.). The distribution of $\varepsilon$ values was defined based on measured data. If the initial δ$^{15}$N$_{NO3}$ is known, $\varepsilon$ for δ$^{15}$N$_{NO3}$ ($\varepsilon_{15N}$) can be determined from the slope of the linear regression line on a plot of $ln(f_d)$ vs. δ$^{15}$N$_{NO3}$ (Böttcher et al., 1990). If the initial δ$^{15}$N$_{NO3}$ and $f_d$ are not known, as is the case here, $\varepsilon_{15N}$ can be determined from

the slope of the regression line on a plot of $ln$(NO$_3$-N) vs. δ$^{15}$N$_{NO3}$, which will be the same as on a plot of $ln(f_d)$ vs. δ$^{15}$N$_{NO3}$. In-situ variations in temperature and reaction rates may affect the enrichment factor (Kendall and Aravena, 2000) and this was accounted for by allowing for variation in $\varepsilon_{15N}$ within the Monte Carlo analysis. The enrichment factor for δ$^{18}$O$_{NO3}$ ($\varepsilon_{18O}$) was calculated by multiplying the δ$^{15}$N$_{NO3}$ by a linear coefficient of proportionality determined for each CFO from the slope of the denitrification trend on an isotope cross-plot (see

Section 3.2).

For each realization, initial isotopic values (δ$^{15}$N$_0$ and δ$^{18}$O$_0$) were determined by Solver such that the difference between $f_d$ calculated from δ$^{15}$N$_{NO3}$ and δ$^{18}$O$_{NO3}$ was minimized (<1% difference). The ranges of δ$^{15}$N$_0$ and δ$^{18}$O$_0$ were limited based on measured data and literature values (see 3.2). This approach neglects the effect of mixing of groundwater with differing isotopic values, and is valid if the concentration of NO$_3^-$ in the source is much

greater than background concentrations such that the isotopic composition of NO$_3^-$ is dominated by the agriculturally derived end-member.

### 2.4.2 Quantification of mixing and initial concentrations of Cl⁻ and NO₃-N

A binary mixing model that also accounts for decreasing NO₃-N concentrations in response to denitrification was used to quantify NO₃⁻ attenuation by mixing and estimate the initial concentrations of Cl⁻ and NO₃-N. The measured concentration of Cl⁻ was assumed to be a function of two end-member mixing, described by

$$Cl = f_{\mathrm{m}} Cl_{\mathrm{i}} + (1 - f_{\mathrm{m}}) Cl_{\mathrm{b}}, \tag{3}$$

where $Cl$ is the measured concentration of Cl⁻ in the groundwater sample, $Cl_{\mathrm{i}}$ is the concentration of Cl⁻ at the initial point of entry of the agriculturally derived NO₃⁻ to the groundwater system, $Cl_{\mathrm{b}}$ is the concentration of Cl⁻ in the background ambient groundwater, and $f_{\mathrm{m}}$ is the fraction of water in the sample from the source of agriculturally derived Cl⁻ (and NO₃⁻) remaining in the mixture.

The concentration of NO₃-N was also assumed to be a function of two end-member mixing but with an additional coefficient, $f_{\mathrm{d}}$ (the fraction of NO₃-N remaining after denitrification), applied to account for denitrification. The measured NO₃-N concentration was thus described by

$$NO_3\text{-}N = f_{\mathrm{d}}(f_{\mathrm{m}} NO_3\text{-}N_{\mathrm{i}} + (1 - f_{\mathrm{m}}) NO_3\text{-}N_{\mathrm{b}}), \tag{4}$$

where $NO_3\text{-}N$ is the concentration of NO₃-N measured in the groundwater sample, $NO_3\text{-}N_{\mathrm{i}}$ is the concentration of NO₃-N in the source of agriculturally derived NO₃⁻ at the initial point of entry to the groundwater system, and $NO_3\text{-}N_{\mathrm{b}}$ is the concentration of NO₃-N in the background ambient groundwater. This mixing calculation was only conducted on samples for which NO₃⁻ dominated total-N (NH₃-N <10% of NO₃-N) so that nitrification of NH₃ could be neglected.

If $Cl_{\mathrm{i}}$ is much greater than $Cl_{\mathrm{b}}$ and $NO_3\text{-}N_{\mathrm{i}}$ is much greater than $NO_3\text{-}N_{\mathrm{b}}$, then $f_{\mathrm{m}}$ is insensitive to background concentrations and these terms can be neglected (see 4.2 for further discussion of this assumption). In this case, Eqs. (3) and (4) reduce to

$$Cl = f_{\mathrm{m}} Cl_{\mathrm{i}}, \tag{5}$$

$$NO_3\text{-}N = f_{\mathrm{d}}(f_{\mathrm{m}} NO_3\text{-}N_{\mathrm{i}}). \tag{6}$$

Solving Eq. (6) for $f_{\mathrm{m}}$ and substituting into Eq. (5) yields

$$\frac{NO_3\text{-}N_{\mathrm{i}}}{Cl_{\mathrm{i}}} = \frac{1}{f_{\mathrm{d}}} \frac{NO_3\text{-}N}{Cl}. \tag{7}$$

Thus, for each groundwater sample, the ratio of NO₃-N/Cl⁻ at the initial point of entry of the agriculturally derived NO₃⁻ to the groundwater system $\left(\frac{NO_3\text{-}N_{\mathrm{i}}}{Cl_{\mathrm{i}}}\right)$ can be simply calculated using measured concentrations, and $f_{\mathrm{d}}$ estimated from NO₃⁻ isotope data. This provides a relatively simple method to identify agriculturally derived NO₃⁻ from different sources (e.g., EMS vs. manure piles) if they have different NO₃-N/Cl⁻ ratios. Estimated $Cl_{\mathrm{i}}$ and $NO_3\text{-}N_{\mathrm{i}}$ are reported as the mid-range value with uncertainty described by the minimum and maximum values. These initial concentrations are at the water table for top-down inputs, or at the saturated point of contact between the EMS and the aquifer for leakage from the EMS. This analysis assumes that a sampled water parcel consists of water with agriculturally derived NO₃⁻ that entered the aquifer from one source at one point in time and space and has since mixed with natural ambient groundwater. Any NO₃⁻ produced during nitrification after the anthropogenic source water enters the aquifer is implicitly included in $NO_3\text{-}N_{\mathrm{i}}$. The error in $\frac{NO_3\text{-}N_{\mathrm{i}}}{Cl_{\mathrm{i}}^-}$ was assumed to be dominated by error in the estimated $f_{\mathrm{d}}$, with the measurement error in NO₃-N and Cl⁻ considered negligible.

The initial concentrations of the agriculturally derived NO₃⁻ source ($NO_3\text{-}N_{\mathrm{i}}$ and $Cl_{\mathrm{i}}$) were estimated by simultaneously solving Eqs. (5) and (6) using Excel Solver (GRG nonlinear). The absolute minimum values of

$NO_3\text{-}N_i$ and $Cl_i$ were defined by measured concentrations (e.g., if $Cl_i = Cl$, $f_m = 1$). Maximum values of $NO_3\text{-}N_i$ and $Cl_i$ were defined based on measured concentrations of $NO_3$-N and $Cl^-$ in groundwater and manure filtrate ($NO_3$-N $\leq 150$ mg L$^{-1}$ and Cl$^-$ $\leq 1300$ mg L$^{-1}$; see Section 3.2). These maximum values of $NO_3\text{-}N_i$ and $Cl_i$ correspond to the minimum $f_m$. The value of $f_d$ was assumed to be the mean $f_d$ estimated from $NO_3^-$ isotopes using Eq. (2), and $\frac{NO_3\text{-}N_i}{Cl_i}$ was required to be within one standard deviation of the estimate from Eq. (7).

The resulting estimates of $f_m$ are reported as the mid-range, with uncertainty described by the minimum and maximum values. Larger values of $f_m$ indicate less mixing (a shorter path for advection-dispersion) and suggest a source close to the well. Smaller values of $f_m$ indicate extensive mixing (a longer path for advection-dispersion) and suggest a source further away from the well. The relative contributions of mixing and denitrification to $NO_3^-$ attenuation at each site were evaluated by comparing $f_m$ and $f_d$ for each sample. This analysis was conducted using isotope values from the samples collected on 1 May 2013 at CFO1, which were combined with the Cl$^-$ and $NO_3$-N data from 6 June 2013. At CFO4, results from stable isotopes collected on 27 October 2014 were combined with Cl$^-$ and $NO_3$-N data collected on 7 October 2014.

## 3. Results

### 3.1 Site hydrogeology

#### 3.1.1 CFO1

The geology at CFO1 consists of clay and clay-till interspersed with sand layers of varying thickness to the maximum depth of investigation (20 m BG, bedrock not encountered). Hydraulic conductivities ($K$) calculated from slug tests on wells ranged from $1.2 \times 10^{-7}$ to $4.2 \times 10^{-5}$ m-s$^{-1}$ (n=10) for sand, $1.1 \times 10^{-8}$ to $2.8 \times 10^{-8}$ m s$^{-1}$ (n=2) for clay-till, and $1.6 \times 10^{-9}$ to $3.0 \times 10^{-7}$ m s$^{-1}$ (n=8) for clay. Depth to the water table throughout the study site ranged from 0.5 m at DMW14 to 3.8 m at DMW11. Seasonal water table variations were about 0.5 m with no obvious change in the annual average during the 6-year measurement period. Water table elevation was highest at DMW10 and DMW1 on the west side of the site and lowest at DMW11 on the northeast side of the site (see Supplementary Material). Measured heads indicate groundwater flow from the vicinity of the EMS to the northeast and southeast. Mean horizontal hydraulic gradients at the water table ranged from $4.4 \times 10^{-3}$ to $1.4 \times 10^{-2}$ m m$^{-1}$. Vertical gradients were predominantly downward in the upper 20 m of the profile (mean gradients ranging from $1.8 \times 10^{-3}$ to $0.18$ m m$^{-1}$), with the exception of DMW11 where the vertical gradient was upward (mean gradient $-2.8 \times 10^{-2}$ m m$^{-1}$). Using the geometric mean $K$ for the sand ($5.0 \times 10^{-6}$ m s$^{-1}$) and a lateral head gradient of $1.4 \times 10^{-2}$ m m$^{-1}$ yields a specific discharge (Darcy flux, $q$) of 2.2 m y$^{-1}$. Assuming an effective porosity of 0.3 (Rodvang et al., 1998), the average linear velocity ($\bar{v}$) is 7.4 m y$^{-1}$. This suggests that, in the absence of attenuation by mixing or denitrification, agriculturally derived $NO_3^-$ could have been transported through the groundwater system by advection about 400 m from the EMS since 1960 and 630 m since 1930.

#### 3.1.2 CFO4

The geology at CFO4 consists of about 5 m of clay (with minor till) underlain by sandstone, to the maximum depth investigated (20 m BG). Hydraulic conductivities measured using slug tests on wells were $1.0 \times 10^{-8}$ to $1.0 \times 10^{-5}$ m s$^{-1}$ (n=12) for the clay and sandstone (many shallow wells were screened across the clay-till and into

the sandstone) and $1.0 \times 10^{-5}$ to $2.9 \times 10^{-5}$ m s$^{-1}$ (n=4) for the sandstone. The depth to water table ranged from 1.0 to 3.4 m, increasing from west to east across the study site. Seasonal water table variations were on the order of 1.5 m with water table declines on the order of 0.3 m y$^{-1}$. The horizontal hydraulic gradient was consistently from west to east, with a mean gradient at the water table of $3.9 \times 10^{-3}$ m m$^{-1}$ between BC2 and BMW2 and $4.3 \times 10^{-3}$ m m$^{-1}$ between BMW2 and BMW7. Vertical hydraulic gradients were $4.2 \times 10^{-2}$ to $4.6 \times 10^{-2}$ m m$^{-1}$ downward. Using the geometric mean $K$ for the site ($2.9 \times 10^{-5}$ m s$^{-1}$) and a lateral head gradient of $4.3 \times 10^{-3}$ m m$^{-1}$ yields a $q$ of 0.4 m y$^{-1}$. Assuming an effective porosity of 0.3 yields a $\bar{v}$ of 1.3 m y$^{-1}$. These values suggest that, in the absence of attenuation by mixing or denitrification, anthropogenic NO$_3^-$ could have been transported through the groundwater systems about 10 m by advection between 1995 and the time of sampling.

## 3.2 Values and evolution of stable isotopes of nitrate

The range of isotopic values of NO$_3^-$ in groundwater was similar at both sites (Fig. 2). At CFO1, $\delta^{18}$O$_{NO3}$ ranged from -5.9 to 20.1‰ and $\delta^{15}$N$_{NO3}$ from -5.2 to 61.0‰. At CFO4, $\delta^{18}$O$_{NO3}$ ranged from -1.9 to 31.6‰ and $\delta^{15}$N$_{NO3}$ from -1.3 to 70.5‰. The isotopic values of $\delta^{18}$O$_{NO3}$ in groundwater are commonly assumed to be derived from a mix of a 1/3 atmospheric-derived oxygen (+23.5‰) and 2/3 water-derived oxygen (Xue et al., 2009). Given the average $\delta^{18}$O$_{H2O}$ for both sites (-16‰, see Supplementary Material), a 1/3 atmospheric 2/3 groundwater mix would result in a $\delta^{18}$O$_{NO3}$ of -3.7‰. Manure filtrate from the EMS at CFO1 had $\delta^{15}$N$_{NO3}$ ranging from 0.4 to 5.0‰ and $\delta^{18}$O$_{NO3}$ ranging from 7.1 to 19.0‰. A curve showing the co-evolution of $\delta^{18}$O$_{NO3}$ (mixing of atmospheric $\delta^{18}$O with groundwater-derived $\delta^{18}$O) and $\delta^{15}$N$_{NO3}$ (Rayleigh distillation, $\beta$ = 1.005) during nitrification is shown in Fig. 2. Isotopic values in DMW3, where direct leakage from the EMS was evident, are consistent with partial nitrification following this trend of isotopic evolution ($\delta^{18}$O$_{NO3}$ of -1.2‰ and $\delta^{15}$N$_{NO3}$ of 7.8‰).

At both sites, co-enrichment of $\delta^{18}$O$_{NO3}$ and $\delta^{15}$N$_{NO3}$ characteristic of denitrification was evident in some samples (slopes of 0.42 and 0.72 in Fig. 2a). At CFO1, this includes samples from DP10-2, DMW5, DMW11, DMW12, DP11-12b, and DMW13 (and associated core) and some pore water from cores DC15-22 and DC15-23. These samples had NO$_3$-N concentrations of 0.6 to 23.7 mg L$^{-1}$, $\delta^{18}$O$_{NO3}$ ranging from 4.8 to 20.6‰, and $\delta^{15}$N$_{NO3}$ ranging from 22.9 to 61.3‰. At CFO4, samples exhibiting evidence of denitrification were from BMW2, BMW5, BMW6, BMW7, and BC4. These samples had NO$_3$-N concentrations ranging from 0.4 to 35.1 mg L$^{-1}$, $\delta^{18}$O$_{NO3}$ ranging from 1.6 to 22.1‰, and $\delta^{15}$N$_{NO3}$ ranging from 20.9 to 70.1‰. Although the isotopic values of DMW5 suggest enrichment by denitrification, the data plot away from the rest of the CFO1 data and close to the denitrification trend at CFO4 (Fig. 2), suggesting these samples were affected by some other process (possibly mixing or nitrification); therefore, the fraction of NO$_3$-N remaining in this well was not calculated. Also, well DMW3, which clearly receives leakage from the EMS, did not contain substantial NO$_3$-N and so $f_d$ was not calculated.

In the Monte Carlo analysis the potential range of original isotopic values of the NO$_3^-$ source prior to denitrification ($\delta^{15}$N$_0$ and $\delta^{18}$O$_0$) varied from 5 to 27‰ for $\delta^{15}$N$_{NO3}$ and from -2 to 7‰ for $\delta^{18}$O$_{NO3}$ based on isotopic values measured during this study (Fig. 2a). These values are consistent with literature values for manure-sourced NO$_3^-$, which report $\delta^{15}$N$_{NO3}$ ranging from 5 to 25‰ and $\delta^{18}$O$_{NO3}$ ranging from -5 to 5‰ (Wassenaar, 1995; Wassenaar et al., 2006; Singleton et al., 2007; McCallum et al., 2008; Baily et al., 2011). $\varepsilon_{15N}$ was defined by a normal distribution with a mean of -10‰ and standard deviation of 2.5‰ (Fig. 2b). At CFO1, the coefficient of proportionality between the enrichment factor of $\delta^{15}$N$_{NO3}$ and $\delta^{18}$O$_{NO3}$ was described by a normal distribution with mean of 0.72 and standard deviation of 0.05. At CFO4, the coefficient of proportionality was also described by a

normal distribution with a mean of 0.42 and standard deviation of 0.035 (see Fig. 2a). These enrichment factors are consistent with values from denitrification studies that report $\varepsilon_{15N}$ ranging from -4.0 to -30.0‰ and $\varepsilon_{18O}$ ranging from -1.9 to -8.9‰ (Vogel et al., 1981; Mariotti et al., 1988; Böttcher et al., 1990; Spalding and Parrott, 1994; Mengis et al., 1999; Pauwels et al., 2000; Otero et al., 2009).

### 3.3 Distribution and sources of agricultural nitrate in groundwater

At both sites TN concentrations in filtrate from the EMS and catch-basin were generally an order of magnitude larger than concentrations in groundwater (Table 2). The one exception is well DMW3 at CFO1 which intercepted direct leakage from the EMS (see 3.3.1 for further discussion of this well). The dominant form of N differed between manure filtrate and groundwater. In the EMS filtrate, N was predominately organic-N (TON up to 71%)

or $NH_3$-N (up to 90%), with $NO_x$-N <0.1% of TN. In the catch-basin at CFO1 TON was >99% of TN. In groundwater TN concentrations ranged from <0.25 to 84.6 mg L$^{-1}$, and this N was predominantly $NO_3^-$ (again, with the exception of DMW3).

### 3.3.1 CFO1

Agriculturally derived $NO_3^-$ was generally restricted to the upper 20 m (or less) at CFO1 ($NO_3$-N $\leq$ 0.2 mg L$^{-1}$ and

15 Cl$^-$ $\leq$ 57 mg L$^{-1}$ in seven wells screened at 20 m). The one exception was DP11-12b, which had up to 4.1 mg L$^{-1}$ of $NO_3$-N. The southeast portion of the site also does not appear to have been significantly contaminated by agriculturally derived $NO_3^-$, with $NO_3$-N concentrations < 1 mg L$^{-1}$ in five water table wells (DMW4, DMW6, DMW14, DMW15, DMW16). In DMW6, Cl$^-$ and TN concentrations were elevated (see Supplementary Material) but $NO_3$-N concentrations were < 2 mg L$^{-1}$. Collectively, these data suggest the catch basin is not a significant

source of $NO_3^-$ to the groundwater at this site.

Leakage of manure slurry from the EMS at CFO1 is clearly indicated by the data from DMW3, which feature the highest concentrations of TN in groundwater (up to 548 mg L$^{-1}$) and elevated Cl$^-$, HCO$_3^-$, and DOC in concentrations similar to EMS manure filtrate (see Supplementary Material). Nevertheless, $NO_3$-N concentrations in this well were consistently low (1.1 $\pm$ 2.7 mg L$^{-1}$, n=22). The potential for nitrification in the vicinity of this

well is indicated by $NO_2$-N production (2.7 $\pm$ 8.3 mg L$^{-1}$, n=22). However, the data demonstrate that only a small proportion of the $NH_3$-N in DMW3 (373.4 $\pm$ 79.4 mg L$^{-1}$, n=22) could have been converted to $NO_3^-$ within the subsurface ($NO_3$-N in groundwater $\leq$ 66 mg L$^{-1}$). Further work is required to assess the importance of cation exchange as an attenuation mechanism for direct leakage from the EMS at this site.

Contamination by agricultural $NO_3^-$ that exceeds the drinking water guidelines ($NO_3$-N > 10 mg L$^{-1}$) was observed

in four wells (DMW1, DMW11, DMW13 and DP10-2) and in continuous core (DC15-23). DMW2 and DMW12 also had $NO_3$-N concentrations that were elevated but did not exceed the drinking water guideline ($\leq$ 3.7 mg L$^{-1}$). Given the evidence of partial nitrification in DMW3 (and low $NO_3$-N concentrations), the $NO_3$-N/Cl$^-$ ratio of contamination from the EMS was assumed to be best represented by DP10-2, which is located directly downgradient of the EMS. Data for this well indicate values of $NO_3$-N/Cl$^-$ predominantly ranging from 0.1 to 0.3

with $NO_3\text{-}N_i/Cl_i$ estimated at 0.3 $\pm$ 0.13 (Fig. 4).

The maximum $NO_3$-N concentration in groundwater at CFO1 (66.4 mg L$^{-1}$) was measured in core sample DC15-23 (clay at 2 m bgl, 7 m hydraulically downgradient of DMW3). Pore water extracted from the unsaturated zone (sand) at the top of this core profile contained 865 mg L$^{-1}$ of $NO_3$-N and had a $NO_3$-N/Cl$^-$ ratio of 1.04,

consistent with the ratio of 0.95 in the core sample. Given this consistency, and that $NO_3$-N concentrations in the well immediately up-gradient were low (DMW3), the $NO_3$-N in this core sample was most likely introduced into the groundwater system by vertical infiltration or diffusion from above. In contrast, elevated $NO_3$-N (up to 21.1 mg L$^{-1}$) within the sand between 6 and 12 m depth in this core had $NO_3$-N/Cl$^-$ ratios consistent with an EMS source (0.07 to 0.31). Stable isotope values in pore water from this sand layer do not indicate substantial denitrification ($\delta^{18}$O $\leq$ 5.9‰, $\delta^{15}$N $\leq$ 16.7‰), suggesting these ratios will be similar to the initial ratios at the point of entry to the groundwater system.

In DMW13 (33 m downgradient from DP10-2) the ratio of $NO_3$-$N_i$/$Cl_i$ was 0.75 $\pm$ 0.29, similar to the $NO_3$-N/Cl$^-$ ratio in DC15-23 at 2 m (0.95), which is interpreted as reflecting a top-down source. The $NO_3^-$ in DMW13 is therefore unlikely to be sourced solely from leakage from the EMS, and could be sourced from the adjacent dairy pens or a temporary manure pile that was observed adjacent to this well during core collection in 2015 (or a combination of EMS and top-down sources).

In DMW12 the $NO_3$-$N_i$/$Cl_i$ ratio was not inconsistent with an EMS source, but the hydraulic gradient between DMW2 and DMW12 is negligible, indicating a lack of driving force for advective transport from the EMS towards DMW12. This is also the case for well DMW1, which is up-gradient of the EMS but had elevated $NO_3$-N concentrations (6.5 $\pm$ 3.6, n=18). The source of nitrate in these wells is therefore unlikely to be related to leakage from the EMS, but alternative sources (i.e., nearby temporary manure piles) are not known.

Well DMW11, 470 m from the EMS, had consistently low $NO_3$-N/Cl$^-$ ratios (< 0.05) similar to DP10-2, but estimates of $Cl_i$ were three-fold higher than $Cl_i$ for DP10-2 (Fig. 4b). $NO_3$-$N_i$ and $Cl_i$ estimated for DMW11 were consistent with measured values in that well, indicating a local top-down source. Well DMW11 is located hydraulically downgradient of feedlot pens and adjacent to a solid manure storage area, in a local topographic low. Elevated $NO_3$-N in this well is therefore interpreted to be from surface runoff and top-down infiltration, rather than lateral advection from the EMS.

### 3.3.2 CFO4

At CFO4, measured data indicate that effects from agricultural operations on $NO_3^-$ concentrations in groundwater are restricted to the upper 15 m of the subsurface. $NO_3$-N concentrations in wells screened at 15 m depth were < 0.5 mg L$^{-1}$, with the exception of one sample from BP10-15w (May 2012) with 4.3 mg L$^{-1}$ of $NO_3$-N. Water table wells in the west and north of the study site (BC1, BC2, and BC3) also indicate negligible impacts of agricultural operations, with Cl$^-$ < 10 mg L$^{-1}$ and $NO_3$-N < 0.1 mg L$^{-1}$.

Concentrations of $NO_3$-N >10 mg L$^{-1}$ were measured in three water table wells (BMW2, BMW3, BMW4) adjacent to the EMS, indicating that they have been impacted by the EMS (Fig. 5). Of these, BMW2 had much higher Cl$^-$ concentrations (502 $\pm$ 97 mg L$^{-1}$, n=22 in BMW2 compared to 182 $\pm$ 81 mg L$^{-1}$ in BMW3 and 188 $\pm$ 74 mg L$^{-1}$ in BMW4), and therefore lower $NO_3$-N/Cl$^-$ ratios (<0.05). Cl$^-$ concentrations in BMW2 were consistent with concentrations in the EMS suggesting direct leakage, while stable isotopes of $NO_3^-$ and initial concentrations ($NO_3$-$N_i$ $\geq$ 127 mg L$^{-1}$) indicate substantial denitrification (Table 2, Fig. 6). The $NO_3$-$N_i$/$Cl_i$ ratio in BMW2 is consistent with of measured $NO_3$-N/Cl$^-$ in BMW4, which therefore likely reflects leakage from the EMS without denitrification (consistent with stable isotope of values of $NO_3^-$).

Given that the estimated subsurface travel distance during operations at this site is 10 m, agriculturally derived $NO_3^-$ in other wells not immediately adjacent to the EMS is unlikely to be related to leakage from the EMS. Wells

BMW5 and BMW7 are 60 and 140 m hydraulically downgradient from the EMS, respectively. $NO_3\text{-}N_i/Cl_i$ ratios in these wells were not inconsistent with BMW2 (i.e., the range of values overlap), but given the distance from the EMS the source of $NO_3\text{-}N$ in these wells is most likely the adjacent dairy pens. Concentrations of $NO_3\text{-}N >$ 10 mg L$^{-1}$ were also measured in BC4, which is located 95 m hydraulically upgradient of the EMS. The ratio of $NO_3\text{-}N_i/Cl_i$ at BC4 was the highest at CFO4 (0.6) and did not overlap with BMW2. The $NO_3^-$ in this well is interpreted to have been sourced from an adjacent manure pile, which was observed during the study.

**3.4 Mechanisms of attenuation of agriculturally derived NO₃⁻**

Attenuation of agriculturally derived $NO_3^-$ in groundwater is dominated by denitrification at both CFO1 and CFO4, with estimates of $f_m$ consistently higher than estimates of $f_d$ (Table 3, Fig. 7, Table S10). Calculated $f_d$ values indicate that where denitrification was identified, at least half of the $NO_3\text{-}N$ present at the initial point of entry to the groundwater system has been removed by this attenuation mechanism. Comparison of $NO_3\text{-}N_{mix}$ (the concentration of $NO_3\text{-}N$ that would be measured if mixing was the only attenuation mechanism) with measured concentrations (which reflect attenuation by both mixing and denitrification) suggests that the sample from 20 m depth (DP11-12b) is the only sample that would be below the drinking water guideline if mixing was the only attenuation mechanism (Fig. 8).

At both sites, the stable isotope values of $NO_3^-$ indicate that denitrification proceeds within metres of the source. At CFO1, calculated $f_d$ in well DP10-2 (2 m from the EMS) is $0.52 \pm 0.22$; at CFO4, $f_d$ in well BMW2 (3 m from the EMS) is $0.13 \pm 0.06$. Denitrification also substantially attenuated $NO_3\text{-}N$ concentrations in wells where the source is not the EMS but instead is adjacent solid manure piles (e.g., DMW11 at CFO1, BC4 at CFO4). In BMW6 at CFO4, denitrification completely attenuated the agriculturally derived $NO_3^-$. This well had negligible $NO_3\text{-}N$ ($0.4 \pm 0.2$ mg L$^{-1}$, n=8) and the lowest $f_d$ of 0.01. Measured DOC in this well was consistent with other wells at both sites ($6.9 \pm 1.7$ mg L$^{-1}$, n=3), suggesting DOC depletion does not limit denitrification at these CFO operations.

**4. Discussion**

4.1 Implications for on-farm waste management

Agriculturally derived $NO_3^-$ at these two sites with varying lithology was generally restricted to depths < 20 m, consistent with previous studies at CFOs (Robertson et al., 1996; Rodvang and Simpkins, 2001; Rodvang et al., 2004; Kohn et al., 2016). Attenuation of agriculturally derived $NO_3^-$ in groundwater was a spatially varying combination of mixing and denitrification, with denitrification playing a greater role than mixing at both sites. In the samples for which $f_d$ could be determined, denitrification reduced $NO_3^-$ concentrations by at least half and, in some cases, back to background concentrations. Given that the range of source isotopic composition was allowed to vary to its maximum justifiable extent, these quantitative estimates of denitrification based on stable isotopes of $NO_3^-$ are likely to be conservative. Redox conditions within the groundwater system were not able to be determined in this study due to the sampling method used to collect groundwater from wells screened across low-K formations (well bailed dry then sample collected after water level recovery). However, denitrification appears to proceed within metres of the $NO_3^-$ source, suggesting relatively short sub-surface residence times are required and that redox conditions close to the water table are conducive to denitrification reactions (Critchley et al., 2014; Clague et al., 2015).

The substantial role of denitrification within the saturated glacial sediments at these study sites indicates the potential for significant attenuation of agriculturally derived $NO_3^-$ by denitrification in similar groundwater systems across the North American interior and Europe (Ernstsen et al., 2015; Zirkle et al., 2016). Denitrification in the unsaturated zone is limited by low water contents and oxic conditions, resulting in substantial stores of $NO_3^-$ in vadose zones (Turkeltaub et al., 2016; Ascott et al., 2017). $NO_3^-$ in water that is removed rapidly from site is also unlikely to be substantially attenuated by denitrification due to oxic conditions and rapid transit times (Ernstsen et al., 2015). Therefore, water management focussed on reducing the effects of $NO_3^-$ contamination in similar hydrogeological settings to this study should aim to maximize infiltration into the saturated zone where $NO_3^-$ concentrations can be naturally attenuated, provided that local groundwater isn't used for potable water supply.

At both sites there is evidence of elevated $NO_3^-$ due to leakage from the EMS, but the impact appears to be limited to within metres of the EMS. This suggests that saturation within the clay lining of the EMS has limited the development of extensive secondary porosity that would allow rapid water percolation (Baram et al., 2012). Infiltration of $NO_3^-$ rich water that has passed through temporary solid manure piles and dairy pens has resulted in groundwater $NO_3$-N concentrations as high as those associated with leakage from the EMS (e.g., DMW11, BC4). At CFO4, this is in spite of the presence of clay at surface, reflecting secondary porosity in the upper part of the profile that has led to hydraulic conductivities comparable to sand. This is consistent with the findings of Showers et al. (2008), who investigated sources of $NO_3^-$ at an urbanized dairy farm in North Carolina, USA. Construction of EMS facilities in Alberta has been regulated under the Agriculture Operation Practices Act since 2002, which requires them to be lined with clay to minimise leakage (Lorenz et al., 2014). On-farm waste management should increasingly focus on minimising temporary manure piles that are in direct contact with the soil to reduce $NO_3^-$ contamination associated with dairy farms and feedlots.

### 4.2 Critique of this approach and applicability at other sites

At both sites, leakage from the EMS had $NO_3$-$N_i$/$Cl_i$ of between 0.1 and 0.4, but this alone was not diagnostic of the source. The sources of manure-derived $NO_3^-$ (manure piles vs. EMS) are distinguishable based on $NO_3$-$N_i$/$Cl_i$ ratios, provided there is also an understanding of the history of each site, local hydrogeology, and potential sources. Calculated $f_d$ and $f_m$ generally decreased with increasing subsurface residence time and distance from source, providing additional evidence for source attribution. For example, at CFO4, well BMW2, which is adjacent to the EMS, had the highest $f_m$ (0.92), indicating the least attenuation of $NO_3$ by mixing and consistent with the EMS being the source of $NO_3^-$ to this well.

Calculation of $NO_3$-$N_i$/$Cl_i$ assumed that background concentrations could be neglected in the mixing model. At these study sites, background concentrations are likely to be < 20 mg L$^{-1}$ for Cl$^-$ and < 1 mg L$^{-1}$ for $NO_3$-N. Estimated $NO_3$-$N_i$ values were at least 20 times background $NO_3$-N concentrations, and over 100 times background concentrations in some wells. The estimated $Cl_i$ values were at least three times background concentrations at CFO1 and at least 10 times background concentrations at CFO4. The error introduced by neglecting background concentrations was assessed by comparing $f_m$ calculated with and without background concentrations included, using the full range of values in this study (Fig. 9). Neglecting background concentrations results in overestimation of $f_m$ (i.e. underestimation of the amount of attenuation mixing) with the largest errors when measured concentrations are close to background concentrations. For Cl$^-$ the maximum difference of 0.13

is in the mid-range of $f_m$ values. For NO$_3$-N, the difference is consistently < 0.1 with the largest errors at the lowest values of $f_m$. The uncertainty in $f_m$ is primarily related to uncertainty in the initial concentrations ($Cl_i$ and $NO_3$-$N_i$), which depends on measured Cl$^-$ and NO$_3$-N. The largest uncertainties in $NO_3$-$N_i$ and $Cl_i$ correspond to the lowest measured concentrations (i.e., furthest from the upper limit), with less uncertainty at higher measured concentrations as they approach the maximum values. Temporal variability in $NO_3$-$N_i$/$Cl_i$ for each source could not be determined based on the snapshot isotope sampling conducted, but this could be investigated by measuring NO$_3^-$ isotopes in conjunction with NO$_3$-N and Cl$^-$ at multiple times.

Although applicable at these sites, this approach may not be valid at other sites if additional sources of NO$_3$ in groundwater (e.g. fertilizer or nitrification) are significant, or if NO$_3$ concentrations in groundwater are naturally elevated (Hendry et al., 1984). The combination of the approach outlined here with measurement of groundwater age indicators would allow for better constraints on groundwater flow velocities and determination of denitrification rates (Böhlke and Denver, 1995; Katz et al., 2004; McMahon et al., 2004; Clague et al., 2015).

**4.3 Comparison with isotopic values of NO$_3^-$ in previous studies**

Nitrate isotope values in groundwater at the two CFOs studied were generally consistent with previous studies reporting denitrification of manure-derived NO$_3^-$ at dairy farms (Wassenaar, 1995; Wassenaar et al., 2006; Singleton et al., 2007; McCallum et al., 2008; Baily et al., 2011). However, the isotopic values of NO$_3^-$ in the manure filtrate from the EMS at CFO1, were not consistent with values for manure-sourced NO$_3^-$ reported in other groundwater studies (Wassenaar, 1995; Wassenaar et al., 2006; Singleton et al., 2007; McCallum et al., 2008a; Baily et al., 2011). This is likely to be because nitrification within the EMS was negligible (NO$_3$-N <0.7 mg L$^{-1}$), such that the isotopic values of NO$_3$-N in the manure filtrate reflect volatilization of NH$_3$ and partial nitrification within the EMS. $\delta^{18}$O$_{NO3}$ values may also have been affected by evaporative enrichment of the $\delta^{18}$O$_{H2O}$ being incorporated into NO$_3^-$ (Showers et al., 2008).

A number of groundwater samples collected during this study had relatively enriched $\delta^{18}$O$_{NO3}$ (> 15 ‰) with depleted $\delta^{15}$N$_{NO3}$ (< 15‰). Some of these isotopic values are within the range previously reported for NO$_3^-$ derived from inorganic fertilizer ($\delta^{15}$N$_{NO3}$ from -3 to 3‰ and $\delta^{18}$O$_{NO3}$ from -5 to 25‰), with the $\delta^{18}$O$_{NO3}$ depending on whether the NO$_3^-$ is from NH$_4^+$ or NO$_3^-$ in the fertilizer (Mengis et al., 2001; Wassenaar et al., 2006; Xue et al., 2009). To the best of our knowledge, however, no inorganic fertilizers have been applied at these study sites. Another potential source is NO$_3^-$ derived from soil organic N, but this should have $\delta^{15}$N$_{NO3}$ values of 0 to 10‰ and $\delta^{18}$O$_{NO3}$ values of -10 to 15‰ (Durka et al., 1994; Mayer et al., 2001; Mengis et al., 2001; Xue et al., 2009; Baily et al., 2011). Incomplete nitrification of NH$_4^+$ can result in $\delta^{15}$N$_{NO3}$ lower than the manure source (Choi et al., 2003), but as there was no measurable NH$_3$-N in these samples this is also unlikely. These isotope values may reflect the influence of NO$_3^-$ from precipitation, which usually has values ranging from -5 to 5‰ for $\delta^{15}$N$_{NO3}$ and 40 to 60‰ for $\delta^{18}$O$_{NO3}$, and has been reported to dominate NO$_3^-$ isotope values of groundwater under forested landscapes (Durka et al., 1994). Alternatively, they may be affected by microbial immobilization and subsequent mineralization and nitrification, which can mask the source $\delta^{18}$O$_{NO3}$ in aquifers with long residence times (Mengis et al., 2001; Rivett et al., 2008).

## 5. Conclusions

A mixing model constrained by quantitative estimates of denitrification from isotopes substantially improved our understanding of nitrate contamination at these sites. This novel approach has the potential to be widely applied as a tool for monitoring and assessment of groundwater in complex agricultural settings. $NO_3$-N concentrations in excess of the drinking water guideline were measured at both sites, with sources including manure piles, pens and the EMS. Even though these sites are dominated by clay-rich glacial sediments, the input of $NO_3^-$ to groundwater from temporary manure piles and pens resulted in comparable (or greater) $NO_3$-N concentrations than leakage from the EMS. This is attributed to the development of secondary porosity within unsaturated clays. Nitrate attenuation at both sites is dominated by denitrification, which is evident even in wells directly adjacent to the $NO_3^-$ source. In the wells for which denitrification was identified, concentrations of agriculturally-derived $NO_3^-$ had been reduced by at least half and, in some wells, completely. In the absence of denitrification all but one of these wells would have had $NO_3$-N concentrations above the drinking water guideline.

These results indicate that infiltration to groundwater systems in glacial sediments where $NO_3^-$ can be naturally attenuated is likely to be preferable to off-farm export via runoff or drainage networks, provided that local groundwater isn't a potable water source. On-farm management of manure waste at similar operations should increasingly focus on limiting manure piles that are in direct contact with the soil to limit $NO_3^-$ contamination of groundwater.

*Acknowledgements*

This research was supported by Alberta Agriculture and Forestry (AAF) and the Natural Resources Conservation Board (NRCB), who provided assistance with field work and laboratory analysis. Funding was also provided by a Natural Sciences and Engineering Research Council of Canada (NSERC) Industrial Research Chair (IRC) (184573) awarded to MJH. The authors thank Barry Olson at AAF for reviewing the manuscript. Our thanks also to the local producers, whose cooperation made this research possible.

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

**Table 1. Details of groundwater monitoring wells and continuous core collection at CFO1 and CFO4 (all screens installed at bottom of the well).**

| Site | Well/Core hole ID | Type[†] | Lateral distance from EMS* (m) | Ground elevation (m asl) | Total depth (m below ground) | Screen length (m) | Lithology of screened interval | $K$ (m s$^{-1}$) |
|---|---|---|---|---|---|---|---|---|
| CFO1 | DMW1 | WTW | 60 | 869.7 | 5.0 | 4.0 | Sand | |
| | DMW2 | WTW | 10 | 867.2 | 6.0 | 4.0 | Sand | $1.2 \times 10^{-7}$ |
| | DMW3 | WTW | 2 | 867.5 | 3.7 | 2.0 | Sand | |
| | DMW4 | WTW | 160 | | 4.2 | 4 | Sand | $1.3 \times 10^{-6}$ |
| | DMW5 | WTW | 270 | 866.4 | 6.8 | 4.0 | Clayey sand | $1.7 \times 10^{-5}$ |
| | DMW6 | WTW | 310 | | 6.7 | 4 | | |
| | DP10-1 | Piezo | 2 | 867.8 | 18.6 | 0.5 | Clay | $1.6 \times 10^{-9}$ |
| | DP10-2 | Piezo | 2 | 867.9 | 8.0 | 1.5 | Sand | $3.6 \times 10^{-5}$ |
| | DMW10 | WTW | 340 | 868.0 | 7.2 | 3.0 | Clay | $3.0 \times 10^{-7}$ |
| | DP11-10b | Piezo | 340 | 868.0 | 20 | 0.5 | Clay | $2.2 \times 10^{-8}$ |
| | DMW11 | WTW | 470 | 864.8 | 7.0 | 3.0 | Sand and clay | $4.2 \times 10^{-5}$ |
| | DP11-11b | Piezo | 470 | | 20 | 0.5 | Clay | $6.3 \times 10^{-9}$ |
| | DMW12 | WTW | 50 | 867.6 | 7.0 | 3.0 | Sand and clay | $7.4 \times 10^{-6}$ |
| | DP11-12b | Piezo | 50 | 867.6 | 20.1 | 1.0 | Clay | $1.1 \times 10^{-8}$ |
| | DMW13 | WTW | 35 | 867.1 | 7.0 | 3.0 | Sand | $8.9 \times 10^{-6}$ |
| | DP11-13b | Piezo + core | 35 | 867.1 | 20.0 | 0.5 | Clay | |
| | DMW14 | WTW | 105 | 865.7 | 7.0 | 3.0 | Clay | $5.7 \times 10^{-6}$ |
| | DP11-14b | Piezo | 105 | 865.7 | 20.0 | 0.5 | Sand | $1.1 \times 10^{-6}$ |
| | DMW15 | WTW | 185 | | 7.0 | 3 | Clay | $2.4 \times 10^{-8}$ |
| | DP11-15b | Piezo | 185 | | | 20.0 | 0.5 | Clay | $1.4 \times 10^{-7}$ |
| | DMW16 | WTW | 320 | 866.0 | 6.0 | 3.0 | Sand and clay | - |
| | DP11-16b | Piezo | 320 | | 20.0 | 0.5 | Clay | $3.2 \times 10^{-9}$ |
| | DC15-20 | Core | 76 | | 15 | | | |
| | DC15-21 | Core | 45 | | 10.5 | | | |
| | DC15-22 | Core | 22 | | 12 | | | |
| | DC15-23 | Core | 9 | | 15 | | | |
| CFO4 | BC1 | WTW | 110 | 857.0 | 6.9 | 3.1 | Clay and sandstone | |
| | BC2 | WTW | 365 | 859.4 | 7.0 | 3.1 | Clay and sandstone | $2.2 \times 10^{-7}$ |
| | BC3 | WTW | 145 | 858.6 | 6.8 | 3.1 | Clay and sandstone | $1.3 \times 10^{-6}$ |
| | BC4 | WTW | 95 | 858.8 | 5.9 | 3.0 | Clay and sandstone | $3.4 \times 10^{-6}$ |
| | BC5 | WTW | 105 | 859.5 | 7.5 | 4.5 | Clay and sandstone | |
| | BMW1 | WTW | 4 | 858.6 | 7.1 | 3.1 | Clay and sandstone | $4.3 \times 10^{-6}$ |
| | BMW2 | WTW | 3 | 857.9 | 7.5 | 4.5 | Clay and sandstone | $8.5 \times 10^{-7}$ |
| | BMW3 | WTW | 8 | 858.6 | 6.0 | 3.0 | Clay and sandstone | |
| | BMW4 | WTW | 14 | 858.0 | 7.5 | 4.8 | Clay and sandstone | $1.0 \times 10^{-5}$ |
| | BMW5 | WTW | 60 | 858.0 | 7.5 | 4.5 | Clay and sandstone | |
| | BP5-15 | Piezo | 60 | 858.1 | 15.3 | 1.5 | Sandstone | $1.0 \times 10^{-7}$ |
| | BMW6 | WTW | 150 | 856.9 | 7.5 | 4.5 | Clay and sandstone | $4.0 \times 10^{-6}$ |
| | BP6-15 | Piezo | 150 | 856.8 | 15.2 | 1.5 | Sandstone | $3.0 \times 10^{-6}$ |
| | BMW7 | WTW | 140 | 856.7 | 7.5 | 4.5 | Clay and sandstone | $1.0 \times 10^{-6}$ |
| | BP10-15e | Piezo | 4 | 858.2 | 14.9 | 1.5 | Sandstone | $2.9 \times 10^{-5}$ |
| | BP10-15w | Piezo | 10 | 858.0 | 15.0 | 1.5 | Sandstone | $1.0 \times 10^{-5}$ |

*EMS=Earthen manure storage

[†]WTW=water table well, Piezo = piezometer, Core = continuous core

**Table 2. Range of measured concentrations of TN, NH$_3$-N, NO$_x$-N (NO$_2$-N + NO$_3$-N) and TON at each study site. At CFO1 results from monitoring well DMW3 are presented separately because values in this well differed substantially from all other wells.**

| Site | N-pool | TN (mg L$^{-1}$) | NH$_3$-N (mg L$^{-1}$) | NO$_x$-N (mg L$^{-1}$) | TON (mg L$^{-1}$) |
|---|---|---|---|---|---|
| CFO1 | EMS | 550 – 1820 | 275 – 747 | <0.1 – 0.4 | 73 – 1301 |
| | Catch-basin | 200 – 1440 | 2.5 – 7.3 | <0.1 | 196 – 1437 |
| | DMW3 | 278 – 548 | 219 – 479 | <0.1 – 50[*] | 31.3 – 73.9 |
| | Other monitoring wells | <0.25 – 33.4 | <0.05 – 2.9 | <0.1 – 31.4[**] | <0.2 –3.7 |
| CF04 | EMS[^] | 1000 – 1240 | 724 – 747 | 0.25 - 0.29 | 275 –492 |
| | Monitoring wells | <0.25 – 84.6 | <0.05 – 0.23 | <0.1 – 80.4 | <0.2 –13.9 |

[*] NO$_x$-N of 50 mg L$^{-1}$ in DMW3 consisted of 12.6 mg L$^{-1}$ as NO$_3$-N and 37.4 mg L$^{-1}$ as NO$_2$-N.
[**] NO$_x$-N max in groundwater measured in core (NO$_3$-N = 66.4 mg L$^{-1}$, NO$_x$-N = 67.8 mg L$^{-1}$)
[^] Range across three replicates measured on 25 August 2011

**Table 3. Calculated $f_d$ and $f_m$ based on measured Cl$^-$ and NO$_3$-N concentrations and stable isotope values of NO$_3^-$.**

| Study area | Sample ID* | Cl$^-$ (mg L$^{-1}$) | NO$_3$-N (mg L$^{-1}$) | δ$^{15}$N$_{NO3}$ (‰) | δ$^{18}$O$_{NO3}$ (‰) | $f_d$ (mean ± stdev) | $f_m$[**] (mid-range) |
|---|---|---|---|---|---|---|---|
| CFO1 | DP11-13_4.3m | 28.5 | 7.0 | 30.3 | 9.8 | 0.30 ± 0.15 | 0.58 |
| | DP11-13_5.2m | 25.0 | 7.8 | 31.0 | 10.8 | 0.34 ± 0.13 | 0.58 |
| | DP11-13_7m | 72.3 | 12.0 | 31.6 | 10.2 | 0.27 ± 0.13 | 0.65 |
| | DP11-13 _7.9m | 70.8 | 9.1 | 36.4 | 14.0 | 0.17 ± 0.09 | 0.68 |
| | DP11-13_8.8m | 81.7 | 10.9 | 29.6 | 9.9 | 0.32 ± 0.15 | 0.63 |
| | DC15-22_10m | 73.0 | 11.0 | 26.1 | 7.4 | 0.47 ± 0.21 | 0.63 |
| | DP10-2 | 74.5 | 11.8 | 24.2 | 4.8 | 0.52 ± 0.22 | 0.63 |
| | DMW11 | 436.1 | 17.1 | 33.3 | 10.9 | 0.17 ± 0.07 | 0.83 |
| | DMW12 | 78.0 | 2.57 | 29.8 | 14.3 | 0.23 ± 0.10 | 0.54 |
| | DMW13 | 56.7 | 23.7 | 23.0 | 6.8 | 0.56 ± 0.22 | 0.65 |
| | DP11-12b | 95.7 | 0.6 | 35.9 | 17.0 | 0.15 ± 0.08 | 0.54 |
| CFO4 | BC4 | 163.1 | 35.1 | 30.6 | 1.6 | 0.37 ± 0.13 | 0.82 |
| | BMW2 | 595.6 | 16.5 | 41.6 | 8.3 | 0.13 ± 0.06 | 0.92 |
| | BMW5 | 131.2 | 12.9 | 28.9 | 6.5 | 0.34 ± 0.16 | 0.63 |
| | BMW6 | 156.0 | 0.4 | 70.5 | 22.1 | 0.01 ± 0.01 | 0.56 |
| | BMW7 | 134.7 | 11.6 | 34.0 | 5.9 | 0.21 ± 0.11 | 0.68 |

*central depth of core samples, x, indicated as SampleID_xm.
** maximum $f_m$ is 1 for all samples, which implies no mixing.

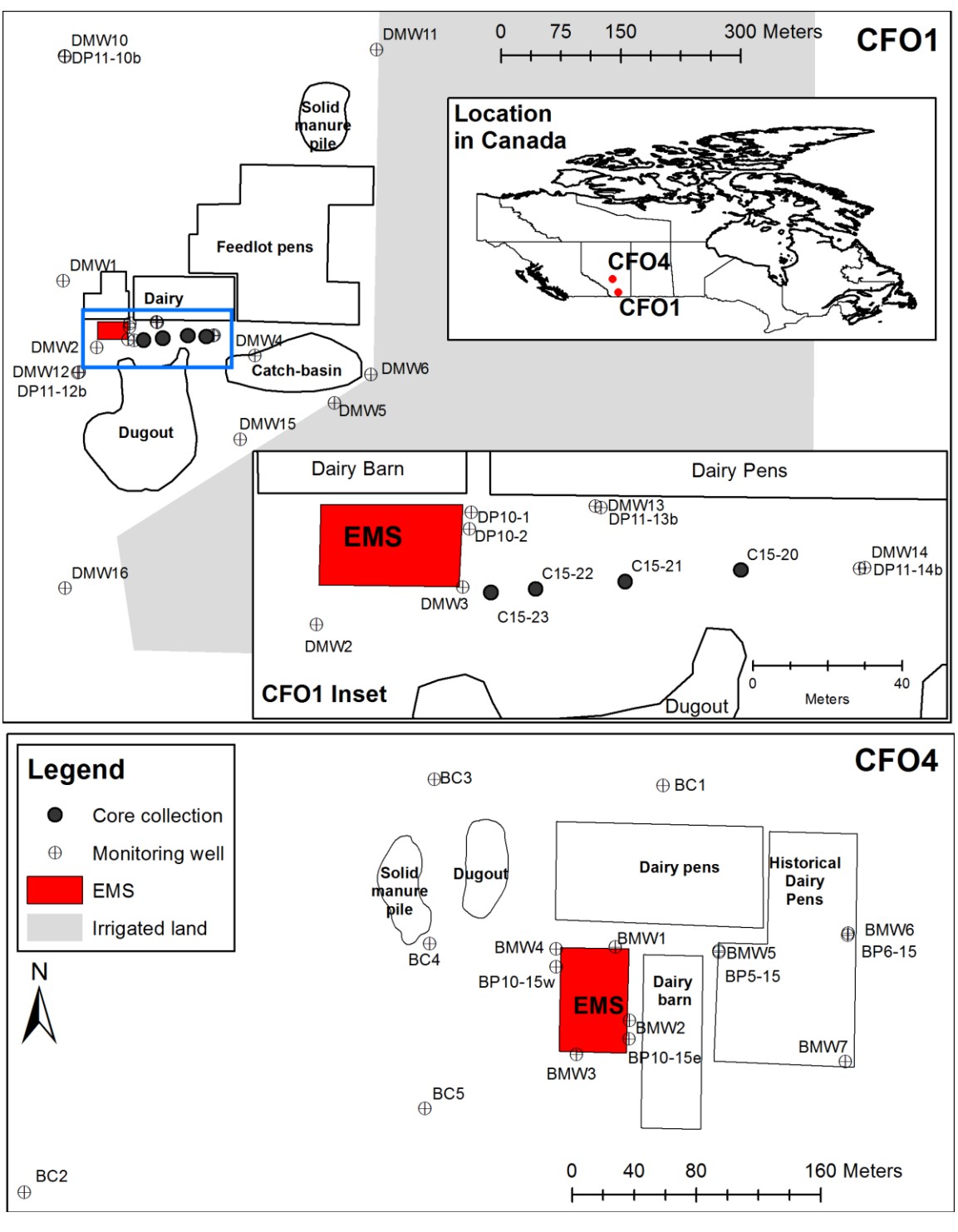

**Figure 1: Map of study sites CFO1 and CFO4, showing locations of groundwater monitoring wells, core collection, earthen manure storages (EMS), dairy and feedlot pens, manure piles, and irrigated land. Blue rectangle indicates extent of CFO1 inset.**

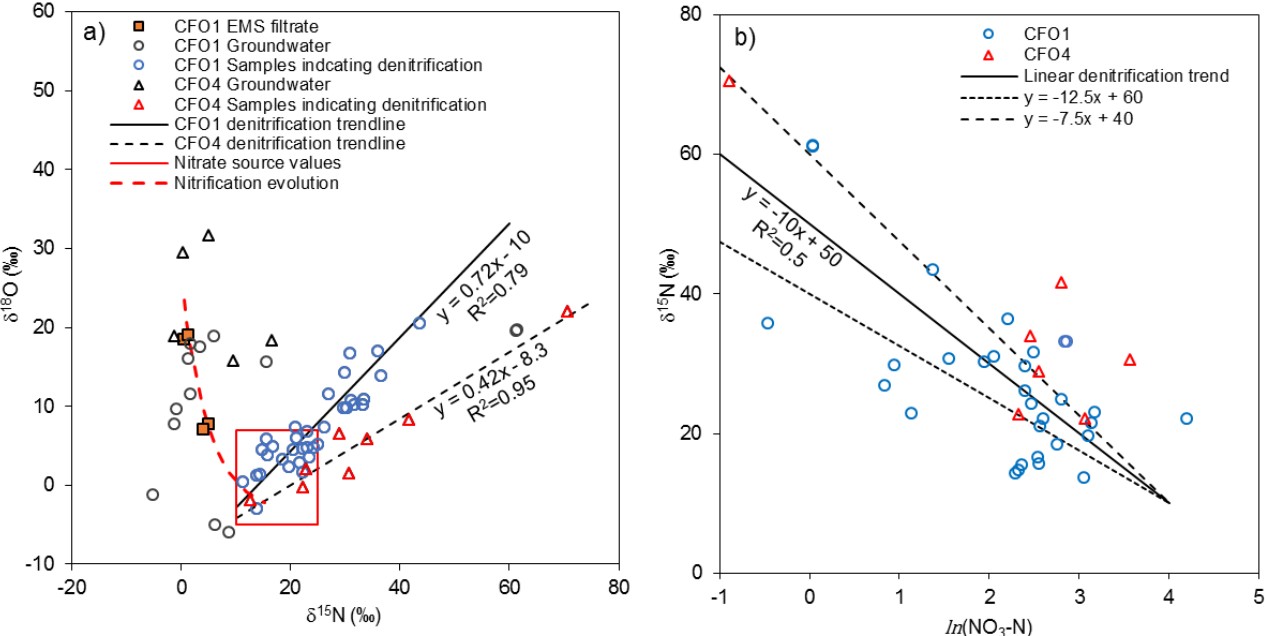

**Figure 2 (a) Cross-plot of stable isotopes of nitrate at CFO1 and CFO4 showing hypothetical nitrification trend, boundary of manure-sourced NO₃⁻ values and linear enrichment trends associated with denitrification, (b) enrichment of $\delta^{15}N_{NO3}$ during denitrification (only samples within source region and with evidence of denitrification are shown) dashed lines represent ±1 std. dev. of enrichment factor ($\varepsilon$ = -10) estimated from measured data.**

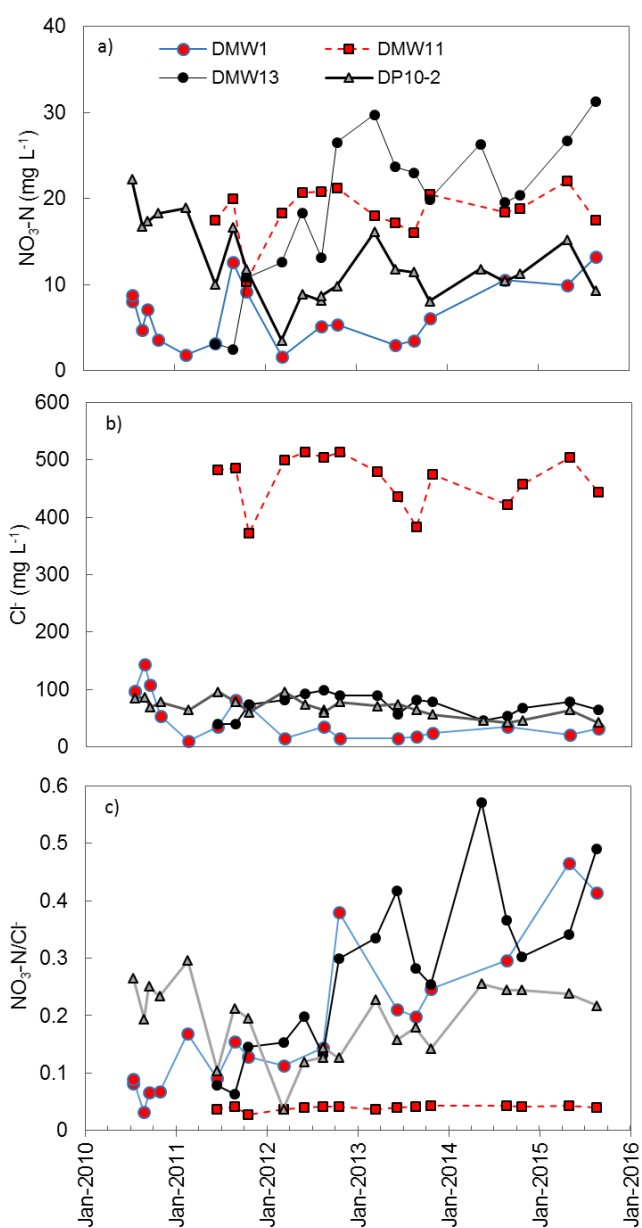

**Figure 3 Temporal variations in (a) NO₃-N, (b) Cl⁻, and (c) NO₃-N/Cl⁻ at CFO1. Only wells with NO₃-N > 10 mg L⁻¹ are shown.**

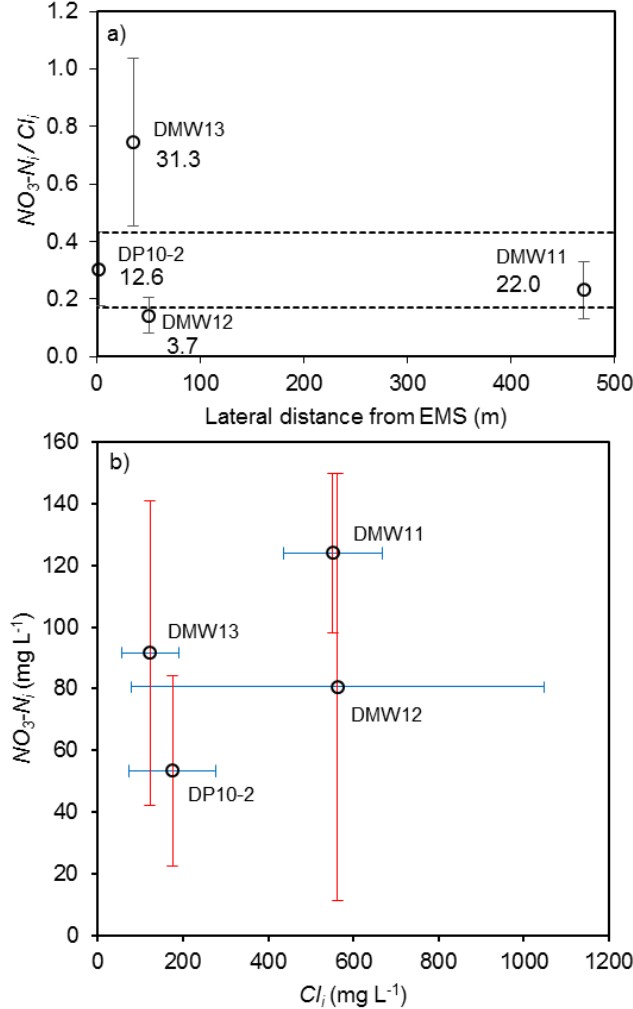

**Figure 4 (a) Estimated $NO_3$-$N_i$/$Cl_i$ ratios (mean and st. dev.) in water table wells with evidence of denitrification at CFO1, plotted with distance from earthen manure storage (EMS), where dashed lines are the upper and lower bounds of DP10-2 (EMS source) and values are maximum measured $NO_3$-N (mg L$^{-1}$). (b) Estimated concentrations of $NO_3$-$N_i$ and $Cl_i$ at CFO1 (mid-range, error bars are max. and min. values).**

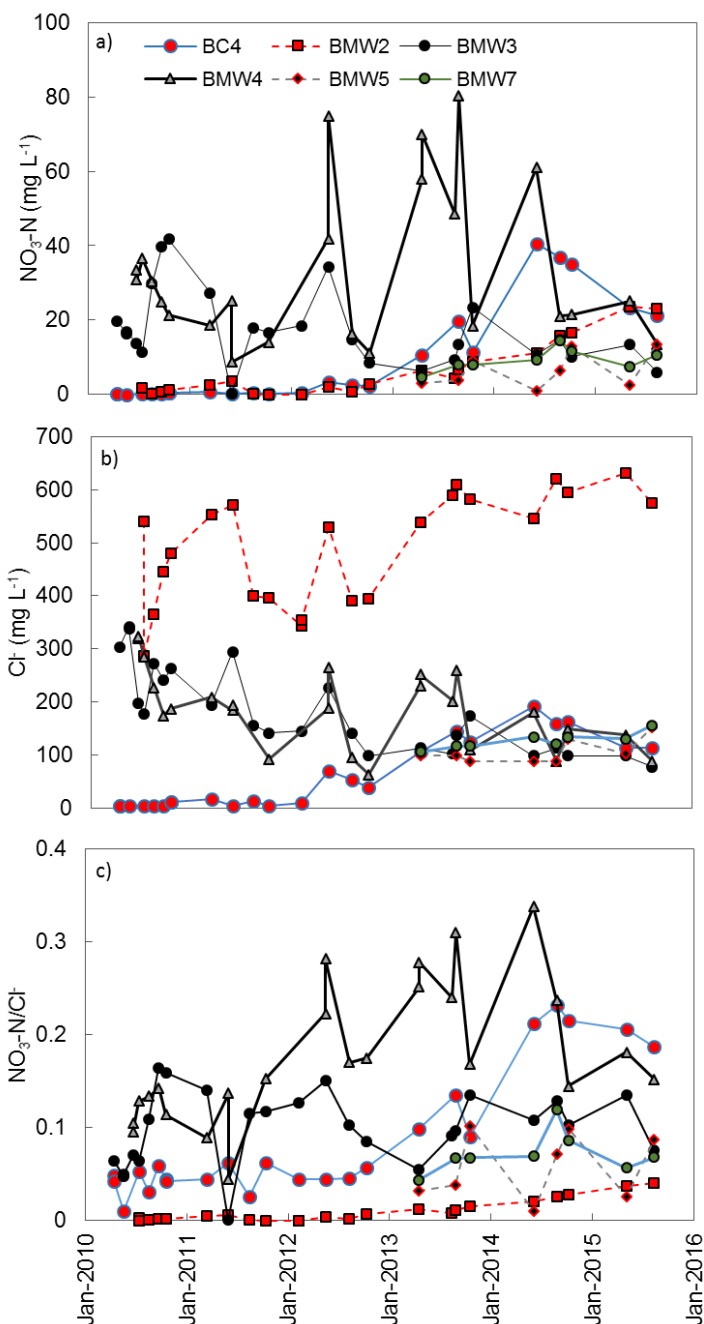

**Figure 5 Temporal variations in (a) NO₃-N, (b) Cl⁻, and (c) NO₃-N/Cl⁻ at CFO4. Only wells with NO₃-N > 10 mg L⁻¹ are shown.**

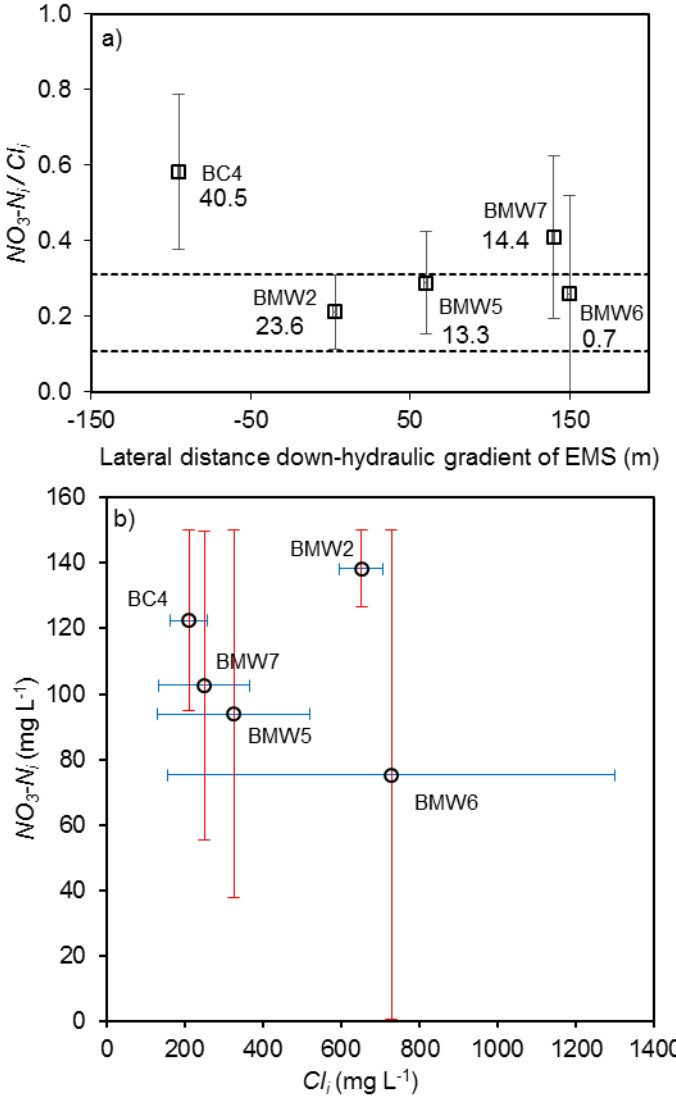

**Figure 6 (a) Estimated** $NO_3$-$N_i$/$Cl_i$ **ratios (mean and st. dev.) in water table wells with evidence of denitrification at CFO4, plotted with distance from earthen manure storage (EMS), where dashed lines are upper and lower bounds of BMW2 (EMS source) and values are maximum measured** $NO_3$-N **(mg L$^{-1}$). (b) Estimated concentrations of** $NO_3$-$N_i$ **and** $Cl_i$ **at CFO1 (mid-range, error bars are max. and min. values).**

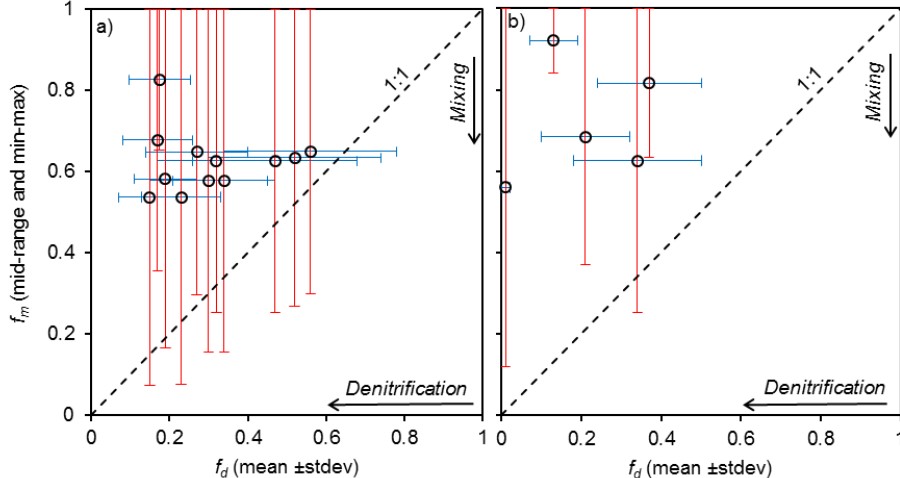

**Figure 7 Relative contributions to NO₃⁻ attenuation by mixing and denitrification, as indicated by estimated $f_m$ and $f_d$ at (a) CFO1 and (b) CFO4, for groundwater samples with denitrification indicated by stable isotope values of NO₃⁻.**

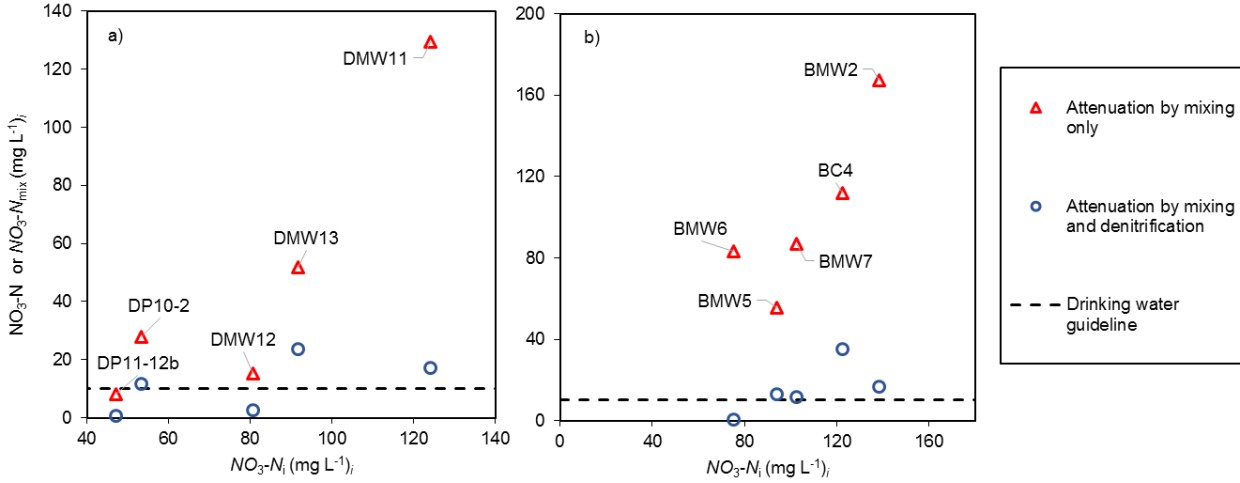

**Figure 8 Measured concentrations of NO₃-N (blue circles - attenuation by mixing and denitrification) and $NO_3$-$N_{mix}$**
10 **(red triangles - attenuation by mixing only) vs mid-range estimate of NO₃-$N_i$ at a) CFO1 and b) CFO4. Dashed lines are drinking water guideline (10 mg L⁻¹ of NO₃-N).**

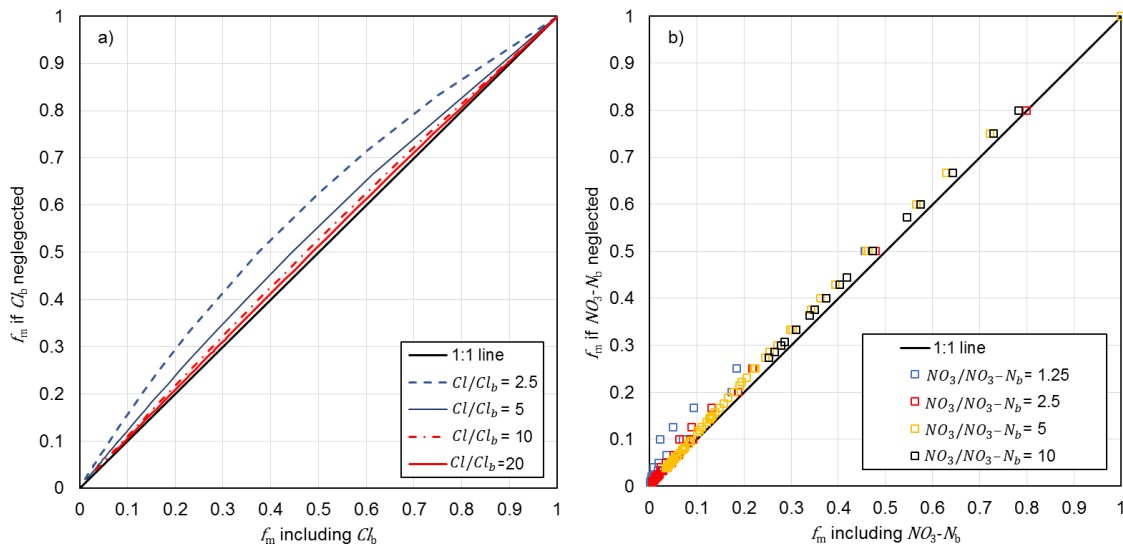

**Figure 9 Effect of neglecting background concentrations ($Cl_b$ or $NO_3$-$N_b$) in the mixing model on calculated $f_m$ over the range of values in this study.**

# Sources and fate of nitrate in groundwater at agricultural operations overlying glacial sediments

Sarah A. Bourke[1,2], Mike Iwanyshyn[3], Jacqueline Kohn[4], M. Jim Hendry[1]

[1]Department of Geological Sciences, University of Saskatchewan, SK, S7N 5C9, Canada
[2]School of Earth Sciences, University of Western Australia, Crawley, WA, 6009, Australia
[3]Natural Resources Conservation Board, Calgary, AB, T2P 0R4, Canada
[4]Alberta Agriculture and Forestry, Irrigation and Farm Water Branch, Edmonton, AB, T6H 5T6, Canada

*Correspondence to*: Sarah A. Bourke (sarah.bourke@uwa.edu.au)

**Abstract.** Leaching of nitrate ($NO_3^-$) from animal waste or fertilizers at agricultural operations can result in $NO_3^-$ contamination of groundwater, lakes, and streams. Understanding the sources and fate of nitrate in groundwater systems in glacial sediments, which underlie many agricultural operations, is critical for managing impacts of human food production on the environment. Elevated $NO_3^-$ concentrations in groundwater can be naturally attenuated through mixing or denitrification. Here we use isotopic enrichment of the stable isotope values of $NO_3^-$ to quantify the amount of denitrification in groundwater at two confined feeding operations overlying glacial sediments in Alberta, Canada. Uncertainty in $\delta^{15}N_{NO3}$ and $\delta^{18}O_{NO3}$ values of the $NO_3^-$ source ~~source~~ and denitrification enrichment factors are accounted for using a Monte Carlo approach. When denitrification could be quantified, we used these values to constrain a mixing model based on $NO_3^-$ and $Cl^-$ concentrations. Using this novel approach we were able to reconstruct the initial $NO_3$-N concentration and $NO_3$-N/$Cl^-$ ratio at the point of entry to the groundwater system. Manure filtrate had total-nitrogen (TN) of up to 1820 mg L$^{-1}$, which was predominantly organic-N and $NH_3$. Groundwater had up to 85 mg L$^{-1}$ TN, which was predominantly $NO_3^-$. The addition of $NO_3^-$ to the local groundwater system from temporary manure piles and pens equalled or exceeded $NO_3^-$ additions ~~due to leaching~~ from earthen manure storages at these sites. On-farm management of manure waste ~~As such, on-farm management of manure waste~~ to limit ~~$NO_3^-$ contamination of groundwater s~~hould therefore increasingly focus on limiting manure piles in direct contact with the soil, and encourage storage in lined lagoons. Nitrate attenuation at both sites ~~is~~ is attributed to a spatially variable combination of mixing and denitrification, but is dominated by denitrification. ~~On site~~Where identified, denitrification reduced agriculturally-derived $NO_3^-$ concentrations by at least half and, in some wells, completely. ~~Therefore, i~~Infiltration to groundwater systems in glacial sediments where $NO_3^-$ can be naturally attenuated is likely preferable to off-farm export via runoff or drainage networks, especially if local groundwater is not used for potable water supply.

## 1 Introduction

The contamination of soil and groundwater with nitrate from agricultural operations is a global water quality issue that has been extensively documented (Power and Schepers, 1989; Spalding and Exner, 1993; Rodvang and Simpkins, 2001; Galloway et al., 2008; Zirkle et al., 2016; Arauzo, 2017; Ascott et al., 2017). Leaching of nitrate ($NO_3^-$) from animal waste or fertilizers can result in groundwater $NO_3^-$ concentrations that exceed drinking water guidelines and pose human health risks (Fan and Steinberg, 1996; Gulis et al., 2002; Yang et al., 2007). The

discharge of high-NO$_3^-$ groundwater, runoff, or drainage can contaminate streams and lakes, resulting in eutrophication and ecosystem decline (Deutsch et al., 2006; Kaushal et al., 2011). In saturated groundwater systems with low oxygen concentrations, elevated NO$_3^-$ can be naturally attenuated by microbial denitrification (Wassenaar, 1995; Robertson et al., 1996; Smith et al., 1996; Tesoriero et al., 2000; Singleton et al., 2007). Concentrations of NO$_3^-$ will also decrease along groundwater flow paths due to attenuation via dilution by hydrodynamic dispersion (referred to hereafter as mixing). Because of these natural attenuation mechanisms, infiltration to groundwater may be preferable to off-site drainage and runoff of nitrate-rich waters. Many agricultural operations are undertaken on fertile soils associated with glacial sediments (Spalding and Exner, 1993; Ernstsen et al., 2015; Zirkle et al., 2016). Understanding the sources and fate of agriculturally derived nitrate in groundwater systems in glacial sediments is therefore critical for managing impacts of human food production on the environment.

Identification of the sources and fate of NO$_3^-$ at agricultural operations can be challenging because of spatial and temporal variations in sources (e.g. earthen manure storage, temporary manure piles, or fertilizer) and ~~the complexity in~~of ~~of~~ hydrogeologic systems (Spalding and Exner, 1993; Rodvang et al., 2004; Showers et al., 2008; Kohn et al., 2016). These spatial and temporal variations can result in complex subsurface solute distributions that are difficult to interpret using classical transect studies or numerical groundwater models (Green et al., 2010; Baily et al., 2011).

Groundwater containing significant agriculturally derived NO$_3^-$ also typically has elevated chloride (Cl$^-$) concentrations (Saffigna and Keeney, 1977; Rodvang et al., 2004; Mencíó et al., 2016). Decreasing NO$_3$-N/Cl$^-$ (or NO$_3^-$/Cl$^-$) ratios have been used to define denitrification based on the assumption that NO$_3^-$ is reactive while Cl$^-$ is non-reactive (conservative), such that denitrification results in a decrease in the NO$_3$-N/Cl$^-$ ratio (Kimble et al., 1972; Weil et al., 1990; Liu et al., 2006; McCallum et al., 2008). However, NO$_3$-N/Cl$^-$ ratios can also change in response to mixing of groundwater with different NO$_3$-N/Cl$^-$ ratios or when groundwater sampling traverses hydraulically disconnected formations (Bourke et al., 2015b). If NO$_3$-N/Cl$^-$ ratios vary among potential sources and the NO$_3$-N/Cl$^-$ ratio at the point of entry to the groundwater system can be reconstructed, this information could be used to show that anthropogenic NO$_3^-$ at different locations within an aquifer is derived from the same or different sources.

The stable isotopes of NO$_3^-$ ($\delta^{15}$N$_{NO3}$ and $\delta^{18}$O$_{NO3}$) provide an alternative approach to characterize the source and fate of NO$_3^-$ in groundwater systems. In agricultural areas, multiple sources of NO$_3^-$ are common and could include precipitation, soil NO$_3^-$, inorganic fertilizer, manure, and septic waste (Komor and Anderson, 1993; Liu et al., 2006; Pastén-Zapata et al., 2014; Clague et al., 2015; Xu et al., 2015). While source identification is theoretically possible using $\delta^{15}$N$_{NO3}$ and $\delta^{18}$O$_{NO3}$ (particularly with a dual-isotope approach), in practice this can be difficult due to geologic heterogeneity, overlapping source values, and the complexity of biologically mediated reactions (Aravena et al., 1993; Wassenaar, 1995; Mengis et al., 2001; Choi et al., 2003; Granger et al., 2008; Vavilin and Rytov, 2015; Xu et al., 2015).

NO$_3^-$ attenuation by denitrification in groundwater systems can be identified based on the characteristic enrichment of $\delta^{15}$N$_{NO3}$ and $\delta^{18}$O$_{NO3}$. Numerous studies have made qualitative assessments that identified denitrification in groundwater using the stable isotope approach (Böttcher et al., 1990; Wassenaar, 1995; Singleton et al., 2007; Baily et al., 2011; Clague et al., 2015; Xu et al., 2015). Recently published papers have also used stable isotopic values of NO$_3^-$ and water as the basis for mixing models in agricultural settings (Ji et al., 2017

;Lentz and Lehersch, 2019). Isotopic fractionation effects can also allow for quantitative assessment of the proportion of substrate that has undergone a given reaction, if enrichment factors and source values are known; as in the case of evpoarative loss of water, for example (Dogramaci et al., 2012). To date, there have been very few attempts to quantify denitrification using dual-isotope enrichment, largely due to uncertainty in source values and enrichment factors (Böttcher et al., 1990, Xue et al., 2009).

The only published calculations of the fraction of $NO_3^-$ remaining after denitrification the that we are aware of assumed a constant enrichment factor and the same isotopic source values across the field site (Otero et al., 2009). However, the enrichment factor will vary across a field site in response to reaction rates (Kendall and Aravena 2000), and isotopic values of even the same type of source (e.g. manure) can vary substantially (Xue et al., 2009). If the varation in source values and enrichment factors can be characterized from measured data then these uncertainties can be accounted for using a Monte Carlo approach (Joerin et al., 2002; Bourke et al., 2015a; Ji et al., 2017), thereby extending the application of the dual-isotope technique to allow for a robust quantitative assessment of denitrification in agricultural settings.

A synthesized analysis of stable isotopes of $NO_3^-$ with additional ionic tracers can further improve the assessment of $NO_3^-$ attenuation mechanisms and sources of $NO_3^-$ in agricultural settings (Showers et al., 2008; Vitòria et al., 2008; Xue et al., 2009; Xu et al., 2015; Ji et al., 2017). We hypothesise that if the amount of denitrification can be quantified based on $\delta^{15}N_{NO3}$ and $\delta^{18}O_{NO3}$, then this estimate of the fraction of $NO_3$-N removed through denitrification can be used to constrain a mixing model based on $NO_3$-N and $Cl^-$ concentrations. This novel approach allows for the ratio of $NO_3$-N/$Cl^-$ at the point of entry to the groundwater system to be reconstructed from measured $NO_3^-$ and $Cl^-$ concentrations (see Section 2.3). Where the $NO_3$-N/$Cl^-$ ratio varies between sources, this ratio can then be used to assess the source of the $NO_3^-$ in groundwater (e.g. temporary manure piles or feeding pens). These data can also then be used to estimate the initial concentrations of $NO_3^-$ and $Cl^-$ at the point of entry to the groundwater system and quantify attenuation by mixing.

In this study, we present the application of this approach at two confined feeding operations (CFOs) in Alberta, Canada, with differing lithologies and durations of operation (Fig. 1). Concentrations of $Cl^-$ and nitrogen species (N-species) and the stable isotopes of $NO_3^-$ were measured in groundwater samples collected from monitoring wells and continuous soil cores, as well as manure filtrate at both sites. These data were interpreted to (1) assess the extent of agriculturally derived $NO_3^-$ in groundwater, (2) identify sources and initial concentrations of $NO_3^-$ at the point of entry to the groundwater system, and (3) assess ~~the dominant~~mixing and denitrification as attenuation mechanisms ~~controlling subsurface $NO_3^-$ distributions~~ at these sites.

## 2 Materials and methods

### 2.1 Experimental sites

This study was conducted using data from two of the five sites investigated by Alberta Agriculture and Forestry during an assessment of the impacts of livestock manure on groundwater quality (Lorenz et al., 2014). To the best of our knowledge (including discussions with farm operators) fertilizers have not been applied at either of these sites. As such, manure waste from livestock is assumed to be the sole source of agricultural nitrogen (N) and elevated $NO_3^-$ concentrations in groundwater at these sites.

The first study site (CFO1) is located 25 km northeast of Lethbridge, Alberta (Fig. 1). Agricultural operations at this site were initiated with the construction of a dairy in 1928, which has the capacity for ~~, with the capacity for the~~150 dairy cattle ~~since the 1960s~~. A feedlot for beef cattle was added in 1960s along with an earthen manure storage (EMS) facility for storing liquid dairy manure (approx. 4 m deep) and a catch-basin that receives surface

water runoff. This feedlot was expanded in the 1980s to the 2000 head capacity it was at the time of this study. There is also a dugout (or slough, a shallow wetland) on site that receives local runoff and an irrigation drainage canal at the southern boundary of the property.

The second study site (CFO4) is located approximately 30 km north of Red Deer, Alberta and 300 km north of CFO1. This dairy and associated EMS (approx. 6 m deep) were constructed in 1995 and the facility had 350 head

of dairy cattle at the time of the study. Runoff will drain either to the small dugout in the north-west of the site, or the natural drainage features (ephemeral ponds or a creek approx. 1.5 km east).

**2.2 Sampling and instrumentation**

**2.2.1 Groundwater monitoring wells**

Groundwater samples were collected from water table wells and piezometers (hereafter both are referred to as

wells) installed at both sites (Table 1). At CFO1, groundwater samples were collected from six individual water table wells (DMW1, DMW2, DMW3, DMW4, DMW5, DMW6) and eight sets of nested wells with one well screened at the water table and one well screened 20 m below ground (BG) (DP10-2 and DP10-1, DMW10 and DP11-10b, DMW11 and DP11-11b, DMW12 and DP11-12b, DMW13 and DP11-13b, DMW14 and DP11-14b, DMW15 and DP11-15b, and DMW16 and DP11-16b). Wells DP10-2 and DP10-1 were located directly adjacent

to the EMS on the hydraulically downgradient side. At CFO4, groundwater samples were collected from eight water table wells (BC1, BC2, BC3, BC4, BC5, BMW1, BMW3, BMW7) and four sets of nested wells, with wells screened across the water table and at 15 m BG. Two of these nests were located adjacent to the EMS (BMW2 and BP10-15e, BMW4 and BP10-15w) and two were hydraulically downgradient of the EMS (BMW5 and BP5-15, BMW6 and BP6-15).

Groundwater samples were collected for ion analysis (Cl⁻ and N~~-~~ species) quarterly between April 2010 and August 2015. All water samples were collected using a bailer after purging (1–3 casing volumes) and stored at $\leq$ 4 °C prior to analysis. Samples for $\delta^{15}N_{NO3}$ and $\delta^{18}O_{NO3}$ were collected from wells at CFO1 on 1 January 2013 and 1 May 2013. Samples for $\delta^{15}N_{NO3}$ and $\delta^{18}O_{NO3}$ at CFO4 were collected on 27 October 2014. Wells were purged prior to sample collection (1–3 casing volumes), and samples filtered into high-density polyethylene

(HDPE) bottles in the field and frozen until analysis.

Hydraulic heads in monitoring wells were determined using manual measurements (approximately monthly, 2010-2015). ~~Rising~~ Hydraulic head ~~head~~ response tests ~~(slug or bail tests)~~ were conducted on the majority of the wells at the sites to determine hydraulic conductivity ($K$) of the formation media surrounding the intake zone ~~on the majority of the wells at the sites~~. These tests were either a slug test (water level decline after water addition),

or bail test (water level recovery after water removal) depending on the location of the water table within the well at the time of testing. K was determined from hydraulic the head responses using the method of Hvorslev (1951).

### 2.2.2 Continuous core

Continuous core was collected at CFO1 immediately adjacent to well DP11-13b on 1 May 2013 (Fig. 1). Additional core samples were collected from 1 to 5 June 2015 along a transect hydraulically downgradient of the southeastern side of the EMS at CFO1 where hydrochemistry data suggested leakage from the EMS (see Section 3). During this 2015 drilling campaign, core samples were collected at four locations (DC15-20, DC15-21, DC15-22, DC15-23) to depths of up to 15 m below surface and distances of up to 100 m from the EMS between wells DMW3 and DP11-14.

Continuous core samples were retrieved using a hollow stem auger (1.5-m core lengths) with 0.3-m sub-samples collected at approximately 1-m intervals ensuring that visually consistent lithology could be sampled. Core samples for $Cl^-$ were stored in Ziploc$^{TM}$ bags and kept cool until analysis. Core samples for N-species analysis were stored in Ziploc bags filled with an atmosphere of argon (99.9% Ar) to minimize oxidation and kept cool until analysis. Subsamples of each core (250-300 g) were placed under 50 MPa pressure in a Carver Series NE mechanical press with a 0.5-μm filter placed at the base of the squeezing chamber, which was placed within an Ar atmosphere to minimize oxidation. A syringe was attached to the base of the apparatus and 15 mL of filtered pore water were collected for analyses within 3.5 to 6.0 h (Hendry et al., 2013).

### 2.2.3 Liquid manure storages

Samples of liquid manure slurry were collected directly from the EMS at both sites and the catch basin (containing local runoff from the feedlot) at CFO1 using a pipe and plunger apparatus to sample from approximately 0.5 m below the surface. The slurry collected was subsequently filtered (0.45 μm) to separate the liquid and solid components. The water filtered from samples collected from the EMS or catch basin is hereafter referred to as manure filtrate.

### 2.3 Laboratory analysis

For gGroundwater samples from wells were analysed by Alberta Agriculture and Forestry (Lethbridge, Alberta). and manure filtrate, cConcentrations of $Cl^-$ were determined using potentiometric titration of $H_2O$, with a detection limit of 5.0 mg $L^{-1}$ and accuracy of 5% (APHA 4500-$Cl^-$ D). Concentrations of $NH_3$ as N ($NH_3$-N), $NO_3^-$ as N ($NO_3$-N), and $NO_2^-$ as N ($NO_2$-N) in groundwater samples from wells and manure filtrate were measured by air-segmented continuous flow analysis (APHA 4500-NH3 G, APHA 4500-NO3-F). Total nitrogen (TN) was determined by high temperature catalytic combustion and chemiluminescence detection using a Shimadzu TOC-V with attached TN unit (ASTM D8083-16). Total organic nitrogen (TON) was calculated by subtracting $NH_3$-N, $NO_3$-N and $NO_2$-N from TN. Bicarbonate ($HCO_3^-$) was analyzedanalysed by titration (APHA 2320 B). Dissolved organic carbon (DOC) was analyzedanalysed by a combustion infrared method (APHA 5310 B) using a Shimadzu TOC-V system. Manure filtrate was analysed by ALS (Saskatoon, Saskatchewan) using similar methods for $Cl^-$ (APHA 4110 B), TN (RMMA A3769 3.3), $NO_3$+$NO_2$ as N (APHA 4500-NO3-F), $NH_3$-N (APHA 4500-NH3 D), $HCO_3^-$ (APHA 2320) and DOC (APHA 5310 B).

Pore-water samples squeezed from continuous core were analyzedanalysed at the University of Saskatchewan (Saskatoon, Canada) for $Cl^-$, $NO_3$-N, and $NO_2$-N using a Dionex IC25 ion chromatograph (IC) coupled to a Dionex As50 autosampler (EPA Method 300.1, accuracy and precision of 5.0%) (Hautman and Munch, 1997). Ammonia as N ($NH_3$-N) was measured by Exova Laboratories using the automated phenate method (APHA Standard 4500-

NH3 G, detection limit of 0.025 mg L$^{-1}$, accuracy of 2% of the measured concentration, and a precision of 5% of the measured concentration).

$\delta^{15}N_{NO3}$ and $\delta^{18}O_{NO3}$ in groundwater samples (from wells and pore water from continuous core) and manure filtrate were measured at the University of Calgary (Calgary, Alberta) using the denitrifier method (Sigman et al., 2001) with an accuracy and precision of 0.3‰ for $\delta^{15}N_{NO3}$ and 0.3‰ for $\delta^{18}O_{NO3}$. Groundwater samples collected for NO$_3^-$ isotope analysis in January 2013 were also analyzed for NO$_3$-N by the University of Calgary (denitrifier technique, Delta+XL).

## 2.4 Modelling approach

### 2.4.1 Quantification of denitrification based on $\delta^{15}N_{NO3}$ and $\delta^{18}O_{NO3}$

Nitrate in groundwater that has undergone denitrification is commonly reported as being identified by enrichment of $\delta^{15}N_{NO3}$ and $\delta^{18}O_{NO3}$ with a slope of about 0.5 on a cross-plot (Clark and Fritz, 1997). However, published studies of denitrification in groundwater report slopes of up to 0.77 (Mengis et al., 1999; Fukada et al., 2003; Singleton et al., 2007). The relationship between isotopic enrichment of $\delta^{15}N_{NO3}$ and $\delta^{18}O_{NO3}$ and the fraction of NO$_3$-N remaining during denitrification can be described by a Rayleigh equation:

$$R = R_0 f_d^{\left(\frac{1}{\beta}-1\right)}, \tag{1}$$

where $R_0$ is the initial isotope ratio (relative to the standard) of the NO$_3^-$ ($\delta^{18}O_{NO3}$ or $\delta^{15}N_{NO3}$), $R$ is the isotopic ratio when fraction $f_d$ of NO$_3^-$ remains, and $\beta$ is the kinetic fractionation factor ($> 1$) (Böttcher et al., 1990; Clark and Fritz, 1997; Otero et al., 2009; Xue et al., 2009). Kinetic fraction effects are commonly also expressed as the enrichment factor, $\varepsilon = \frac{1}{1000(\beta-1)}$ ~~1000($\beta$ − 1)~~. In the case of a constant enrichment factor, $f_d$ can be calculated from measured $\delta^{15}N_{NO3}$ (or $\delta^{18}O_{NO3}$), if the initial $\delta^{15}N_{NO3}$ ($\delta^{15}N_0$) is known; ~~from:~~

$$f_d = exp\left(\frac{\delta^{15}N_{NO3} {\scriptstyle ^{15}NR} - \delta^{15}N_0 {\scriptstyle N_0R_0}}{\varepsilon}\right), \tag{2}$$

~~and t~~The fraction of NO$_3$-N removed from groundwater through denitrification is then given by (1-$f_d$). The concentration of NO$_3$-N that would have been measured if mixing was the only attenuation mechanism (NO$_3$-N$_{mix}$) can also be calculated by dividing the measured concentration by $f_d$.

A sub-set of 20 samples with isotopic values of NO$_3^-$ indicative of denitrification were identified, and for each of these samples $f_d$ (mean and standard deviation) was calculated from Eq. (2) using a Monte Carlo approach with 500 realizations. ~~The value of R was given by the measured isotopic ratio for each sample ($\delta^{18}O_{NO3}$ or $\delta^{15}N_{NO3}$). R$_0$ was allowed to vary randomly within a range of values determined from measured data and literature values.~~ The distribution of $\varepsilon$ values was defined based on measured data. If the initial $\delta^{15}N_{NO3}$ is known, $\varepsilon$ for $\delta^{15}N_{NO3}$ ($\varepsilon_{15N}$) can be determined from the slope of the linear regression line on a plot of $ln(f_d)$ vs. $\delta^{15}N_{NO3}$ (Böttcher et al., 1990). If the initial $\delta^{15}N_{NO3}$ and $f_d$ are not known, as is the case here, $\varepsilon_{15N}$ can be determined from the slope of the regression line on a plot of $ln$(NO$_3$-N) vs. $\delta^{15}N_{NO3}$, which will be the same as on a plot of $ln(f_d)$ vs. $\delta^{15}N_{NO3}$. In-situ variations in temperature and reaction rates may affect the enrichment factor (Kendall and Aravena, 2000) and this was accounted for by allowing for variation in $\varepsilon_{15N}$ within the Monte Carlo analysis. The enrichment factor for $\delta^{18}O_{NO3}$ ($\varepsilon_{18O}$) was calculated by multiplying the $\delta^{15}N_{NO3}$ by a linear coefficient of proportionality determined for each CFO from the slope of the denitrification trend on an isotope cross-plot (see Section 3.2).

For each realization, initial isotopic values ($\delta^{15}N_0$ and $\delta^{18}O_0$) were determined by Solver such that the difference between $f_d$ calculated from $\delta^{15}N_{NO3}$ and $\delta^{18}O_{NO3}$ was minimized (<1% difference). The ranges of $\delta^{15}N_0$ and $\delta^{18}O_0$ were limited based on measured data and literature values (see 3.2). This approach neglects the effect of mixing of groundwater with differing isotopic values, and is valid if the concentration of $NO_3^-$ in the source is much

greater than background concentrations such that the isotopic composition of $NO_3^-$ is dominated by the agriculturally derived end-member.

**2.4.2 Quantification of mixing and initial concentrations of $Cl^-$ and $NO_3$-N**

A binary mixing model that also accounts for decreasing $NO_3$-N concentrations in response to denitrification was used to quantify $NO_3^-$ attenuation by mixing and estimate the initial concentrations of $Cl^-$ and $NO_3$-N. The

measured concentration of $Cl^-$ was assumed to be a function of two end-member mixing, described by

$$Cl = f_m Cl_i + (1 - f_m)Cl_b, \tag{3}$$

where $Cl$ is the measured concentration of $Cl^-$ in the groundwater sample, $Cl_i$ is the concentration of $Cl^-$ at the initial point of entry of the agriculturally derived $NO_3^-$ to the groundwater system, $Cl_b$ is the concentration of $Cl^-$ in the background ambient groundwater, and $f_m$ is the fraction of water in the sample from the source of

agriculturally derived $Cl^-$ (and $NO_3^-$) remaining in the mixture.

The concentration of $NO_3$-N was also assumed to be a function of two end-member mixing but with an additional coefficient, $f_d$ (the fraction of $NO_3$-N remaining after denitrification), applied to account for denitrification. The measured $NO_3$-N concentration was thus described by

$$NO_3\text{-}N = f_d(f_m NO_3\text{-}N_i + (1 - f_m)NO_3\text{-}N_b), \tag{4}$$

where $NO_3$-N is the concentration of $NO_3$-N measured in the groundwater sample, $NO_3$-$N_i$ is the concentration of $NO_3$-N in the source of agriculturally derived $NO_3^-$ at the initial point of entry to the groundwater system, and $NO_3$-$N_b$ is the concentration of $NO_3$-N in the background ambient groundwater. This mixing calculation was only conducted on samples for which $NO_3^-$ dominated total-N ($NH_3$-N <10% of $NO_3$-N) so that nitrification of $NH_3$ could be neglected.

If $Cl_i$ is much greater than $Cl_b$ and $NO_3$-$N_i$ is much greater than $NO_3$-$N_b$, then $f_m$ is insensitive to background concentrations and these terms can be neglected (see ~~Section~~ 4.2 for further discussion of this assumption). In this case, Eqs. (3) and (4) reduce to

$$Cl = f_m Cl_i, \tag{5}$$

$$NO_3\text{-}N = f_d(f_m NO_3\text{-}N_i). \tag{6}$$

Solving Eq. (6) for $f_m$ and substituting into Eq. (5) yields

$$\frac{NO_3\text{-}N_i}{Cl_i} = \frac{1}{f_d}\frac{NO_3\text{-}N}{Cl}. \tag{7}$$

Thus, for each groundwater sample, the ratio of $NO_3$-N/$Cl^-$ at the initial point of entry of the agriculturally derived $NO_3^-$ to the groundwater system $\left(\frac{NO_3\text{-}N_i}{Cl_i}\right)$ can be simply calculated using measured concentrations, and $f_d$ estimated from $NO_3^-$ isotope data. This provides a relatively simple method to identify agriculturally derived $NO_3^-$

from different sources (e.g., EMS vs. manure piles) if they have different $NO_3$-N/$Cl^-$ ratios. Estimated $Cl_i$ and $NO_3$-$N_i$ are reported as the mid-range value with uncertainty described by the minimum and maximum values. These initial concentrations are at the water table for top-down inputs, or at the saturated point of contact between the EMS and the aquifer for leakage from the EMS. This analysis assumes that a sampled water parcel consists of

water with agriculturally derived $NO_3^-$ that entered the aquifer from one source at one point in time and space and has since mixed with natural ambient groundwater. Any $NO_3^-$ produced during nitrification after the anthropogenic source water enters the aquifer is implicitly included in $NO_3$-$N_i$. The error in $\frac{NO_3\text{-}N_i}{Cl_i^-}$ was assumed to be dominated by error in the estimated $f_d$, with the measurement error in $NO_3$-N and $Cl^-$ considered negligible.

The initial concentrations of the agriculturally derived $NO_3^-$ source ($NO_3$-$N_i$ and $Cl_i$) were estimated by simultaneously solving Eqs. (5) and (6) using Excel Solver (GRG nonlinear). The absolute minimum values of $NO_3$-$N_i$ and $Cl_i$ were defined by measured concentrations (e.g., if $Cl_i$=$Cl$, $f_m$=1). Maximum values of $NO_3$-$N_i$ and $Cl_i$ were defined based on measured concentrations of $NO_3$-N and $Cl^-$ in groundwater and manure filtrate ($NO_3$-N $\leq 150$ mg L$^{-1}$ and Cl $\leq 1300$ mg L$^{-1}$; see Section 3.2). These maximum values of $NO_3$-$N_i$ and $Cl_i$ correspond to the minimum $f_m$. The value of $f_d$ was assumed to be the mean $f_d$ estimated from $NO_3^-$ isotopes using Eq. (2), and $\frac{NO_3\text{-}N_i}{Cl_i}$ was required to be within one standard deviation of the estimate from Eq. (7).

The resulting estimates of $f_m$ are reported as the mid-range, with uncertainty described by the minimum and maximum values. Larger values of $f_m$ indicate less mixing (a shorter path for advection-dispersion) and suggest a source close to the well. Smaller values of $f_m$ indicate extensive mixing (a longer path for advection-dispersion) and suggest a source further away from the well. The relative contributions of mixing and denitrification to $NO_3^-$ attenuation at each site were evaluated by comparing $f_m$ and $f_d$ for each sample. This analysis was conducted using isotope values from the samples collected on 1 May 2013 at CFO1, which were combined with the $Cl^-$ and $NO_3$-N data from 6 June 2013. At CFO4, results from stable isotopes collected on 27 October 2014 were combined with $Cl^-$ and $NO_3$-N data collected on 7 October 2014.

## 3. Results

### 3.1 Site hydrogeology

### 3.1.1 CFO1

The geology at CFO1 consists of clay and clay-till interspersed with sand layers of varying thickness to the maximum depth of investigation (20 m BG, bedrock not encountered). Hydraulic conductivities ($K$) calculated from slug tests on wells ranged from $1.2\times10^{-7}$ to $4.2\times10^{-5}$ m-s$^{-1}$ (n=10) for sand, $1.1\times10^{-8}$ to $2.8\times10^{-8}$ m s$^{-1}$ (n=2) for clay-till, and $1.6\times10^{-9}$ to $3.0\times10^{-7}$ m s$^{-1}$ (n=8) for clay. Depth to the water table throughout the study site ranged from 0.5 m at DMW14 to 3.8 m at DMW11. Seasonal water table variations were about 0.5 m with no obvious change in the annual average during the 6-year measurement period. Water table elevation was highest at DMW10 and DMW1 on the west side of the site and lowest at DMW11 on the northeast side of the site (see Supplementary Material). Measured heads indicate groundwater flow from the vicinity of the EMS to the northeast and southeast. Mean horizontal hydraulic gradients at the water table ranged from $4.4\times10^{-3}$ to $1.4\times10^{-2}$ m m$^{-1}$. Vertical gradients were predominantly downward in the upper 20 m of the profile (mean gradients ranging from $1.8\times10^{-3}$ to 0.18 m m$^{-1}$), with the exception of DMW11 where the vertical gradient was upward (mean gradient -2.8×10$^{-2}$ m m$^{-1}$). Using the geometric mean $K$ for the sand (5.0 x 10$^{-6}$ m s$^{-1}$) and a lateral head gradient of $1.4\times10^{-2}$ m m$^{-1}$ yields a specific discharge (Darcy flux, $q$) of 2.2 m y$^{-1}$. Assuming an effective porosity of 0.3 (Rodvang et al., 1998), the average linear velocity ($\bar{v}$) is 7.4 m y$^{-1}$. This suggests that, in the absence of attenuation by mixing or

denitrification, agriculturally derived $NO_3^-$ could have been transported through the groundwater system by advection about 400 m from the EMS since 1960 and 630 m since 1930.

### 3.1.2 CFO4

The geology at CFO4 consists of about 5 m of clay (with minor till) underlain by sandstone, to the maximum depth investigated (20 m BG). Hydraulic conductivities measured using slug tests on wells were $1.0 \times 10^{-8}$ to $1.0 \times 10^{-5}$ m s$^{-1}$ (n=12) for the clay and sandstone (many shallow wells were screened across the clay-till and into the sandstone) and $1.0 \times 10^{-5}$ to $2.9 \times 10^{-5}$ m s$^{-1}$ (n=4) for the sandstone. The depth to water table ranged from 1.0 to 3.4 m, increasing from west to east across the study site. Seasonal water table variations were on the order of 1.5 m with water table declines on the order of 0.3 m y$^{-1}$. The horizontal hydraulic gradient was consistently from west to east, with a mean gradient at the water table of $3.9 \times 10^{-3}$ m m$^{-1}$ between BC2 and BMW2 and $4.3 \times 10^{-3}$ m m$^{-1}$ between BMW2 and BMW7. Vertical hydraulic gradients were $4.2 \times 10^{-2}$ to $4.6 \times 10^{-2}$ m m$^{-1}$ downward. Using the geometric mean $K$ for the site ($2.9 \times 10^{-5}$ m s$^{-1}$) and a lateral head gradient of $4.3 \times 10^{-3}$ m m$^{-1}$ yields a $q$ of 0.4 m y$^{-1}$. Assuming an effective porosity of 0.3 yields a $\bar{v}$ of 1.3 m y$^{-1}$. These values suggest that, in the absence of attenuation by mixing or denitrification, anthropogenic $NO_3^-$ could have been transported through the groundwater systems about 10 m by advection between 1995 and the time of sampling.

### 3.2 Values and evolution of stable isotopes of nitrate

~~Manure filtrate from the EMS at CFO1 had $\delta^{15}N_{NO3}$ ranging from 0.4 to 5.0‰ and $\delta^{18}O_{NO3}$ ranging from 7.1 to 19.0‰. A curve showing the co-evolution of $\delta^{18}O_{NO3}$ (mixing of atmospheric $\delta^{18}O$ with groundwater-derived $\delta^{18}O$) and $\delta^{15}N_{NO3}$ (Rayleigh distillation, $\beta = 1.005$) during nitrification is shown in Fig. 2. Isotopic values in DMW3, where direct leakage from the EMS was evident, are consistent with partial nitrification following this trend of isotopic evolution ($\delta^{18}O_{NO3}$ of -1.2‰ and $\delta^{15}N_{NO3}$ of 7.8‰).~~

The range of isotopic values of $NO_3^-$ in groundwater ~~is~~ was similar at both sites (Fig. 2). At CFO1, $\delta^{18}O_{NO3}$ ranged from -5.9 to 20.1‰ and $\delta^{15}N_{NO3}$ from -5.2 to 61.0‰. At CFO4, $\delta^{18}O_{NO3}$ ranged from -1.9 to 31.6‰ and $\delta^{15}N_{NO3}$ from -1.3 to 70.5‰. The isotopic values of $\delta^{18}O_{NO3}$ in groundwater are commonly assumed to be derived from a mix of a 1/3 atmospheric-derived oxygen (+23.5‰) and 2/3 water-derived oxygen (Xue et al., 2009). Given the average $\delta^{18}O_{H2O}$ for both sites (-16‰, see Supplementary Material), a 1/3 atmospheric 2/3 groundwater mix would result in a $\delta^{18}O_{NO3}$ of -3.7‰. Manure filtrate from the EMS at CFO1 had $\delta^{15}N_{NO3}$ ranging from 0.4 to 5.0‰ and $\delta^{18}O_{NO3}$ ranging from 7.1 to 19.0‰. A curve showing the co-evolution of $\delta^{18}O_{NO3}$ (mixing of atmospheric $\delta^{18}O$ with groundwater-derived $\delta^{18}O$) and $\delta^{15}N_{NO3}$ (Rayleigh distillation, $\beta = 1.005$) during nitrification is shown in Fig. 2. Isotopic values in DMW3, where direct leakage from the EMS was evident, are consistent with partial nitrification following this trend of isotopic evolution ($\delta^{18}O_{NO3}$ of -1.2‰ and $\delta^{15}N_{NO3}$ of 7.8‰).

At both sites, co-enrichment of $\delta^{18}O_{NO3}$ and $\delta^{15}N_{NO3}$ characteristic of denitrification was evident in some samples (slopes of 0.42 and 0.72 in Fig. 2a). At CFO1, this includes samples from DP10-2, DMW5, DMW11, DMW12, DP11-12b, and DMW13 (and associated core) and some pore water from cores DC15-22 and DC15-23. These samples had $NO_3$-N concentrations of 0.6 to 23.7 mg L$^{-1}$, $\delta^{18}O_{NO3}$ ranging from 4.8 to 20.6‰, and $\delta^{15}N_{NO3}$ ranging from 22.9 to 61.3‰. At CFO4, samples exhibiting evidence of denitrification were from BMW2, BMW5, BMW6, BMW7, and BC4. These samples had $NO_3$-N concentrations ranging from 0.4 to 35.1 mg L$^{-1}$, $\delta^{18}O_{NO3}$ ranging

from 1.6 to 22.1‰, and $\delta^{15}N_{NO3}$ ranging from 20.9 to 70.1‰. Although the isotopic values of DMW5 suggest enrichment by denitrification, the data plot away from the rest of the CFO1 data and close to the denitrification trend at CFO4 (Fig. 2), suggesting these samples were affected by some other process (possibly mixing or nitrification); therefore, the fraction of $NO_3$-N remaining in this well was not calculated. Also, well DMW3, which clearly receives leakage from the EMS, did not contain substantial $NO_3$-N and so $f_d$ was not calculated.

In the Monte Carlo analysis tThe potential range of original isotopic values of the $NO_3^-$ source prior to denitrification ($\delta^{15}N_0$ and $\delta^{18}O_0R_0$) varied from 5 to 27‰ for $\delta^{15}N_{NO3}$ and from -2 to 7‰ for $\delta^{18}O_{NO3}$ based on isotopic values measured during this study (Fig. 2a). These values are consistent with literature values for manure-sourced $NO_3^-$, which report $\delta^{15}N_{NO3}$ ranging from 5 to 25‰ and $\delta^{18}O_{NO3}$ ranging from -5 to 5‰ (Wassenaar, 1995; Wassenaar et al., 2006; Singleton et al., 2007; McCallum et al., 2008; Baily et al., 2011).

The enrichment factor of $\delta\varepsilon_{15N_{NO3}}$ was defined by a normal distribution with a mean of -10‰ and standard deviation of 2.5‰ (Fig. 2b). At CFO1, the coefficient of proportionality between the enrichment factor of $\delta^{15}N_{NO3}$ and $\delta^{18}O_{NO3}$ was described by a normal distribution with mean of 0.72 and standard deviation of 0.05. At CFO4, the coefficient of proportionality was also described by a normal distribution with a mean of 0.42 and standard deviation of 0.035 (see Fig. 2a). These enrichment factors are consistent with values from denitrification studies that report $\varepsilon_{15N}$ ranging from -4.0 to -30.0‰ and $\varepsilon_{18O}$ ranging from -1.9 to -8.9‰ (Vogel et al., 1981; Mariotti et al., 1988; Böttcher et al., 1990; Spalding and Parrott, 1994; Mengis et al., 1999; Pauwels et al., 2000; Otero et al., 2009).

### 3.3 Distribution and sources of agricultural nitrate in groundwater

At both sites TN concentrations in filtrate from the EMS and catch-basin were generally an order of magnitude larger than concentrations in groundwater (Table 2). The one exception is well DMW3 at CFO1 which intercepted direct leakage from the EMS (see 3.3.1 for further discussion of this well). The dominant form of N differed between manure filtrate and groundwater. In the EMS filtrate, N was predominately organic-N (TON up to 71%) or $NH_3$-N (up to 90%), with $NO_x$-N <0.1% of TN. In the catch-basin at CFO1 TON was >99% of TN. In groundwater TN concentrations ranged from <0.25 to 84.6 mg L$^{-1}$, and this N was predominantly $NO_3^-$ (again, with the exception of DMW3).

### 3.3.1 CFO1

Agriculturally derived $NO_3^-$ was predominantly generally restricted to the upper 20 m (or less) at CFO1 ($NO_3$-N ≤ 0.2 mg L$^{-1}$ and Cl$^-$ ≤ 57 mg L$^{-1}$ in seven wells screened at 20 m). The one exception was DP11-12b, which had up to 4.1 mg L$^{-1}$ of $NO_3$-N. The southeast portion of the site also does not appear to have been significantly contaminated by agriculturally derived $NO_3^-$, with $NO_3$-N concentrations < 1 mg L$^{-1}$ in five water table wells (DMW4, DMW6, DMW14, DMW15, DMW16). In DMW6, Cl$^-$ and TN concentrations were elevated (see Supplementary Material) but $NO_3$-N concentrations were < 2 mg L$^{-1}$. Collectively, these data suggest the catch basin is not a significant source of $NO_3^-$ to the groundwater at this site.

Leakage of manure slurry from the EMS at CFO1 is clearly indicated by the data from DMW3, which feature the highest concentrations of TN in groundwater (up to 548 mg L$^{-1}$) and elevated Cl$^-$, $HCO_3^-$, and DOC in concentrations similar to EMS manure filtrate (see Supplementary Material). Nevertheless, $NO_3$-N concentrations in this well were consistently low (1.1 ± 2.7 mg L$^{-1}$, n=22). The potential for nitrification in the vicinity of this

well is indicated by NO$_2$-N production (2.7 ± 8.3 mg L$^{-1}$, n=22). However, the data demonstrate that only a small proportion of the NH$_3$-N in DMW3 (373.4 ± 79.4 mg L$^{-1}$, n=22) could have been converted to NO$_3^-$ within the subsurface (NO$_3$-N in groundwater ≤ 66 mg L$^{-1}$) (NO$_3$-N/Cl$^-$ ratio of 0.95). Further work is required to assess the importance of cation exchange as an attenuation mechanism for direct leakage from the EMS at this site.

5 Contamination by agricultural NO$_3^-$ that exceeds the drinking water guidelines (NO$_3$-N > 10 mg L$^{-1}$) was observed in four wells (DMW1, DMW11, DMW13 and DP10-2) and in continuous core (DC15-23). DMW2 and DMW12 also had NO$_3$-N concentrations that were elevated but did not exceed the drinking water guideline (≤ 3.7 mg L$^{-1}$). Given the evidence of partial nitrification in DMW3 (and low NO$_3$-N concentrations), the NO$_3$-N/Cl$^-$ ratio of contamination from the EMS was assumed to be best represented by DP10-2, which is located directly 10 downgradient of the EMS. Data for this well indicate values of NO$_3$-N/Cl$^-$ predominantly ranging from 0.1 to 0.3 with $NO_3$-$N_i$/$Cl_i$ estimated at 0.3 ± 0.13 (Fig. 4).

The maximum NO$_3$-N concentration in groundwater at CFO1 (66.4 mg L$^{-1}$) was measured in core sample DC15-23 (clay at 2 m bgl, 7 m hydraulically downgradient of DMW3). The NO$_3$-N in this core sample was most likely 15 introduced into the groundwater system by vertical infiltration or diffusion from above. PPore water extracted from the unsaturated zone (sand) at the top of this core profile contained 865 mg L$^{-1}$ of NO$_3$-N and had a NO$_3$-N/Cl$^-$ ratio of 1.04, consistent with the ratio of 0.95 in the core sample. Given this consistency, and that The NO$_3$-N concentrations in the well immediately up-gradient were low (DMW3), the NO$_3$-N in this core sample was most likely introduced into the groundwater system by vertical infiltration or diffusion from above. In contrast, 20 Contamination by agricultural NO$_3^-$ that exceeds the drinking water guidelines (NO$_3$-N > 10 mg L$^{-1}$) was observed in wells DMW2 and DMW12 also had NO$_3$-N concentrations that were elevated but did not exceed the drinking water guideline (≤ 3.7 mg L$^{-1}$). up to 40 m hydraulically downgradient of the EMS (DMW13, DP10-2) and in well DMW11 situated 470 from the EMS (Fig. 3). DMW1, located upgradient of the EMS, also had concentrations of NO$_3$-N > 10 mg L$^{-1}$ with an increasing trend, but the source of this NO$_3^-$ is not clear. DMW2 and DMW12 also 25 had NO$_3$-N concentrations that were elevated but did not exceed the drinking water guideline (≤ 3.7 mg L$^{-1}$). Given the evidence of incomplete partial nitrification in DMW3, the NO$_3$-N/Cl$^-$ ratio of contamination from the EMS was assumed to be best represented by DP10-2, which is located directly downgradient of the EMS. Data for this well indicate values of NO$_3$-N/Cl$^-$ predominantly ranging from 0.1 to 0.3 with $NO_3$-$N_i$/$Cl_i$ estimated at 0.3 ± 0.13 (Fig. 4). Advective transport from DMW3 is also the likely source ofelevated NO$_3$-N (up to 21.1 mg L$^{-1}$) 30 within the sand between 6 and 12 m depth in this core had in DC15-23. NO$_3$-N/Cl$^-$ ratios consistent with an EMS source (0.07 to 0.31)in these samples ranged from 0.07 to 0.31, consistent with DP10-2. Stable isotope values in pore water from this sand layer do not indicate substantial denitrification (δ$^{18}$O ≤ 5.9‰, δ$^{15}$N ≤ 16.7‰), suggesting these ratios will be similar to the initial ratios at the point of entry to the groundwater system.

In DMW13 (33 m downgradient from DP10-2) tIn contrast, thehe ratio of $NO_3$-$N_i$/$Cl_i$ in DMW13 (33 m 35 downgradient from DP10-2) was 0.75 ± 0.29. is more similar to the NO$_3$-N/Cl$^-$ ratio in DC15-23 at 2 m (0.95), which is interpreted as reflecting a top-down source. The NO$_3^-$ in DMW13 is therefore unlikely to be sourced solely from leakage from the EMS, and could be sourced from the adjacent dairy pens or a temporary manure pile that was observed adjacent to this well during core collection in 2015 (or a combination of EMS and top-down sources).

~~The $NO_3\text{-}N_i/Cl_i$ ratio I~~in DMW12 ~~the~~ $NO_3\text{-}N_i/Cl_i$ ratio ~~is~~ was not inconsistent with an EMS source, but the hydraulic gradient between DMW2 and DMW12 is negligible, indicating a lack of driving force for advective transport from the EMS towards DMW12. This is also the case for well DMW1, which is up-gradient of the EMS but had elevated $NO_3$-N concentrations (6.5 ± 3.6, n=18). The source of nitrate in these wells is therefore unlikely to be related to leakage from the EMS, but alternative sources (i.e., nearby temporary manure piles) are not known. Well DMW11, 470 m from the EMS, had ~~had con~~consistently low $NO_3$-N/Cl⁻ ratios (< 0.05)~~. The $NO_3\text{-}N_i/Cl_i$ ratio indicated by DMW11 was~~ similar to DP10-2, but estimates of $Cl_i$ were three-fold higher than $Cl_i$ for DP10-2 (Fig. 4b). ~~, but estimates of $Cl_i$ indicate Cl⁻ sourced from inputs with three-fold higher Cl⁻ concentrations than the source to DP10-2 (Fig. 4b). $NO_3\text{-}N_i$ and $Cl_i$ estimated for DMW11 were consistent with measured values in that well, indicating a local top-down source.~~ Well DMW11 is located hydraulically downgradient of feedlot pens and adjacent to a solid manure storage area~~. Well DMW11 is also~~, in a local topographic low. ~~and~~ Elevated $NO_3$-N in this well is therefore interpreted to be ~~likely receiving $NO_3$-N and Cl⁻~~ from surface runoff and top-down infiltration, rather than lateral advection from the EMS ~~in addition to subsurface groundwater flow~~. ~~Well DMW11 had high $NO_3\text{-}N_i$ and $Cl_i$ consistent with measured values in that well, indicating a local top-down source that is likely the nearby solid manure pile.~~

### 3.3.2 CFO4

At CFO4, measured data indicate that effects from agricultural operations on $NO_3^-$ concentrations in groundwater are restricted to the upper 15 m of the subsurface. $NO_3$-N concentrations in wells screened at 15 m depth were < 0.5 mg L⁻¹, with the exception of one sample from BP10-15w (May 2012) with 4.3 mg L⁻¹ of $NO_3$-N. Water table wells in the west and north of the study site (BC1, BC2, and BC3) also indicate negligible impacts of agricultural operations, with Cl⁻ < 10 mg L⁻¹ and $NO_3$-N < 0.1 mg L⁻¹.

Concentrations of $NO_3$-N >–10 mg L⁻¹ were measured in three water table wells (BMW2, BMW3, BMW4) ~~installed~~ adjacent to the EMS, indicating that they have been impacted by the EMS (Fig. 5). Of these, BMW2 had much higher Cl⁻ concentrations (502 ± 97 mg L⁻¹, n=22 in BMW2 compared to 182 ± 81 mg L⁻¹ in BMW3 and 188 ± 74 mg L⁻¹ in BMW4), and therefore lower $NO_3$-N/Cl⁻ ratios (<0.05). ~~Given the elevated~~ Cl⁻ concentrations in ~~this well~~BMW2 were consistent with concentrations in the EMS suggesting ~~,~~ direct leakage~~,~~ while ~~s~~from the ~~EMS was assumed to be the source. St~~table isotopes of $NO_3^-$ and initial concentrations ($NO_3\text{-}N_i \geq 127$ mg L⁻¹) indicate substantial denitrification (Table 2, Fig. 6) ~~in BMW2, with estimated $NO_3\text{-}N_i \geq 127$ mg L⁻¹~~. The ~~and an~~ $NO_3\text{-}N_i/Cl_i$ ratio in BMW2 is consistent with ~~of~~ 0.1 to 0.3 measured $NO_3$-N/Cl⁻ ~~(Fig. 6)~~ in ~~. This ratio is consistent with data from well~~ BMW4, which ~~is immediately adjacent to the EMS (on the upgradient side) and~~ therefore likely reflects leakage from the EMS without denitrification (~~based on~~consistent with stable isotope~~s~~ of values of $NO_3^-$). ~~$NO_3$-N/Cl⁻ ratios measured in BMW4 were predominantly 0.1 to 0.3, consistent with the reconstructed $NO_3\text{-}N_i/Cl_i$ ratio in BMW2.~~

Given that the estimated subsurface travel distance during operations at this site is 10 m, ~~A~~agriculturally derived $NO_3^-$ in other wells not immediately adjacent to the EMS is unlikely to be related to leakage from the EMS. Wells BMW5 and BMW7 are 60 and 140 m hydraulically downgradient from the EMS, respectively. $NO_3\text{-}N_i/Cl_i$ ratios in these wells were not inconsistent with BMW2 (i.e., the range of values overlap), but given the distance from the EMS ~~but advective transport is only likely to have transported solutes around 10 m since the EMS was~~

installed (see Section 3.1.2). As such, the source of NO$_3$-N in these wells is most likely the adjacent dairy pens rather than the EMS. Concentrations of NO$_3$-N > 10 mg L$^{-1}$ were also measured in BC4, which is located 95 m hydraulically upgradient of the EMS. The ratio of $NO_3$-$N_i$/$Cl_i$ at BC4 was the highest at CFO4 (0.6) and did not overlap with BMW2. This indicates that tThe NO$_3^-$ in this well is interpreted to have been was sourced from an adjacent manure pile, which was observed during the study.

**3.4 Mechanisms of attenuation of agriculturally derived NO$_3^-$**

Attenuation of agriculturally derived NO$_3^-$ in groundwater is dominated by denitrification at both CFO1 and CFO4, with estimates of $f_m$ consistently higher than estimates of $f_d$ (Table 3, Fig. 7, Table S10). Calculated $f_d$ values indicate that where denitrification was identified, suggest that at least half of the NO$_3$-N present at the initial point of entry to the groundwater system has been removed by denitrificationthis attenuation mechanism. Comparison of $NO_3$-$N_{mix}$ (the concentration of NO$_3$-N that would be measured if mixing was the only attenuation mechanism) with measured concentrations (which reflect attenuation by both mixing and denitrification) suggests that the sample from 20 m depth (DP11-12b) is the only sample that would be below the drinking water guideline if mixing was the only attenuation mechanism (Fig. 8).

At both sites, the stable isotope values of NO$_3^-$ indicate that denitrification proceeds within metres of the source. At CFO1, calculated $f_d$ in well DP10-2 (2 m from the EMS) is 0.52 ± 0.22; at CFO4, $f_d$ in well BMW2 (3 m from the EMS) is 0.13 ± 0.06. Denitrification also substantially attenuated NO$_3$-N concentrations in wells where the source is not the EMS but instead is adjacent solid manure piles (e.g., DMW11 at CFO1, BC4 at CFO4). In BMW6 at CFO4, denitrification completely attenuated the agriculturally derived NO$_3^-$. This well had negligible NO$_3$-N (0.4 ± 0.2 mg L$^{-1}$, n=8) and the lowest $f_d$ of 0.01. Measured DOC in this well was consistent with other wells at both sites (6.9 ± 1.7 mg L$^{-1}$, n=3), suggesting DOC depletion does not limit denitrification at these CFO operations. Calculated $f_d$ and $f_m$ should decrease with increasing subsurface residence time and distance from source. Data from wells support the source identification based on concentrations of NO$_3$-N and Cl$^-$ and NO$_3$-N/Cl$^-$ ratios (see Section 3.3). Well DMW11 (470 m from the EMS) had the highest $f_m$ at CFO1 (0.83), indicating less mixing and suggesting the anthropogenic source of NO$_3^-$ in this well is relatively close, which is consistent with the adjacent the solid manure pile being the source of NO$_3^-$ to this well. At CFO4, well BMW2, which is adjacent to the EMS, had the highest $f_m$ (0.92), indicating the least attenuation of NO$_3^-$ by mixing and consistent with the EMS being the source of NO$_3^-$ to this well.

**4. Discussion**

4.1 Implications for on-farm waste management

Agriculturally derived NO$_3^-$ at these two sites with varying lithology is was generally restricted to depths < 20 m, consistent with previous studies at CFOs (Robertson et al., 1996; Rodvang and Simpkins, 2001; Rodvang et al., 2004; Kohn et al., 2016). Attenuation of agriculturally derived NO$_3^-$ in groundwater is was a spatially varying combination of mixing and denitrification, with denitrification playing a greater role than mixing at both sites. In the samples for which $f_d$ could be determined, denitrification reduced NO$_3^-$ concentrations by at least half and, in some cases, back to background concentrations. Given that the range of source isotopic composition was allowed

to vary to its maximum justifiable extent, these quantitative estimates of denitrification based on stable isotopes of $NO_3^-$ are likely to be conservative. Redox conditions within the groundwater system were not able to be determined in this study due to the sampling method used to collect groundwater from wells screened across low-K formations (well bailed dry then sample collected after water level recovery). However, ~~D~~denitrification

appears to proceed within metres of the $NO_3^-$ source, suggesting relatively short sub-surface residence times are required and that redox conditions ~~at the~~close to the water table ~~may be~~are conducive to denitrification reactions (Critchley et al., 2014; Clague et al., 2015).

The substantial role of denitrification within the saturated glacial sediments at these study sites indicates the potential for significant attenuation of agriculturally derived $NO_3^-$ by denitrification in similar groundwater

systems across the North American interior and Europe (Ernstsen et al., 2015; Zirkle et al., 2016). Denitrification in the unsaturated zone is limited by low water contents and oxic conditions, resulting in substantial stores of $NO_3^-$ in vadose zones (Turkeltaub et al., 2016; Ascott et al., 2017). $NO_3^-$ in water that is removed rapidly from site is also unlikely to be substantially attenuated by denitrification due to oxic conditions and rapid transit times (Ernstsen et al., 2015). Therefore, water management focussed on reducing the effects of $NO_3^-$ contamination in

similar hydrogeological settings to this study should aim to maximize infiltration into the saturated zone where $NO_3^-$ concentrations can be naturally attenuated, provided that local groundwater isn't used for potable water supply.

At both sites there is evidence of elevated $NO_3^-$ due to leakage from the EMS, but the impact appears to be limited to within metres of the EMS. This suggests that saturation within the clay lining of the EMS has limited the

20 development of extensive secondary porosity that would allow rapid water percolation (Baram et al., 2012). Infiltration of $NO_3^-$ rich water that has passed through temporary solid manure piles and dairy pens has resulted in groundwater $NO_3$-N concentrations as high as those associated with leakage from the EMS (e.g., DMW11, ~~DMW13,~~ BC4). At CFO4, this is in spite of the presence of clay at surface, ~~which is attributable to~~reflecting secondary porosity in the upper part of the profile that has led to hydraulic conductivities comparable to sand.

This is consistent with the findings of Showers et al. (2008), who investigated sources of $NO_3^-$ at an urbanized dairy farm in North Carolina, USA. Construction of EMS facilities in Alberta has been regulated under the Agriculture Operation Practices Act since 2002, which requires them to be lined with clay to minimise leakage (Lorenz et al., 2014). ~~The results of this study suggest that o~~On-farm waste management should increasingly focus on minimising temporary manure piles that are in direct contact with the soil to reduce $NO_3^-$ contamination

associated with dairy farms and feedlots.

~~The absence of direct leakage from the EMS at CFO4 suggests that saturation within the clay lining of the EMS has limited the development of extensive secondary porosity that would allow rapid water percolation (Baram et al., 2012). Elevated $NH_3$-N concentrations in the water table well at the southeast corner of the EMS at CFO1 (DMW3) do indicate direct leakage from the EMS, but because nitrification within the EMS is minimal, this has~~

35 ~~not resulted in elevated $NO_3$-N in this well. Two possibilities for the fate of $NH_3$-N in DMW3 are attenuation by cation exchange and oxidation to $NO_3$-N within the groundwater system. Measured $NO_3$-N concentrations in groundwater represent only a small fraction ($\leq 10\%$) of $NH_3$-N within the EMS (or DMW3), suggesting oxidation to $NO_3^-$ within the aquifer may be limited. Further work is required to assess the importance of cation exchange as an attenuation mechanism for direct leakage from the EMS at this site.~~

## 4.2 Critique of this approach and applicability at other sites

~~The sources of manure-derived NO₃⁻ (manure piles vs. EMS) are distinguishable based on $NO_3$-$N_i$/$Cl_i$ ratios, provided there is also an understanding of the history of each site, local hydrogeology, and potential sources.~~ At both sites, leakage from the EMS had $NO_3$-$N_i$/$Cl_i$ of between 0.1 and 0.4, but this alone was not diagnostic of the source. The sources of manure-derived NO₃⁻ (manure piles vs. EMS) are distinguishable based on $NO_3$-$N_i$/$Cl_i$ ratios, provided there is also an understanding of the history of each site, local hydrogeology, and potential sources. Calculated $f_d$ and $f_m$ generally decreased with increasing subsurface residence time and distance from source, providing additional evidence for source attribution. For example, at CFO4, well BMW2, which is adjacent to the EMS, had the highest $f_m$ (0.92), indicating the least attenuation of $NO_3$ by mixing and consistent with the EMS being the source of NO₃⁻ to this well.

~~Estimation of~~Calculation of $NO_3$-$N_i$/$Cl_i$ ~~assumes~~assumed that background concentrations could be neglected in the mixing ~~calculation~~model. ~~The error associated with this assumption increases as source concentrations and measured concentrations approach background concentrations.~~ At these study sites, background concentrations are likely to be < 20 mg L⁻¹ for Cl⁻ and < 1 mg L⁻¹ for NO₃-N. ~~Based on these values, e~~Estimated $NO_3$-$N_i$ values ~~we~~are at least 20 times background NO₃-N concentrations, and over 100 times background concentrations in some wells. The estimated $Cl_i$ values ~~are~~were at least three times background concentrations at CFO1 and at least 10 times background concentrations at CFO4. ~~In this study we applied a two-end member mixing model and assumed that background concentrations can be neglected.~~ The error introduced by neglecting background concentrations was assessed by comparing $f_m$ calculated with and without background concentrations included, using the full range of values in this study (Fig. 9). Neglecting background concentrations results in overestimation of $f_m$ (i.e. underestimation of the amount of attenuation mixing) with the largest errors when measured concentrations are close to background concentrations. For Cl⁻ the maximum difference of 0.13 is in the mid-range of $f_m$ values. For NO₃-N, the difference is consistently < 0.1 with the largest errors at the lowest values of $f_m$. The uncertainty in $f_m$ is primarily related to uncertainty in the initial concentrations ($Cl_i$ and $NO_3$-$N_i$), which depends on measured Cl⁻ and NO₃-N. The largest uncertainties in $NO_3$-$N_i$ and $Cl_i$ correspond to the lowest measured concentrations (i.e., furthest from the upper limit), with less uncertainty at higher measured concentrations as they approach the maximum values. Temporal variability in $NO_3$-$N_i$/$Cl_i$ for each source could not be determined based on the snapshot isotope sampling conducted, but this could be investigated by measuring NO₃⁻ isotopes in conjunction with NO₃-N and Cl⁻ at multiple times.

Although applicable at these sites, this approach may not be valid at other sites if additional sources of $NO_3$ in groundwater (e.g. fertilizer or nitrification) are significant, or if $NO_3$ concentrations in groundwater are naturally elevated (Hendry et al., 1984). The combination of the approach outlined here with measurement of groundwater age indicators would allow for better constraints on groundwater flow velocities and determination of denitrification rates (Böhlke and Denver, 1995; Katz et al., 2004; McMahon et al., 2004; Clague et al., 2015).

## 4.3 Comparison with isotopic values of NO₃⁻ in previous studies

Nitrate isotope values in groundwater at the two CFOs studied ~~are~~were generally consistent with previous studies reporting denitrification of manure-derived NO₃⁻ at dairy farms (Wassenaar, 1995; Wassenaar et al., 2006; Singleton et al., 2007; McCallum et al., 2008; Baily et al., 2011). However, ~~T~~the isotopic values of NO₃⁻ in the manure filtrate from the EMS at CFO1, were ~~generally i~~not ~~i~~nconsistent with values for manure-sourced NO₃⁻

reported in other groundwater studies (Wassenaar, 1995; Wassenaar et al., 2006; Singleton et al., 2007; McCallum et al., 2008a; Baily et al., 2011). This is likely to be because nitrification within the EMS was negligible (NO$_3$-N <0.7 mg L$^{-1}$), such that the isotopic values of NO$_3$-N in the manure filtrate reflect volatilization of NH$_3$ and partial nitrification within the EMS. $\delta^{18}O_{NO3}$ values may also have been affected by evaporative enrichment of the $\delta^{18}O_{H2O}$ being incorporated into NO$_3^-$ (Showers et al., 2008).

However, aA number of groundwater samples collected for during the presentthis study had relatively enriched $\delta^{18}O_{NO3}$ (> 15 ‰) with depleted $\delta^{15}N_{NO3}$ (< 15‰). Some of these isotopic values are within the range previously reported for NO$_3^-$ derived from inorganic fertilizer ($\delta^{15}N_{NO3}$ from -3 to 3‰ and $\delta^{18}O_{NO3}$ from -5 to 25‰), with the $\delta^{18}O_{NO3}$ depending on whether the NO$_3^-$ is from NH$_4^+$ or NO$_3^-$ in the fertilizer (Mengis et al., 2001; Wassenaar et al., 2006; Xue et al., 2009). To the best of our knowledge, however, no inorganic fertilizers have been applied at these study sites. Another potential source is NO$_3^-$ derived from soil organic N, but this should have $\delta^{15}N_{NO3}$ values of 0 to 10‰ and $\delta^{18}O_{NO3}$ values of -10 to 15‰ (Durka et al., 1994; Mayer et al., 2001; Mengis et al., 2001; Xue et al., 2009; Baily et al., 2011). Incomplete nitrification of NH$_4^+$ can result in $\delta^{15}N_{NO3}$ lower than the manure source (Choi et al., 2003), but as there was no measurable NH$_3$-N in these samples this is also unlikely. These isotope values may reflect the influence of NO$_3^-$ from precipitation, which usually has values ranging from -5 to 5‰ for $\delta^{15}N_{NO3}$ and 40 to 60‰ for $\delta^{18}O_{NO3}$, and has been reported to dominate NO$_3^-$ isotope values of groundwater under forested landscapes (Durka et al., 1994). Alternatively, they may be affected by microbial immobilization and subsequent mineralization and nitrification, which can mask the source $\delta^{18}O_{NO3}$ in aquifers with long residence times (Mengis et al., 2001; Rivett et al., 2008).

The isotopic values of NO$_3^-$ in the manure filtrate from the EMS at CFO1, were generally inconsistent with values for manure-sourced NO$_3^-$ reported in other groundwater studies (Wassenaar, 1995; Wassenaar et al., 2006; Singleton et al., 2007; McCallum et al., 2008a; Baily et al., 2011). This is likely to be because nitrification within the EMS was negligible (NO$_3$-N <0.7 mg L$^{-1}$), such that the isotopic values of NO$_3$-N in the manure filtrate reflect volatilization of NH$_3$ and partial nitrification within the EMS. $\delta^{18}O_{NO3}$ values may also have been affected by evaporative enrichment of the $\delta^{18}O_{H2O}$ being incorporated into NO$_3^-$ (Showers et al., 2008).

## 5. Conclusions

A mixing model constrained by quantitative estimates of denitrification from isotopes substantially improved our understanding of nitrate contamination at these sites. This novel approach has the potential to be widely applied as a tool for monitoring and assessment of groundwater in complex agricultural settings. NO$_3$-N concentrations in excess of the drinking water guideline were measured at both sites, with sources including manure piles, pens and the EMS. Even though these sites are dominated by clay-rich glacial sediments, the input of NO$_3^-$ to groundwater from temporary manure piles and pens resulted in comparable (or greater) NO$_3$-N concentrations than leakage from the EMS. This is attributed to the development of secondary porosity within unsaturated clays. On-farm management of manure waste should increasingly focus on limiting manure piles that are in direct contact with the soil to limit NO$_3^-$ contamination of groundwater. Nitrate attenuation at both sites is dominated by denitrification, which is evident even in wells directly adjacent to the NO$_3^-$ source. In the wells for which denitrification was identified, concentrations ofOn-site denitrification agriculturally-derived reduced agriculturally derived NNO$_3^-$ concentrations had been reduced by at least half and, in some wells, completely. In

the absence of denitrification all but one of these wells would have had NO$_3$-N concentrations above the drinking water guideline.

These results indicate that infiltration to groundwater systems in glacial sediments where NO$_3^-$ can be naturally attenuated is likely to be preferable to off-farm export via runoff or drainage networks, provided that local groundwater isn't a potable water source. On-farm management of manure waste at similar operations should increasingly focus on limiting manure piles that are in direct contact with the soil to limit NO$_3^-$ contamination of groundwater.

*Acknowledgements*

This research was supported by Alberta Agriculture and Forestry (AAF) and the Natural Resources Conservation Board (NRCB), who provided assistance with field work and laboratory analysis. Funding was also provided by a Natural Sciences and Engineering Research Council of Canada (NSERC) Industrial Research Chair (IRC) (184573) awarded to MJH. The authors thank Barry Olson at AAF for reviewing the manuscript. Our thanks also to the local producers, whose cooperation made this research possible.

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

**Table 1. Details of groundwater monitoring wells and continuous core collection at CFO1 and CFO4 (all screens installed at bottom of the well).**

| Site | Well/Core hole ID | Type[†] | Lateral distance from EMS* (m) | Ground elevation (m asl) | Total depth (m below ground) | Screen length (m) | Lithology of screened interval | $K$ (m s$^{-1}$) |
|---|---|---|---|---|---|---|---|---|
| CFO1 | DMW1 | WTW | 60 | 869.7 | 5.0 | 4.0 | Sand | |
| | DMW2 | WTW | 10 | 867.2 | 6.0 | 4.0 | Sand | $1.2 \times 10^{-7}$ |
| | DMW3 | WTW | 2 | 867.5 | 3.7 | 2.0 | Sand | |
| | DMW4 | WTW | 160 | | 4.2 | 4 | Sand | $1.3 \times 10^{-6}$ |
| | DMW5 | WTW | 270 | 866.4 | 6.8 | 4.0 | Clayey sand | $1.7 \times 10^{-5}$ |
| | DMW6 | WTW | 310 | | 6.7 | 4 | | |
| | DP10-1 | Piezo | 2 | 867.8 | 18.6 | 0.5 | Clay | $1.6 \times 10^{-9}$ |
| | DP10-2 | Piezo | 2 | 867.9 | 8.0 | 1.5 | Sand | $3.6 \times 10^{-5}$ |
| | DMW10 | WTW | 340 | 868.0 | 7.2 | 3.0 | Clay | $3.0 \times 10^{-7}$ |
| | DP11-10b | Piezo | 340 | 868.0 | 20 | 0.5 | Clay | $2.2 \times 10^{-8}$ |
| | DMW11 | WTW | 470 | 864.8 | 7.0 | 3.0 | Sand and clay | $4.2 \times 10^{-5}$ |
| | DP11-11b | Piezo | 470 | | 20 | 0.5 | Clay | $6.3 \times 10^{-9}$ |
| | DMW12 | WTW | 50 | 867.6 | 7.0 | 3.0 | Sand and clay | $7.4 \times 10^{-6}$ |
| | DP11-12b | Piezo | 50 | 867.6 | 20.1 | 1.0 | Clay | $1.1 \times 10^{-8}$ |
| | DMW13 | WTW | 35 | 867.1 | 7.0 | 3.0 | Sand | $8.9 \times 10^{-6}$ |
| | DP11-13b | Piezo + core | 35 | 867.1 | 20.0 | 0.5 | Clay | |
| | DMW14 | WTW | 105 | 865.7 | 7.0 | 3.0 | Clay | $5.7 \times 10^{-6}$ |
| | DP11-14b | Piezo | 105 | 865.7 | 20.0 | 0.5 | Sand | $1.1 \times 10^{-6}$ |
| | DMW15 | WTW | 185 | | 7.0 | 3 | Clay | $2.4 \times 10^{-8}$ |
| | DP11-15b | Piezo | 185 | | 20.0 | 0.5 | Clay | $1.4 \times 10^{-7}$ |
| | DMW16 | WTW | 320 | 866.0 | 6.0 | 3.0 | Sand and clay | - |
| | DP11-16b | Piezo | 320 | | 20.0 | 0.5 | Clay | $3.2 \times 10^{-9}$ |
| | DC15-20 | Core | 76 | | 15 | | | |
| | DC15-21 | Core | 45 | | 10.5 | | | |
| | DC15-22 | Core | 22 | | 12 | | | |
| | DC15-23 | Core | 9 | | 15 | | | |
| CFO4 | BC1 | WTW | 110 | 857.0 | 6.9 | 3.1 | Clay and sandstone | |
| | BC2 | WTW | 365 | 859.4 | 7.0 | 3.1 | Clay and sandstone | $2.2 \times 10^{-7}$ |
| | BC3 | WTW | 145 | 858.6 | 6.8 | 3.1 | Clay and sandstone | $1.3 \times 10^{-6}$ |
| | BC4 | WTW | 95 | 858.8 | 5.9 | 3.0 | Clay and sandstone | $3.4 \times 10^{-6}$ |
| | BC5 | WTW | 105 | 859.5 | 7.5 | 4.5 | Clay and sandstone | |
| | BMW1 | WTW | 4 | 858.6 | 7.1 | 3.1 | Clay and sandstone | $4.3 \times 10^{-6}$ |
| | BMW2 | WTW | 3 | 857.9 | 7.5 | 4.5 | Clay and sandstone | $8.5 \times 10^{-7}$ |
| | BMW3 | WTW | 8 | 858.6 | 6.0 | 3.0 | Clay and sandstone | |
| | BMW4 | WTW | 14 | 858.0 | 7.5 | 4.8 | Clay and sandstone | $1.0 \times 10^{-5}$ |
| | BMW5 | WTW | 60 | 858.0 | 7.5 | 4.5 | Clay and sandstone | |
| | BP5-15 | Piezo | 60 | 858.1 | 15.3 | 1.5 | Sandstone | $1.0 \times 10^{-7}$ |
| | BMW6 | WTW | 150 | 856.9 | 7.5 | 4.5 | Clay and sandstone | $4.0 \times 10^{-6}$ |
| | BP6-15 | Piezo | 150 | 856.8 | 15.2 | 1.5 | Sandstone | $3.0 \times 10^{-6}$ |
| | BMW7 | WTW | 140 | 856.7 | 7.5 | 4.5 | Clay and sandstone | $1.0 \times 10^{-6}$ |
| | BP10-15e | Piezo | 4 | 858.2 | 14.9 | 1.5 | Sandstone | $2.9 \times 10^{-5}$ |
| | BP10-15w | Piezo | 10 | 858.0 | 15.0 | 1.5 | Sandstone | $1.0 \times 10^{-5}$ |

*EMS=Earthen manure storage

[†]WTW=water table well, Piezo = piezometer, Core = continuous core

**Table 2. Range of measured concentrations of TN, NH$_3$-N, NO$_x$-N (NO$_2$-N + NO$_3$-N) and TON at each study site. At CFO1 results from monitoring well DMW3 are presented separately because values in this well differed substantially from all other wells.**

| Site | N-pool | TN (mg L$^{-1}$) | NH$_3$-N (mg L$^{-1}$) | NO$_x$-N (mg L$^{-1}$) | TON (mg L$^{-1}$) |
|---|---|---|---|---|---|
| CFO1 | EMS | 550 – 1820 | 275 – 747 | <0.1 – 0.4 | 73 – 1301 |
| | Catch-basin | 200 – 1440 | 2.5 – 7.3 | <0.1 | 196 – 1437 |
| | DMW3 | 278 – 548 | 219 – 479 | <0.1 – 50[*] | 31.3 – 73.9 |
| | Other monitoring wells | <0.25 – 33.4 | <0.05 – 2.9 | <0.1 – 31.4[**] | <0.2 –3.7 |
| CF04 | EMS[^] | 1000 – 1240 | 724 – 747 | 0.25 - 0.29 | 275 –492 |
| | Monitoring wells | <0.25 – 84.6 | <0.05 – 0.23 | <0.1 – 80.4 | <0.2 –13.9 |

[*] NO$_x$-N of 50 mg L$^{-1}$ in DMW3 consisted of 12.6 mg L$^{-1}$ as NO$_3$-N and 37.4 mg L$^{-1}$ as NO$_2$-N.
[**] NO$_x$-N max in groundwater measured in core (NO$_3$-N = 66.4 mg L$^{-1}$, NO$_x$-N = 67.8 mg L$^{-1}$)
[^] Range across three replicates measured on 25 August 2011

**Table 3. Calculated $f_d$ and $f_m$ based on measured Cl$^-$ and NO$_3$-N concentrations and stable isotope values of NO$_3^-$.**

| Study area | Sample ID* | Cl$^-$ (mg L$^{-1}$) | NO$_3$-N (mg L$^{-1}$) | $\delta^{15}N_{NO3}$ (‰) | $\delta^{18}O_{NO3}$ (‰) | $f_d$ (mean ± stdev) | $f_m$[**] (mid-range) |
|---|---|---|---|---|---|---|---|
| CFO1 | DP11-13_4.3m | 28.5 | 7.0 | 30.3 | 9.8 | 0.30 ± 0.15 | 0.58 |
| | DP11-13_5.2m | 25.0 | 7.8 | 31.0 | 10.8 | 0.34 ± 0.13 | 0.58 |
| | DP11-13_7m | 72.3 | 12.0 | 31.6 | 10.2 | 0.27 ± 0.13 | 0.65 |
| | DP11-13 _7.9m | 70.8 | 9.1 | 36.4 | 14.0 | 0.17 ± 0.09 | 0.68 |
| | DP11-13_8.8m | 81.7 | 10.9 | 29.6 | 9.9 | 0.32 ± 0.15 | 0.63 |
| | DC15-22_10m | 73.0 | 11.0 | 26.1 | 7.4 | 0.47 ± 0.21 | 0.63 |
| | DP10-2 | 74.5 | 11.8 | 24.2 | 4.8 | 0.52 ± 0.22 | 0.63 |
| | DMW11 | 436.1 | 17.1 | 33.3 | 10.9 | 0.17 ± 0.07 | 0.83 |
| | DMW12 | 78.0 | 2.57 | 29.8 | 14.3 | 0.23 ± 0.10 | 0.54 |
| | DMW13 | 56.7 | 23.7 | 23.0 | 6.8 | 0.56 ± 0.22 | 0.65 |
| | DP11-12b | 95.7 | 0.6 | 35.9 | 17.0 | 0.15 ± 0.08 | 0.54 |
| CFO4 | BC4 | 163.1 | 35.1 | 30.6 | 1.6 | 0.37 ± 0.13 | 0.82 |
| | BMW2 | 595.6 | 16.5 | 41.6 | 8.3 | 0.13 ± 0.06 | 0.92 |
| | BMW5 | 131.2 | 12.9 | 28.9 | 6.5 | 0.34 ± 0.16 | 0.63 |
| | BMW6 | 156.0 | 0.4 | 70.5 | 22.1 | 0.01 ± 0.01 | 0.56 |
| | BMW7 | 134.7 | 11.6 | 34.0 | 5.9 | 0.21 ± 0.11 | 0.68 |

*central depth of core samples, x, indicated as SampleID_xm.
** maximum $f_m$ is 1 for all samples, which implies no mixing.

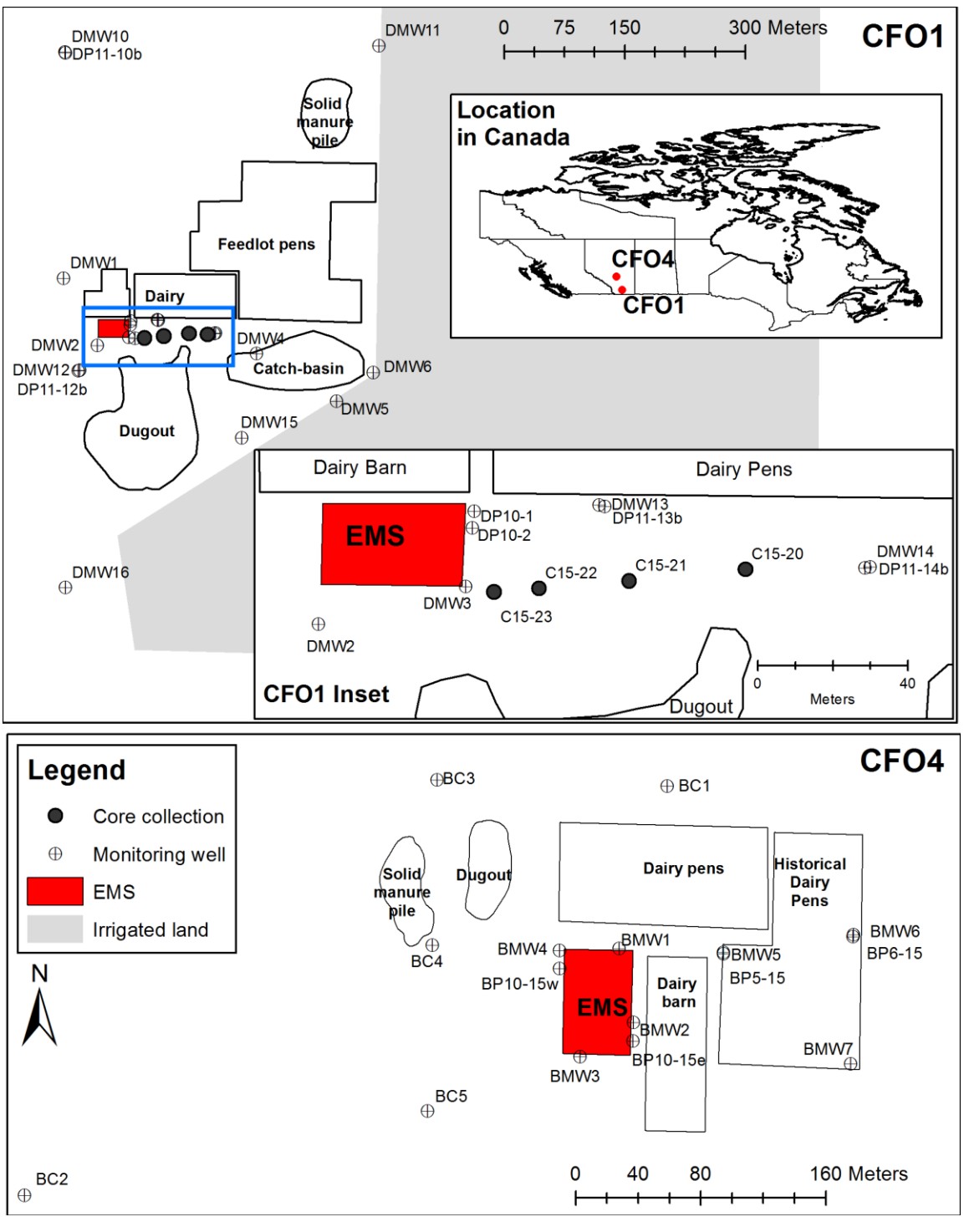

**Figure 1: Map of study sites CFO1 and CFO4, showing locations of groundwater monitoring wells, core collection, earthen manure storages (EMS), dairy and feedlot pens, manure piles, and irrigated land. Blue rectangle indicates extent of CFO1 inset.**

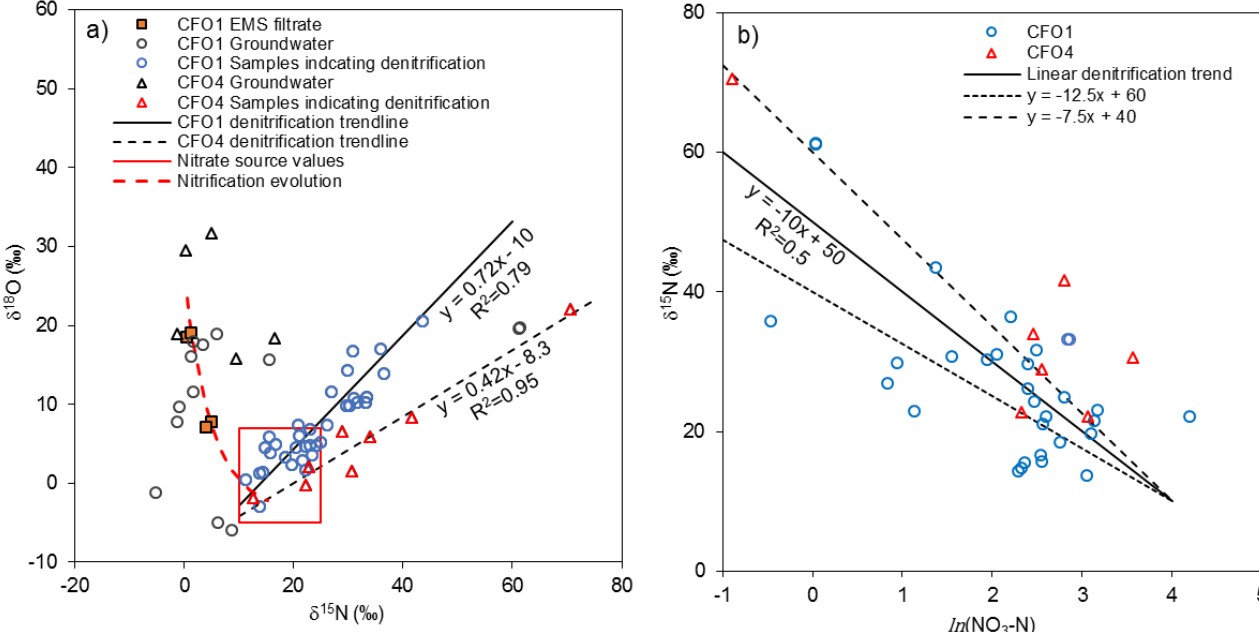

**Figure 2 (a)** Cross-plot of stable isotopes of nitrate at CFO1 and CFO4 showing hypothetical nitrification trend, boundary of manure-sourced NO$_3^-$ values and linear enrichment trends associated with denitrification, **(b)** enrichment of δ$^{15}$N$_{NO3}$ during denitrification (only samples within source region and with evidence of denitrification are shown) dashed lines represent ±1 std. dev. of enrichment factor (ε = -10) estimated from measured data.

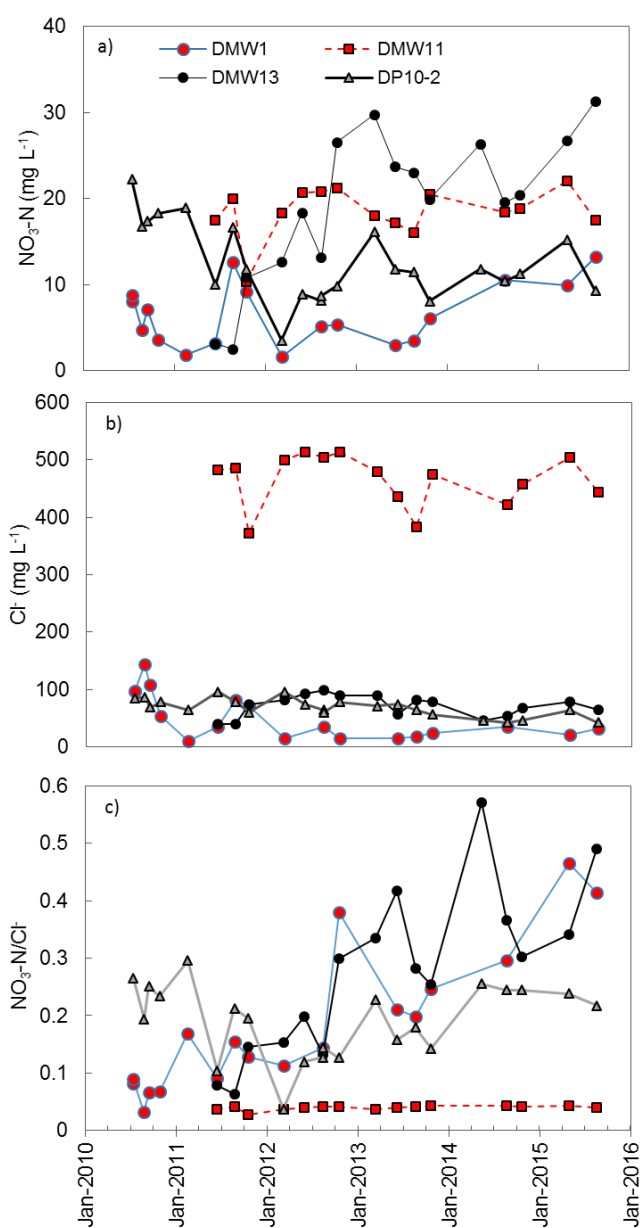

**Figure 3 Temporal variations in (a) NO₃-N, (b) Cl⁻, and (c) NO₃-N/Cl⁻ at CFO1. Only wells with NO₃-N > 10 mg L⁻¹ are shown.**

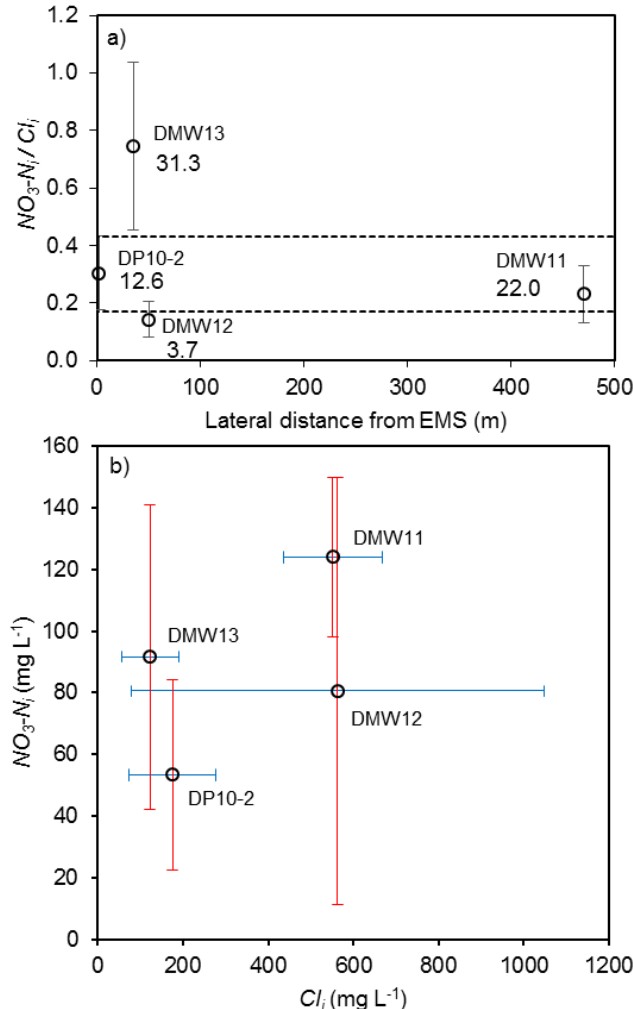

**Figure 4 (a) Estimated $NO_3$-$N_i$/$Cl_i$ ratios (mean and st. dev.) in water table wells with evidence of denitrification at CFO1, plotted with distance from earthen manure storage (EMS), where dashed lines are the upper and lower bounds of DP10-2 (EMS source) and values are maximum measured $NO_3$-N (mg L$^{-1}$). (b) Estimated concentrations of $NO_3$-$N_i$ and $Cl_i$ at CFO1 (mid-range, error bars are max. and min. values).**

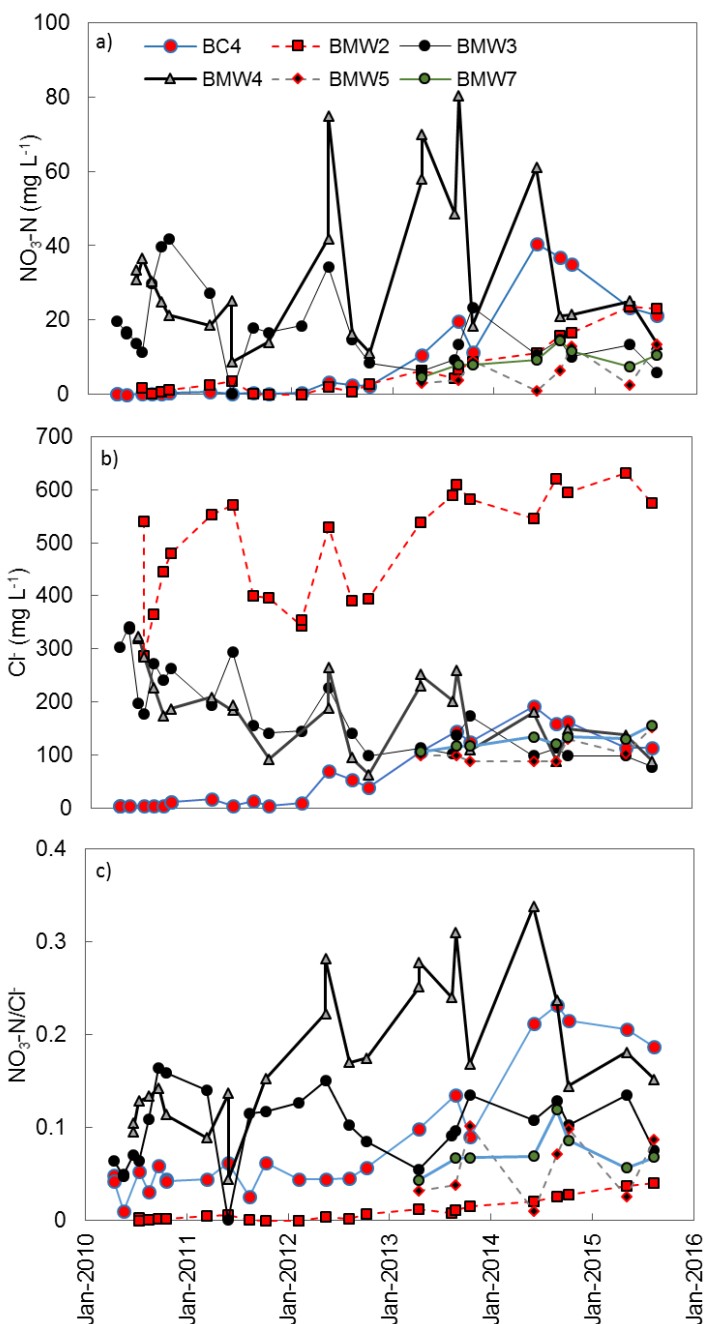

**Figure 5 Temporal variations in (a) NO₃-N, (b) Cl⁻, and (c) NO₃-N/Cl⁻ at CFO4. Only wells with NO₃-N > 10 mg L⁻¹ are shown.**

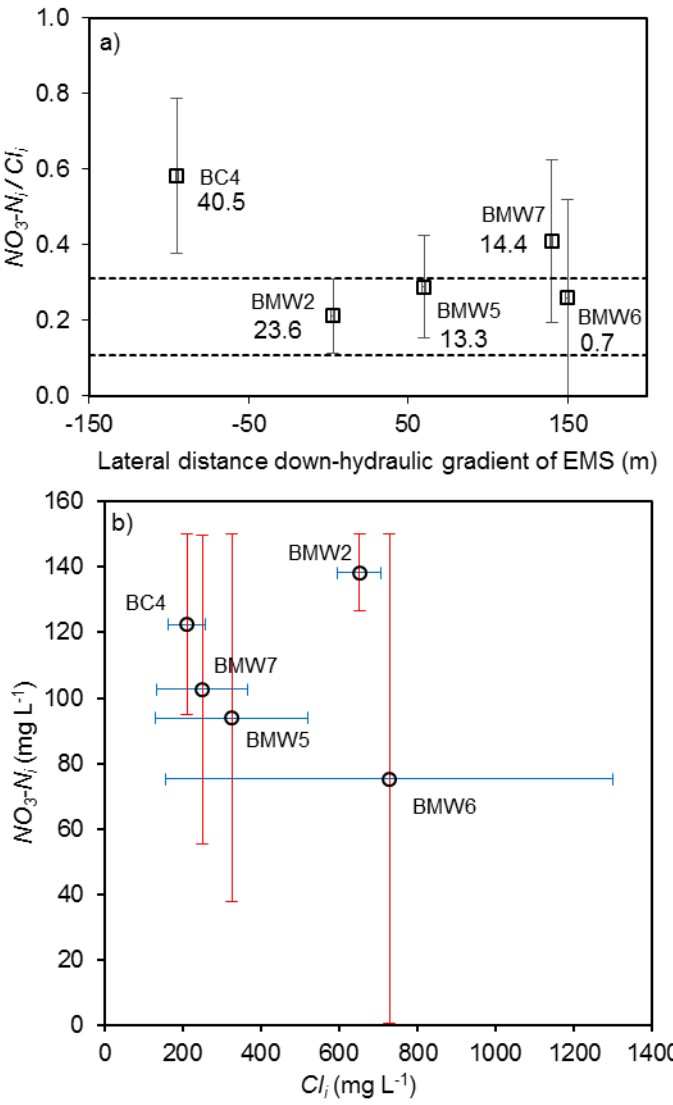

**Figure 6 (a) Estimated *NO₃-Nᵢ/Clᵢ* ratios (mean and st. dev.) in water table wells with evidence of denitrification at CFO4, plotted with distance from earthen manure storage (EMS), where dashed lines are upper and lower bounds of BMW2 (EMS source) and values are maximum measured NO₃-N (mg L⁻¹). (b) Estimated concentrations of *NO₃-Nᵢ* and *Clᵢ* at CFO1 (mid-range, error bars are max. and min. values).**

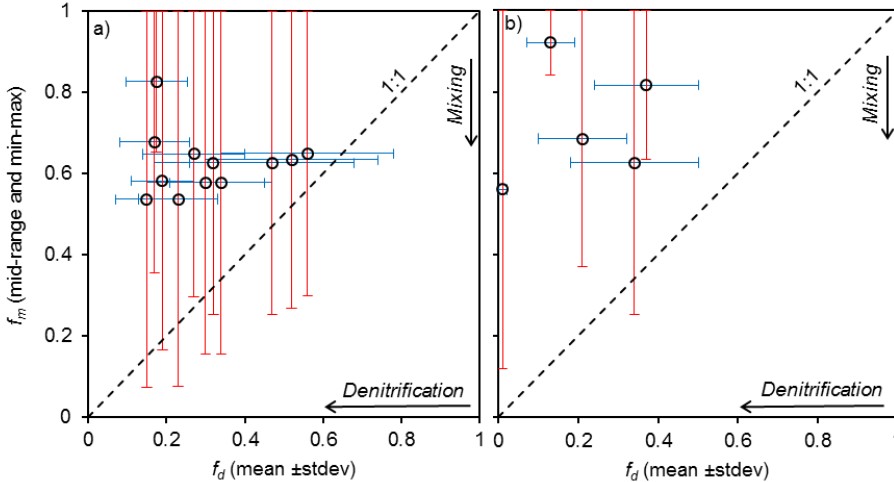

**Figure 7 Relative contributions to NO₃⁻ attenuation by mixing and denitrification, as indicated by estimated $f_m$ and $f_d$ at (a) CFO1 and (b) CFO4, for groundwater samples with denitrification indicated by stable isotope values of NO₃⁻.**

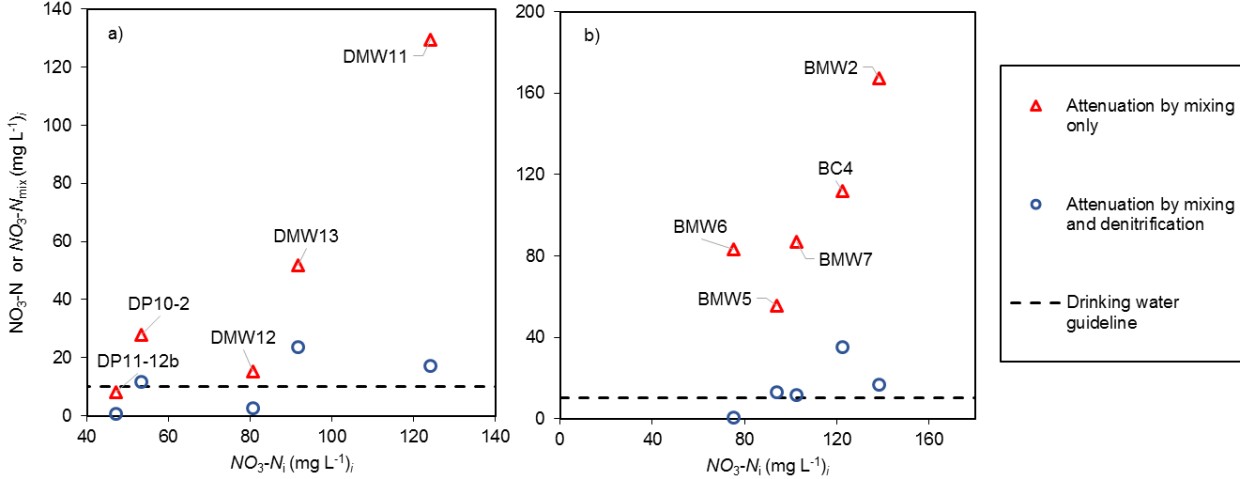

**Figure 8 Measured concentrations of NO₃-N (blue circles - attenuation by mixing and denitrification) and $NO_3$-$N_{mix}$**
10 **(red triangles - attenuation by mixing only) vs mid-range estimate of NO₃-$N_i$ at a) CFO1 and b) CFO4. Dashed lines are drinking water guideline (10 mg L⁻¹ of NO₃-N).**

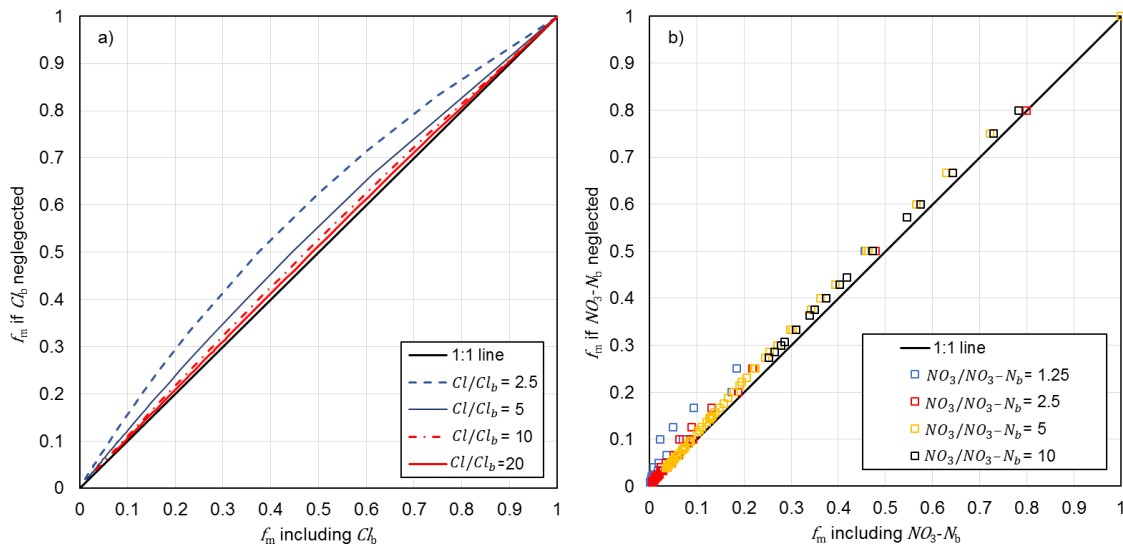

**Figure 9 Effect of neglecting background concentrations ($Cl_b$ or $NO_3$-$N_b$) in the mixing model on calculated $f_m$ over the range of values in this study.**