# Peer review of "Sources and fate of nitrate in groundwater at agricultural operations overlying glacial sediments"

_Hydrology and Earth System Sciences, 2018_

## Short Comment (SC1) · 27 Feb 2018

Dear authors,

Thank you for this interesting manuscript, in a challenging environment no doubts. I have a few suggestions to make for you to consider. First, I think providing a summary of N transport and transformations (including dissolved organic nitrogen and $NH_4^+$) would have been useful. I note that data on $NH_4^+$ and DON (DON = TKN - $NH_4^+$) is presented in Supplementary Material, but a summary in the main text is warranted. To receiving environments, whether the N is DON, $NH_4^+$ or $NO_3^-$ matters little.

[Figure]

There is also limited information presented on the redox conditions in the three 'bio-geochemical environments' present (waste piles/vadose zone/groundwater). Do these differ? This matters because the best conditions for N attenuation will be when we have alternating aerobic and anaerobic environments during transport. Is this the case here? Is there an effect of a shallower vadose zone at one of the sites (less nitrification)?

I would also suggest to de-emphasise 'denitrification' as a removal mechanism for nitrate unless you have additional measurements to back this up. In what seems like an heterogeneous environment, a range of processes could remove $NH_4+$ and $NO_3-$ (anammox, $NH_3$ volatilisation, dissimilatory $NO_3-$ reduction to $NH_4$...). It would be more prudent perhaps to talk about 'nitrate removal' instead of denitrification? The large variations in the stable isotope of nitrate data argues for a more complicated story than just nitrification-denitrification.

I hope this is helpful.

---

## Author Comment (AC1) · 6 Apr 2018

Hi Sebastian, Thank you for your thoughtful comments on the manuscript, and my apologies that it has taken me so long to reply.

Thanks for letting me know that a summary of DON, NH4 etc in the body of the text is helpful to the reader. I can do that, no problem.

Your second point is about redox conditions. I totally agree with you that it would be preferable to have measurements of redox conditions at these sites. The vadose zone and fractured upper part of the clay are likely to have been oxic, and below that there is

often a colour change that suggests reducing conditions, but we did not measure redox directly. I would imagine that the redox conditions could vary within a manure pile, with oxic conditions at the margin and reducing conditions deeper in the pile, but we also did not measure this. I will look for published studies from similar sites that can provide further guidance on likely redox conditions at these sites.

We do have measurements of DO in the monitoring wells (see Table attached), but on considering the sampling method I didn't feel confident that they reflected in-situ groundwater concentrations. Because of the low-hydraulic conductivities the wells are bailed dry and then allowed to recover, then DO was measured potentially days after the well was bailed dry. I have previously conducted an experiment on hydrochemical equilibration of groundwater in an evaporation pan, and the DO was fully equilibrated with the atmosphere in 1 day. As such, I couldn't rule out that the DO concentrations measured in the recovered wells had partially equilibrated with atmosphere in some samples. Do you think it's preferable to report these data anyway, and discuss the potential for equilibration within the well prior to measurement?

And finally, on denitrification, to me this is strongly supported by the denitrification trend in the nitrate isotope data. While other mechanisms may be at play, I don't think we have measured evidence for them and discussion in the paper would be purely speculative. Happy to discuss any of the above further.

I realise the discussion period closes next week, so feel free to email me directly: sarah.bourke@uwa.edu.au

Cheers, Sarah

Please also note the supplement to this comment: https://www.hydrol-earth-syst-sci-discuss.net/hess-2018-31/hess-2018-31-AC1-supplement.pdf
31, 2018.

**Supplement:**

**1   MEASURED pH, DO AND ION CONCENTRATIONS**

2   Table R1 Measured pH, DO and concentrations of chloride ($Cl^-$), bicarbonate ($HCO_3^-$), dissolved
3   organic carbon (DOC) and N-species in groundwater wells and water filtered from the EMS and
4   catch-basin at CFO1 (mean ± standard deviation).

[revised manuscript text omitted]

6 Table R2 Measured pH, DO and concentrations of chloride (Cl⁻), bicarbonate (HCO₃⁻), dissolved
7 organic carbon (DOC) and N-species in groundwater wells and water filtered from the EMS at CFO4
8 (mean ± standard deviation).

| Sample ID | pH | DO (mg L⁻¹) | Cl⁻ (mg L⁻¹) | HCO₃⁻ (mg L⁻¹) | DOC (mg L⁻¹) | NH₃-N (mg L⁻¹) | NO₃-N* (mg L⁻¹) | NO₂-N (mg L⁻¹) | TKN (mg L⁻¹) |
|---|---|---|---|---|---|---|---|---|---|
| EMS filtrate | - | - | 1074 ± 379 (n=2) | 5795 ± 1544 (n=2) | 3367 (n=1) | 865 ± 182 (n=2) | <0.7 (n=2) | - | - |
| BC1 | 7.2 ± 0.7 (n=9) | 1.2 ± 1.7 (n=8) | <10 (n=11) | 494 ± 13 (n=11) | 5.0 ± 0.8 (n=4) | <0.1 (n=11) | <0.1 (n=11) | <0.1 (n=11) | <0.2 (n=11) |
| BC2 | 7.0 ± 0.5 (n=10) | 1.3 ± 1.4 (n=9) | 6 ± 3 (n=12) | 516 ± 33 (n=12) | 6.0 ± 3.0 (n=4) | <0.1 (n=12) | 1.1 ± 2.7 (n=12) | <0.1 (n=12) | 0.3 ± 0.2 (n=12) |
| BC3 | 7.0 ± 0.5 (n=11) | 0.8 ± 0.8 (n=10) | <5 (n=13) | 504 ± 21 (n=13) | 6.9 ± 2.9 (n=4) | <0.1 (n=13) | 0.1 ± 0.1 (n=13) | <0.1 (n=13) | 0.2 ± 0.1 (n=13) |
| BC4 | 7.2 ± 0.3 (n=21) | 1.3 ± 1.9 (n=10) | 58 ± 64 (n=24) | 576 ± 110 (n=24) | 9.2 ± 3.5 (n=9) | <0.1 (n=24) | 8.8 ± 13.2 (n=24) | <0.1 (n=24) | 0.8 ± 0.8 (n=24) |
| BC5 | 7.1 ± 0.2 (n=6) | 2.3 ± 1.5 (n=6) | 26 ± 6 (n=8) | 498 ± 51 (n=8) | 6.8 ± 3.1 (n=3) | <0.1 (n=8) | 5.7 ± 1.5 (n=8) | <0.1 (n=8) | 0.6 ± 0.4 (n=8) |
| BMW1 | 6.9 ± 0.2 (n=23) | 1.3 ± 1.1 (n=11) | 305 ± 251 (n=28) | 926 ± 190 (n=28) | 21.5 ± 12.4 (n=11) | <0.1 (n=28) | 2.2 ± 2.5 (n=28) | <0.1 (n=28) | 1.1 ± 0.9 (n=28) |
| BMW2 | 6.4 ± 1.5 (n=20) | 3.1 ± 1.8 (n=12) | 502 ± 97 (n=22) | 1186 ± 87 (n=22) | 20.2 ± 4.9 (n=9) | <0.1 (n=22) | 6.0 ± 7.4 (n=22) | 0.1 ± 0.1 (n=22) | 1.7 ± 0.4 (n=22) |
| BMW3 | 6.9 ± 0.3 (n=20) | 1.5 ± 1.3 (n=9) | 182 ± 81 (n=25) | 881 ± 146 (n=25) | 15.6 ± 3.3 (n=9) | <0.1 (n=25) | 17.4 ± 10.3 (n=25) | 0.1 ± 0.1 (n=25) | 1.6 ± 0.8 (n=25) |
| BMW4 | 7.0 ± 0.2 (n=17) | 1.3 ± 1.1 (n=11) | 188 ± 74 (n=24) | 666 ± 55 (n=24) | 12.0 ± 3.3 (n=11) | <0.1 (n=24) | 33.6 ± 21.1 (n=24) | 0.2 ± 0.3 (n=24) | 2.6 ± 2.9 (n=24) |
| BMW5 | 6.8 ± 0.2 (n=6) | 1.5 ± 1.5 (n=6) | 106 ± 23 (n=8) | 975 ± 163 (n=8) | 8.6 ± 1.3 (n=3) | <0.1 (n=8) | 6.5 ± 4.8 (n=8) | 0.1 ± 0 (n=8) | 0.7 ± 0.3 (n=8) |
| BMW6 | 7.0 ± 0.3 (n=6) | 1.6 ± 1.3 (n=6) | 156 ± 18 (n=8) | 538 ± 27 (n=8) | 6.9 ± 1.7 (n=3) | <0.1 (n=8) | 0.4 ± 0.2 (n=8) | 0.1 ± 0 (n=8) | 0.5 ± 0.1 (n=8) |
| BMW7 | 6.8 ± 0.2 (n=6) | 2.4 ± 2.4 (n=6) | 127 ± 15 (n=8) | 699 ± 65 (n=8) | 8.1 ± 2.8 (n=3) | <0.1 (n=8) | 9.2 ± 3.0 (n=8) | 0.1 ± 0 (n=8) | 0.7 ± 0.4 (n=8) |
| BP10-15e | 7.3 ± 0.2 (n=15) | 1.3 ± 1.3 (n=9) | 7 ± 4 (n=19) | 493 ± 33 (n=19) | 3.4 ± 0.4 (n=7) | 0.1 ± 0.1 (n=19) | 0.1 ± 0.1 (n=19) | <0.1 (n=19) | 0.2 ± 0.1 (n=19) |
| BP10-15w | 7.2 ± 0.1 (n=9) | 0.8 ± 0.6 (n=9) | <5 (n=17) | 507 ± 11 (n=17) | 3.5 ± 0.6 (n=4) | <0.2 (n=17) | 0.3 ± 1.0 (n=17) | <0.1 (n=17) | 0.2 ± 0.1 (n=17) |
| BP5-15 | 7.2 ± 0.1 (n=6) | 1.5 ± 1.2 (n=6) | <5 (n=8) | 509 ± 12 (n=8) | 5.0 ± 1.1 (n=3) | <0.1 (n=8) | <0.1 (n=8) | <0.1 (n=8) | <0.2 (n=8) |
| BP6-15 | 7.1 ± 0.4 (n=6) | 1.2 ± 1.2 (n=6) | <5 (n=7) | 487 ± 7 (n=7) | 3.3 ± 1.1 (n=3) | <0.2 (n=7) | <0.1 (n=7) | <0.1 (n=7) | <0.2 (n=7) |

9 **\*For EMS filtrate this value is NO₃-N+NO₂-N**

10

---

## Referee Comment (RC1) · 6 May 2018

1. General comments It is interesting to assess the sources and fate of agriculturally derived $NO_3-N$ by the concentration of $NO_3-N$'and 'Cl−. The idea of using fd and fm to quantify the impact of denitrification and dispersion are good. The tables and figures were displayed clearly and easy to understand.

2. Specific comments However, some specific scientific questions should be answered in this manuscript. Although the error in fm introduced by neglecting Clb was discussed by authors, however, the error range may be underestimated. The largest error (calculated as 23% by authors) may be double as the Clb (assumed as 10 mg/L by authors)

[Figure]

increased to 20 mg/L. The suggestion to improve this part in manuscript is to use an equation related to the ratio of Clb / Cli.

I would also suggest to share the Excel sheet or program used by this manuscript. (page 6, line 35 to line 38).

3. Technical corrections There are several technical corrections should be done before it can be published. 1)I notice that, the last paragraph of "introduction" belongs to "experimental site description" of "methodology". (page 3, line 7 to line 20). 2)This manuscript didn't mention what is the sampling depth for the "water table wells" in the "methodology" section. To my understanding, there were little difference between groundwater monitor well and groundwater sampling well. Normally, the groundwater sampling well take water samples in a specific range of depths. 3)Sampling frequency. I'm not sure if the sample frequency of the chloride and N species were high enough to draw the conclusion, since the sample sizes was less than 30 and standard deviation seems not low. 4)Text clarity. Section 3.1 and 3.2 mentioned several "water table wells" labeled as DMW11, DMW14, etc. However, it's not easy for reader to look for those wells from figure 1 (the site description map). So, it seems that improvement of the structure of the sections might be needed.
* * *

---

## Referee Comment (RC2) · Anonymous Referee #2 · 16 Jul 2018

This paper presents a study of using isotope (N15 and O18) to study the nitrate mixing and transport. Denitrification of nitrate was considered in this study. The authors argued that their study used isotope in a quantitative way, different from the qualitative way of previous study. This however seems an overstatement to me, because they only used the isotope data to calculate the mixing ratios and denitrification coefficients. This is not very quantitative. I also have a few questions, listed below, related to their calculation of the mixing ratios and denitrification coefficients.

The authors used in equation (4) of the two-member mixing method to calculate the nitrate mixing ratio. This does not seem right to me, because there should be more than

two nitrate sources, such as the background ambient groundwater, the direct infiltration from fertilizer, the manure source, and the transformation from ammonium to nitrate. I doubt that the two end-member method is adequate to consider the multiple sources.

To account for the denitrification, the authors used a denitrification coefficient in equation (4). While this is conceptually OK, it does not consider that denitrification is a kinetic process (zeroth-order or first order). In other words, the denitrification coefficient used in the paper cannot reflect the kinetics of denitrification.

As to denitrification, it is unclear to what extent denitrification occurs in the aquifer. The plots in Figure 2 do not support the conclusion on denitrification, because the slopes shown in Figure 2(a) are not 0.5. For well CFO4, the slope of 0.42 is close to 0.5m, and the data listed in Table 2 and the well locations shown in Figure 1 indeed support the conclusion of denitrification. But what about well CFO1?

The authors said somewhere in the manuscript that the initial nitrate concentration can be neglected. I do not think that it is a reasonable assumption for agricultural areas.

---

## Author Comment (AC2) · 10 Aug 2018

See attached supplementary pdf for responses to both referees comments

Please also note the supplement to this comment:
https://www.hydrol-earth-syst-sci-discuss.net/hess-2018-31/hess-2018-31-AC2-supplement.pdf

---

## Author Comment (AC3) · 10 Aug 2018

**Response to Referees**

**Sources and fate of nitrate in groundwater at agricultural operations overlying glacial sediments" by Sarah A. Bourke et al.**

**Referee #1**

**1. General comments** It is interesting to assess the sources and fate of agriculturally derived NO3−N by the concentration of NO3-NâAL'and â ˘ AL'Cl ˘ −. The idea of using fd and fm to quantify the impact of denitrification and dispersion are good. The tables and figures were displayed clearly and easy to understand.

*Thanks for taking the time to review, we are pleased that you saw value in the manuscript.*

**2. Specific comments** However, some specific scientific questions should be answered in this manuscript. Although the error in fm introduced by neglecting Clb was discussed by authors, however, the error range may be underestimated. The largest error (calculated as 23% by authors) may be double as the Clb (assumed as 10 mg/L by authors) C1 increased to 20 mg/L. The suggestion to improve this part in manuscript is to use an equation related to the ratio of Clb / Cli. I would also suggest to share the Excel sheet or program used by this manuscript. (page 6, line 35 to line 38).

*Our thanks to the reviewer for drawing our attention to this section of the manuscript. We propose to update this discussion of the error introduced by neglecting background concentrations in response to this comment and Referee 2's comment about background $NO_3$.*

*The assumption that background concentrations can be neglected is a very useful simplifying assumption and is consistent with our understanding of the sites investigated. As such, we prefer to retain it in the final manuscript. Rather than altering the mathematical treatment, we suggest including a new figure (Figure 9) demonstrating the influence of background concentrations on the calculated $f_m$.*

*In lieu of an excel spreadsheet we propose adding a new table to Supplementary Material outlining values of each of terms in the mixing model calculation for each sample. The only thing this Table doesn't include is the solver code in Excel, which is rudimentary.*

***Proposed change***

*Delete text p12 L35-37: if we assume a $Cl_b$ of 10 mg L-1 and a Cli of 100 mg L-1, the error in $f_m$ introduced by neglecting $Cl_b$ is 9%; if $Cl_b$ is 20 mg L-1, the error is 23%.*

*Replace with text: The error introduced by neglecting background concentrations was assessed by comparing $f_m$ calculated with and without background concentrations included, using the full range of values in this study (Fig. 9). Neglecting background concentrations results in overestimation of $f_m$ (i.e. underestimation of the amount of attenuation mixing) with the largest errors when measured concentrations are close to background concentrations. For Cl the maximum difference of 0.13 is in the mid-range of $f_m$ values. For $NO_3$-N, the difference is consistently < 0.1 with the largest errors at the lowest values of $f_m$.*

[Figure]

*Figure 9 Effect of neglecting background concentrations in the mixing model on calculated $f_m$ over the range of values in this study.*

*Insert new Table S10 in Supplementary Material:*

*Table S10 Constraining values and results of mixing model calculations*

| Sample ID | Cl | NO$_3$-N | $f_d$ | NO$_3$-N$_i$/Cl$_i$ | Cl$_i$ (mg L$^{-1}$) | | NO$_3$-N$_i$ (mg L$^{-1}$) | | $f_m$ | |
|---|---|---|---|---|---|---|---|---|---|---|
| | (mg L$^{-1}$) | (mg L$^{-1}$) | (mean ± stdev) | (mean ± stdev) | min | max | min | max | min | max |
| **CFO1** | | | | | | | | | | |
| DMW11 | 436.1 | 17.1 | 0.17 ± 0.07 | 0.23 ± 0.10 | 436 | 667 | 98 | 150 | 0.65 | 1 |
| DMW12 | 78.0 | 2.6 | 0.23 ± 0.10 | 0.14 ± 0.06 | 78 | 1047 | 11 | 150 | 0.07 | 1 |
| DMW13 | 56.7 | 23.7 | 0.56 ± 0.22 | 0.75 ± 0.29 | 57 | 189 | 42 | 141 | 0.30 | 1 |
| DP10-2 | 74.5 | 11.8 | 0.52 ± 0.22 | 0.30 ± 0.13 | 74 | 277 | 23 | 84 | 0.27 | 1 |
| DP11-12b | 95.7 | 0.6 | 0.15 ± 0.08 | 0.04 ± 0.02 | 96 | 1300 | 4.2 | 90 | 0.07 | 1 |
| DC15-22_10m | 73.0 | 11.0 | 0.47 ± 0.21 | 0.32 ± 0.14 | 73 | 289 | 23 | 93 | 0.25 | 1 |
| DP11-13_4.3m | 28.5 | 7.0 | 0.30 ± 0.15 | 0.82 ± 0.41 | 29 | 184 | 23 | 150 | 0.15 | 1 |
| DP11-13_5.2m | 25.0 | 7.8 | 0.34 ± 0.13 | 0.91 ± 0.35 | 25 | 160 | 23 | 146 | 0.16 | 1 |
| DP11-13_7m | 72.3 | 12.0 | 0.27 ± 0.13 | 0.62 ± 0.30 | 72 | 244 | 45 | 150 | 0.30 | 1 |
| DP11-13 _7.9m | 70.8 | 9.1 | 0.17 ± 0.09 | 0.76 ± 0.40 | 71 | 199 | 54 | 150 | 0.36 | 1 |
| DP11-13_8.8m | 81.7 | 11.0 | 0.32 ± 0.15 | 0.89 ± 0.42 | 82 | 323 | 39 | 150 | 0.25 | 1 |
| **CFO4** | | | | | | | | | | |
| BC4 | 163.1 | 35.1 | 0.37 ± 0.13 | 0.58 ± 0.20 | 163 | 258 | 95 | 150 | 0.63 | 1 |
| BMW2 | 595.6 | 16.5 | 0.13 ± 0.06 | 0.21 ± 0.10 | 596 | 707 | 127 | 150 | 0.84 | 1 |
| BMW5 | 131.2 | 12.9 | 0.34 ± 0.16 | 0.29 ± 0.14 | 131 | 520 | 38 | 150 | 0.25 | 1 |
| BMW6 | 156.0 | 0.4 | 0.01 ± 0.01 | 0.26 ± 0.26 | 156 | 1300 | 0.4 | 150 | 0.12 | 1 |
| BMW7 | 134.7 | 11.6 | 0.21 ± 0.11 | 0.41 ± 0.22 | 135 | 365 | 55 | 150 | 0.37 | 1 |

**3. Technical corrections** There are several technical corrections should be done before it can be published.

1)I notice that, the last paragraph of "introduction" belongs to "experimental site description" of "methodology". (page 3, line 7 to line 20).

*We agree that Lines 8-15 on p3 are a description of the sites and these can be moved to the top of methods as a site description subsection.*

***Proposed change**￼*

*Final paragraph of Introduction becomes:*

In this study, we present the application of this approach at two confined feeding operations (CFOs) in Alberta, Canada, with differing lithologies and durations of operation. Concentrations of $Cl^-$ and nitrogen species (N-species) and the stable isotopes of $NO_3^-$ were measured in groundwater samples collected from monitoring wells and continuous soil cores, as well as manure filtrate at both sites. These data were interpreted to (1) assess the extent of agriculturally derived $NO_3^-$ in groundwater, (2) identify sources and initial concentrations of $NO_3^-$ at the point of entry to the groundwater system, and (3) assess the dominant attenuation mechanisms controlling subsurface $NO_3^-$ distributions at these sites.

*A new sub-section added to Methods as follows, and numbering of subsequent sections of Methods updated accordingly:*

*2.1 Experimental site description*

The first study area (CFO1), located 25 km northeast of Lethbridge, Alberta, was established in 1928 and had approximately 150 head of dairy cattle at the time of the study (Fig. 1). An associated earthen manure storage (EMS) facility for storing liquid dairy manure was constructed in the 1960s. A 2000-head beef feedlot, established in the 1960s, was also present at CFO1. The second study area (CFO4), located approximately 30 km north of Red Deer, Alberta and 300 km north of CFO1, was constructed in 1995 (including an EMS) and had 350 head of dairy cattle at the time of the study. To the best of our knowledge, fertilizers have not been applied at either of these sites, and infiltration of manure waste is assumed to be the cause of elevated $NO_3^-$ concentrations in the local groundwater.

2)This manuscript didn't mention what is the sampling depth for the "water table wells" in the "methodology" section. To my understanding, there were little difference between groundwater monitor well and groundwater sampling well. Normally, the groundwater sampling well take water samples in a specific range of depths.

*The screen intervals of all wells are presented in Table 1 (cited p3 L25) along with a description of monitoring wells in 2.1.1. Table 1 reports total well depth and screen length along with the statement in the caption that all screens are at the bottom of the well – which allows the reader to easily determine screen depth for each well.*

*The distinction between water table wells and piezometers is about the screen length and potential for the screen interval to include part of the unsaturated zone. For a water table well the screen interval is ~4 m and is screened so that water levels will be within the screen interval throughout seasonal or annual water table fluctuations. Piezometers are screened at discrete depths within the*

*aquifer and in this study screen lengths were usually 0.5 m. The full length of these piezometers remains within the saturated zone at all times. This distinction is relatively standard within the North American hydrogeology community.*

*We provide this information for the benefit of readers, but once defining these terms simply refer to both as monitoring wells through-out the manuscript. We feel that this approach provides a good balance between providing detailed information if the reader desires it without unnecessarily complicating the text.*

***Suggest no change made***

3)Sampling frequency. I'm not sure if the sample frequency of the chloride and N species were high enough to draw the conclusion, since the sample sizes was less than 30 and standard deviation seems not low.

*The Cl and $NO_3$ concentrations were measured at monthly to quarterly sampling intervals over a period of approximately 5 years and adequately capture temporal variation (see Figures 3 and 5). However, the isotope data are effectively a snapshot in time and do not capture temporal variation. This is already noted in the text (p12 L40-P13 L2).*

*A sample size of 30 (note that Cl and $NO_3$ data set is larger than this) is not unusually small for a study of nitrate in groundwater using isotopes. A brief survey of published papers yields: n= 16 (Mengis et al., 2001), n=29 (Mariotti et al., 1988), n = 24 (Durka et al., 1994).*

*It isn't entirely clear which conclusion(s) the reviewer thinks are not supported by the data. We acknowledge the limitations of the individual data sets, which is why the conclusions were drawn from a synthesised analysis of multiple lines of evidence that included the spatial and temporal distribution of $NO_3$ in groundwater and sources, the isotopic composition of that NO3 and the mixing model results.*

*We believe that the conclusion that denitrification is proceeding in the groundwater system and that denitrification reduces $NO_3$ concentrations substantially at the farm-scale is strongly supported by the data. The attribution of sources has more uncertainty in it, but nonetheless, we feel that the spatial distribution of $NO_3$ as well as the mixing ratio analysis supports the conclusion that earthen manure storages are not the largest source of agricultural nitrate in groundwater at these sites.*

***Suggest no change***

4)Text clarity. Section 3.1 and 3.2 mentioned several "water table wells" labeled as DMW11, DMW14, etc. However, it's not easy for reader to look for those wells from figure 1 (the site description map)

*It's not clear exactly what the issue is here. DMW11 is clearly visible at the top of Figure 1, DMW14 is clearly visible on the RHS of the inset (area covered delineated by blue rectangle as stated in caption), which is included specifically so that these closely spaced wells can be identified. If further guidance can be provided we have no problem making adjustments so that it is easier for the reader to understand. Perhaps just having the Figure 1 imbedded in the text rather than at the end of the manuscript will help?*

***Suggest no change made***

**Referee #2**

This paper presents a study of using isotope (N15 and O18) to study the nitrate mixing and transport. Denitrification of nitrate was considered in this study. The authors argued that their study used isotope in a quantitative way, different from the qualitative way of previous study. This however seems an overstatement to me, because they only used the isotope data to calculate the mixing ratios and denitrification coefficients. This is not very quantitative.

*In the Introduction, we state that this paper uses nitrate isotopes to identify denitrification and quantify the fraction of nitrate that has been denitrified. This is in contrast to other published papers, which use nitrate isotopes to identify denitrification, but do not use the isotope fractionation to quantitatively estimate the fraction of $NO_3$ that has been lost through denitrification. We believe this is a clear distinction and our results are correctly described as quantitative.*

*To highlight the quantitative nature of the analysis we propose an additional figure (Figure 8) that shows the reduction in NO3-N concentrations associated with mixing and denitrification.*

**_Proposed change_**

*Additional text added to Section 3.4 Mechanisms of attenuation and associated figure:*

*The concentration of $NO_3$-N that would be measured if mixing was the only attenuation mechanism ($NO_3$-$N_{mix}$) was calculated by dividing the measured concentration by $f_d$. Comparison with measured concentrations (which reflect attenuation by both mixing and denitrification) suggests that the sample from 20 m depth (DP11-12b) is the only sample that would be below the drinking water guideline if mixing was the only attenuation mechanism (Fig. 8).*

[Figure]

*Figure 8 Measured concentrations of $NO_3$-N (blue circles - attenuation by mixing and denitrification) and $NO_3$-N/$f_d$ (red triangles - attenuation by mixing only) vs mid-range estimate of $NO_3$-$N_i$. Dashed line is drinking water guideline (10 mg $L^{-1}$ of $NO_3$-N).*

I also have a few questions, listed below, related to their calculation of the mixing ratios and denitrification coefficients. The authors used in equation (4) of the two-member mixing method to calculate the nitrate mixing ratio. This does not seem right to me, because there should be more than C1 two nitrate sources, such as the background ambient groundwater, the direct infiltration

from fertilizer, the manure source, and the transformation from ammonium to nitrate. I doubt that the two end-member method is adequate to consider the multiple sources.

*To the best of our knowledge (including interviews with long-time landowners/farmers at the sites, historic air photos) fertilizer (other than manure) has not been applied at the sites. As such, manure is the only source of agricultural nitrate at these study sites, which is stated in the original manuscript (p3 L13-15).*

*The nitrate in groundwater will have originally been ammonium, and $NH_3$ dominates N in the EMS at both sites (see Supp material). However, the data demonstrate that $NH_3$ is generally a relatively small component of total-N in groundwater (<10%) so that $NH_3$ can be neglected in the mixing model, which only considers the N-pool in the groundwater system.*

*At CFO4 ammonium concentrations in the groundwater system are negligible (consistently < 0.23 mg/L. At CFO1, well DMW3 directly adjacent to the EMS has $NH_3$ is present at high concentrations in and $NO_3$ concentrations were low, but the mixing calculation were not conducted on data from this well. In samples for which the mixing calculation was conducted $NH_3$ was <10% of total-N (note that $NH_3$ wasn't measured in core from DP11-13 were collected at depths <9 m. Likely $NH_3$ concentrations in these samples were inferred from the water table well at this location which was screened to 7 m depth). Also, $NO_2$ concentrations in groundwater at CFO1, which would be expected to be elevated in the presence of nitrification (Vogel et al., 1981), was consistently < 0.5 mg/L (see p12 L14-23 and Supplementary material). The one exception is core sample DP22_6.5m, which will be removed from the mixing calculation results (Tables, Figures and text).*

*As such, we believe that it is reasonable to use a two-end member mixing model for the samples reported at these sites, where the end-members represent manure-based $NO_3$ and background (pre-agricultural) concentrations.*

**Proposed change**

*Mixing calculation results from core sample DP22_6.5m will be removed from the mixing calculation results (Tables, Figures and text).*

*Additional text p7 L11:*

This mixing calculation was only conducted on samples for which $NO_3$ dominated total-N ($NH_3$-N <10% of $NO_3$-N) so that nitrification of $NH_3$ could be neglected.

*Additional text p12 L37:*

In this study we applied a two-end member mixing model and assumed that background concentrations can be neglected. Although applicable at these sites, this approach may not be valid at other sites if additional sources of $NO_3$ in groundwater (e.g. fertilizer or nitrification) are significant, or if $NO_3$ concentrations in groundwater are naturally elevated (Hendry et al., 1984).

To account for the denitrification, the authors used a denitrification coefficient in equation (4). While this is conceptually OK, it does not consider that denitrification is a kinetic process (zeroth-order or first order). In other words, the denitrification coefficient used in the paper cannot reflect the kinetics of denitrification.

*In this study we assume that fractionation of $NO_3$ in groundwater during denitrification follows a Rayleigh distillation process, as described in Section 2.3.1. This approach has been used in numerous*

*previously published studies of denitrification in groundwater (Böttcher et al., 1990; Otero et al., 2009; Xue et al., 2009).*

*The kinetics of the reaction and rates of denitrification are likely to be reflected in the epsilon value (Kendall and Aravena, 2000). In this study we determined a global epsilon of -10 based on data across both sites. In the model, epsilon values were allowed to vary in accordance with a normal distribution (mean = -10, stdev = -2.5), which will reflect a range of possible reaction rates.*

**Proposed change**

*Additional references at p5 L19 citing previous studies that used a Rayleigh distillation approach to assess denitrification in groundwater: (Böttcher et al., 1990; Otero et al., 2009; Xue et al., 2009).*

As to denitrification, it is unclear to what extent denitrification occurs in the aquifer. The plots in Figure 2 do not support the conclusion on denitrification, because the slopes shown in Figure 2(a) are not 0.5. For well CFO4, the slope of 0.42 is close to 0.5m, and the data listed in Table 2 and the well locations shown in Figure 1 indeed support the conclusion of denitrification. But what about well CFO1?

*We assume that the reviewer takes issue with the slope of the isotopic enrichment trend at CFO1 (0.72) as being not close enough to the general trend of 0.5 reported in some studies (Durka et al., 1994). However the value of 0.72 is not unreasonable given the range of values reported for denitrification of groundwater in the published literature of 0.47 – 0.66 (Singleton et al., 2007), 0.67 (Mengis et al., 1999), 0.77 (Fukada et al., 2003). This will be clarified in the revised text.*

**Proposed change**

*Update text p5 L13-14: Nitrate in groundwater that has undergone denitrification is commonly reported as being identified by enrichment of $\delta^{15}N_{NO3}$ and $\delta^{18}O_{NO3}$ with a slope of about 0.5 on a cross-plot (Clark and Fritz, 1997). However, published studies of denitrification in groundwater report slopes of up to 0.77 (Mengis et al., 1999, Singelton et al., 2007, Fukada et al., 2003).*

The authors said somewhere in the manuscript that the initial nitrate concentration can be neglected. I do not think that it is a reasonable assumption for agricultural areas.

*Presumably the reviewer is suggesting that there can be an historical legacy of nitrate in agricultural areas. This is true, and we consider "background" as not having been influenced by agricultural activity (whether this is recent or historical). This assumption that $NO_{3b}$ can be neglected underpins the simplification of the mathematics and is valid for these agricultural areas.  This approach would not be suitable at sites with naturally elevated nitrate concentrations in groundwater.*

**Proposed Change**

*A new figure (Figure 8) will be added to the revised manuscript and a discussion of the effect of neglecting background concentrations updated – see response to Referee 1.*

*Additional text as previously proposed p12 L37:*

*In this study we applied a two-end member mixing model and assumed that background concentrations can be neglected. Although applicable at these sites, this approach may not be valid*

*at other sites if additional sources of NO₃ in groundwater  (e.g. fertilizer or nitrification) are significant, or if NO₃ concentrations in groundwater are naturally elevated (Hendry et al., 1984).*

**References**

Böttcher J, Strebel O, Voerkelius S, Schmidt HL. Using isotope fractionation of nitrate-nitrogen and nitrate-oxygen for evaluation of microbial denitrification in a sandy aquifer. Journal of Hydrology 1990; 114: 413-424.

Durka W, Schulze E-D, Gebauer G, Voerkeliust S. Effects of forest decline on uptake and leaching of deposited nitrate determined from 15N and 18O measurements. Nature 1994; 372: 765-767.

Fukada T, Hiscock KM, Dennis PF, Grischek T. A dual isotope approach to identify denitrification in groundwater at a river-bank infiltration site. Water Research 2003; 37: 3070-3078.

Hendry MJ, McCready RGL, Gould WD. Distribution, source and evolution of nitrate in a glacial till of southern Alberta, Canada. Journal of Hydrology 1984; 70: 177-198.

Kendall C, Aravena R. Nitrate Isotopes in Groundwater Systems. In: Cook P, Herczeg A, editors. Environmental Tracers in Subsurface Hydrology. Springer US, 2000, pp. 261-297.

Mariotti A, Landreau A, Simon B. 15N isotope biogeochemistry and natural denitrification process in groundwater: Application to the chalk aquifer of northern France. Geochimica et Cosmochimica Acta 1988; 52: 1869-1878.

Mengis M, Schif SL, Harris M, English MC, Aravena R, Elgood RJ, et al. Multiple Geochemical and Isotopic Approaches for Assessing Ground Water NO3– Elimination in a Riparian Zone. Ground Water 1999; 37: 448-457.

Mengis M, Walther U, Bernasconi SM, Wehrli B. Limitations of using δ18O for the source identification of nitrate in agricultural soils. Environmental science & technology 2001; 35: 1840-1844.

Otero N, Torrentó C, Soler A, Menció A, Mas-Pla J. Monitoring groundwater nitrate attenuation in a regional system coupling hydrogeology with multi-isotopic methods: The case of Plana de Vic (Osona, Spain). Agriculture, Ecosystems & Environment 2009; 133: 103-113.

Singleton M, Esser B, Moran J, Hudson G, McNab W, Harter T. Saturated zone denitrification: potential for natural attenuation of nitrate contamination in shallow groundwater under dairy operations. Environmental science & technology 2007; 41: 759-765.

Vogel JC, Talma AS, Heaton THE. Gaseous nitrogen as evidence for denitrification in groundwater. Journal of Hydrology 1981; 50: 191-200.

Xue D, Botte J, De Baets B, Accoe F, Nestler A, Taylor P, et al. Present limitations and future prospects of stable isotope methods for nitrate source identification in surface-and groundwater. Water Research 2009; 43: 1159-1170.

---

## Author Response (AR1)

**Response to Referees**

**Sources and fate of nitrate in groundwater at agricultural operations overlying glacial sediments" by Sarah A. Bourke et al.**

**Referee #1**

**1. General comments** It is interesting to assess the sources and fate of agriculturally derived NO3−N by the concentration of NO3-NâAL'and â ˇ AL'Cl ˇ −. The idea of using fd and fm to quantify the impact of denitrification and dispersion are good. The tables and figures were displayed clearly and easy to understand.

*Thanks for taking the time to review, we are pleased that you saw value in the manuscript.*

**2. Specific comments** However, some specific scientific questions should be answered in this manuscript. Although the error in fm introduced by neglecting Clb was discussed by authors, however, the error range may be underestimated. The largest error (calculated as 23% by authors) may be double as the Clb (assumed as 10 mg/L by authors) C1 increased to 20 mg/L. The suggestion to improve this part in manuscript is to use an equation related to the ratio of Clb / Cli. I would also suggest to share the Excel sheet or program used by this manuscript. (page 6, line 35 to line 38).

*Our thanks to the reviewer for drawing our attention to this section of the manuscript; neglecting background concentrations was also raised by reviewer 2. The assumption that background concentrations can be neglected is a very useful simplifying assumption and is consistent with our understanding of the sites investigated. As such, we prefer to retain it in the final manuscript. Rather than altering the mathematical treatment, we have added a new figure (Figure 9) demonstrating that the influence of background concentrations on the calculated $f_m$ is negligible in most cases. We have also updated the discussion around this assumption (see p14 of marked up manuscript).*

[Figure]

*Figure 9 Effect of neglecting background concentrations ($Cl_b$ or $NO_3$-$N_b$) in the mixing model on calculated $f_m$ over the range of values in this study.*

*In lieu of an excel spreadsheet we have added a new table (Table S10) to the Supplementary Material outlining values of each of terms in the mixing model calculation for each sample. The only thing this Table doesn't include is the solver code in Excel, which is rudimentary.*

*Table S10 Constraining values and results of mixing model calculations*

| Sample ID | Cl (mg L$^{-1}$) | NO$_3$-N (mg L$^{-1}$) | $f_d$ (mean ± stdev) | NO$_3$-N$_i$/Cl$_i$ (mean ± stdev) | Cl$_i$ (mg L$^{-1}$) min | Cl$_i$ (mg L$^{-1}$) max | NO$_3$-N$_i$ (mg L$^{-1}$) min | NO$_3$-N$_i$ (mg L$^{-1}$) max | $f_m$ min | $f_m$ max |
|---|---|---|---|---|---|---|---|---|---|---|
| **CFO1** | | | | | | | | | | |
| DMW11 | 436.1 | 17.1 | 0.17 ± 0.07 | 0.23 ± 0.10 | 436 | 667 | 98 | 150 | 0.65 | 1 |
| DMW12 | 78.0 | 2.6 | 0.23 ± 0.10 | 0.14 ± 0.06 | 78 | 1047 | 11 | 150 | 0.07 | 1 |
| DMW13 | 56.7 | 23.7 | 0.56 ± 0.22 | 0.75 ± 0.29 | 57 | 189 | 42 | 141 | 0.30 | 1 |
| DP10-2 | 74.5 | 11.8 | 0.52 ± 0.22 | 0.30 ± 0.13 | 74 | 277 | 23 | 84 | 0.27 | 1 |
| DP11-12b | 95.7 | 0.6 | 0.15 ± 0.08 | 0.04 ± 0.02 | 96 | 1300 | 4.2 | 90 | 0.07 | 1 |
| DC15-22_10m | 73.0 | 11.0 | 0.47 ± 0.21 | 0.32 ± 0.14 | 73 | 289 | 23 | 93 | 0.25 | 1 |
| DP11-13_4.3m | 28.5 | 7.0 | 0.30 ± 0.15 | 0.82 ± 0.41 | 29 | 184 | 23 | 150 | 0.15 | 1 |
| DP11-13_5.2m | 25.0 | 7.8 | 0.34 ± 0.13 | 0.91 ± 0.35 | 25 | 160 | 23 | 146 | 0.16 | 1 |
| DP11-13_7m | 72.3 | 12.0 | 0.27 ± 0.13 | 0.62 ± 0.30 | 72 | 244 | 45 | 150 | 0.30 | 1 |
| DP11-13 _7.9m | 70.8 | 9.1 | 0.17 ± 0.09 | 0.76 ± 0.40 | 71 | 199 | 54 | 150 | 0.36 | 1 |
| DP11-13_8.8m | 81.7 | 11.0 | 0.32 ± 0.15 | 0.89 ± 0.42 | 82 | 323 | 39 | 150 | 0.25 | 1 |
| **CFO4** | | | | | | | | | | |
| BC4 | 163.1 | 35.1 | 0.37 ± 0.13 | 0.58 ± 0.20 | 163 | 258 | 95 | 150 | 0.63 | 1 |
| BMW2 | 595.6 | 16.5 | 0.13 ± 0.06 | 0.21 ± 0.10 | 596 | 707 | 127 | 150 | 0.84 | 1 |
| BMW5 | 131.2 | 12.9 | 0.34 ± 0.16 | 0.29 ± 0.14 | 131 | 520 | 38 | 150 | 0.25 | 1 |
| BMW6 | 156.0 | 0.4 | 0.01 ± 0.01 | 0.26 ± 0.26 | 156 | 1300 | 0.4 | 150 | 0.12 | 1 |
| BMW7 | 134.7 | 11.6 | 0.21 ± 0.11 | 0.41 ± 0.22 | 135 | 365 | 55 | 150 | 0.37 | 1 |

**3. Technical corrections** There are several technical corrections should be done before it can be published.

1) I notice that, the last paragraph of "introduction" belongs to "experimental site description" of "methodology". (page 3, line 7 to line 20).

*We agree that Lines 8-15 on p3 are a description of the sites and these have been moved to the top of methods as a separate site description subsection (2.1 Experimental Sites). This new section contains an expanded description of these two study sites (see p4 of marked up manuscript).*

2) This manuscript didn't mention what is the sampling depth for the "water table wells" in the "methodology" section. To my understanding, there were little difference between groundwater monitor well and groundwater sampling well. Normally, the groundwater sampling well take water samples in a specific range of depths.

*The screen intervals of all wells are presented in Table 1 (cited p3 L25) along with a description of monitoring wells in 2.1.1. Table 1 reports total well depth and screen length along with the statement in the caption that all screens are at the bottom of the well – which allows the reader to easily determine screen depth for each well. We feel that this is an efficient way of presenting the data and would prefer to retain it in the manuscript.*

*The distinction between water table wells and piezometers is about the screen length and potential for the screen interval to include part of the unsaturated zone. For a water table well the screen interval is ~4 m (at these sites) and is screened so that water levels will be within the screen interval throughout seasonal or annual water table fluctuations. Piezometers are screened at discrete depths within the aquifer and in this study screen lengths were usually 0.5 m. The full length of these piezometers remains within the saturated zone at all times. This distinction is relatively standard within the North American hydrogeology community.*

*We provide this information for the benefit of readers, but once defining these terms simply refer to both as monitoring wells through-out the manuscript. We feel that this approach provides a good balance between providing detailed information if the reader desires it without unnecessarily complicating the text.*

***No change made***

3)Sampling frequency. I'm not sure if the sample frequency of the chloride and N species were high enough to draw the conclusion, since the sample sizes was less than 30 and standard deviation seems not low.

*The Cl and $NO_3$ concentrations were measured at monthly to quarterly sampling intervals over a period of approximately 5 years and adequately capture temporal variation (see Figures 3 and 5). However, the isotope data are effectively a snapshot in time and do not capture temporal variation. This was already noted in the text (p12 L40-P13 L2 of original manuscript).*

*A sample size of 30 (note that Cl and $NO_3$ data set is larger than this) is not unusually small for a study of nitrate in groundwater using isotopes. A brief survey of published papers yields: n= 16 (Mengis et al., 2001), n=29 (Mariotti et al., 1988), n = 24 (Durka et al., 1994).*

*It isn't entirely clear which conclusion(s) the reviewer thinks are not supported by the data. We acknowledge the limitations of the individual data sets, which is why the conclusions were drawn from a synthesised analysis of multiple lines of evidence that included the spatial and temporal distribution of $NO_3$ in groundwater and sources, the isotopic composition of that NO3 and the mixing model results.*

*We believe that the conclusion that denitrification is proceeding in the groundwater system and that denitrification reduces $NO_3$ concentrations substantially at the farm-scale is strongly supported by the data. The attribution of sources has more uncertainty in it, but nonetheless, we feel that the spatial distribution of $NO_3$ as well as the mixing ratio analysis supports the conclusion that temporary piles and pens are equal or more significant sources of agricultural nitrate in groundwater at these sites.*

***No change made***

4)Text clarity. Section 3.1 and 3.2 mentioned several "water table wells" labeled as DMW11, DMW14, etc. However, it's not easy for reader to look for those wells from figure 1 (the site description map)

*It's not clear exactly what the issue is here. DMW11 is clearly visible at the top of Figure 1, DMW14 is clearly visible on the RHS of the inset (area covered delineated by blue rectangle as stated in caption), which is included specifically so that these closely spaced wells can be identified. If further guidance can be provided we have no problem making adjustments so that it is easier for the reader to understand. Perhaps just having the Figure 1 imbedded in the text rather than at the end of the manuscript will help?*

**No change made**

**Referee #2**

This paper presents a study of using isotope (N15 and O18) to study the nitrate mixing and transport. Denitrification of nitrate was considered in this study. The authors argued that their study used isotope in a quantitative way, different from the qualitative way of previous study. This however seems an overstatement to me, because they only used the isotope data to calculate the mixing ratios and denitrification coefficients. This is not very quantitative.

*This manuscript presents the first application of the dual-isotopic enrichment of $NO_3$ to quantify the fraction of $NO_3$ removed that includes uncertainty in source values and enrichment factors. This type of calculation is commonly made using other isotopes, (e.g. calculating the amount of water lost to evaporation), but has not as yet been utilised in NO3 studies due to uncertainties in source values and enrichment factors. This is in stark contrast to the vast majority of published papers that used nitrate isotopes to identify the process of denitrification, or to define end-members for mixing calculations. We believe that this is a clear distinction and our analysis approach and results are correctly described as quantitative.*

*The text of the introduction has been extensively updated to clarify our approach and highlight the novel contribution of this manuscript to published literature (p2-3 marked up manuscript). We have also included an additional figure (Figure 8) that shows the reduction in $NO_3$-N concentrations associated with mixing and denitrification to emphasize the quantitative nature of our results, which is discussed in Section 3.4 of the revised manuscript.*

[Figure]

*Figure 8 Measured concentrations of NO₃-N (blue circles - attenuation by mixing and denitrification) and NO₃-N_mix (red triangles - attenuation by mixing only) vs mid-range estimate of NO₃-N_i at a) CFO1 and b) CFO4. Dashed lines are drinking water guideline (10 mg L⁻¹ of NO₃-N).*

I also have a few questions, listed below, related to their calculation of the mixing ratios and denitrification coefficients. The authors used in equation (4) of the two-member mixing method to calculate the nitrate mixing ratio. This does not seem right to me, because there should be more than C1 two nitrate sources, such as the background ambient groundwater, the direct infiltration from fertilizer, the manure source, and the transformation from ammonium to nitrate. I doubt that the two end-member method is adequate to consider the multiple sources.

*To the best of our knowledge (including interviews with long-time landowners/farmers at the sites, historic air photos) fertilizer (other than manure) has not been applied at the sites. As such, manure is the only source of agricultural nitrate at these study sites, which is stated in the original manuscript (p3 L13-15). This has now been clarified in the new subsection 2.1 (p4 marked up manuscript)*

*The nitrate in groundwater will have originally been organic-N or ammonium, and NH₃ dominates N in the EMS at both sites. However, the data demonstrate that NH₃ is generally a relatively small component of total-N in groundwater (<10%) so that NH₃ can be neglected in the mixing model, which only considers the N-pool in the groundwater system. A new Table has been added (Table 2) that summarises the range of values of each of the components of the N-pool. Additional description of these values has also been added to the beginning of the Section 3.3 (p10 marked up manuscript).*

*Table 2. Range of measured concentrations of TN, NH₃-N, NO_x-N (NO₂-N + NO₃-N) and TON at each study site. At CFO1 results from monitoring well DMW3 are presented separately because values in this well differed substantially from all other wells.*

| Site | N-pool | TN (mg L⁻¹) | NH₃-N (mg L⁻¹) | NO_x-N (mg L⁻¹) | TON (mg L⁻¹) |
|------|--------|-------------|----------------|-----------------|--------------|
| CFO1 | EMS | 550 – 1820 | 275 – 747 | <0.1 – 0.4 | 73 – 1301 |
| | Catch-basin | 200 – 1440 | 2.5 – 7.3 | <0.1 | 196 – 1437 |
| | DMW3 | 278 – 548 | 219 – 479 | <0.1 – 50* | 31.3 – 73.9 |
| | Other monitoring wells | <0.25 – 33.4 | <0.05 – 2.9 | <0.1 – 31.4** | <0.2 –3.7 |
| CF04 | EMS^ | 1000 – 1240 | 724 – 747 | 0.25 - 0.29 | 275 –492 |
| | Monitoring wells | <0.25 – 84.6 | <0.05 – 0.23 | <0.1 – 80.4 | <0.2 –13.9 |

*NO_x-N of 50 mg L⁻¹ in DMW3 consisted of 12.6 mg L⁻¹ as NO₃-N and 37.4 mg L⁻¹ as NO₂-N.
**NO_x-N max in groundwater measured in core (NO₃-N = 66.4 mg L⁻¹, NO_x-N = 67.8 mg L⁻¹)
^Range across three replicates measured on 25 August 2011

*At CFO4 ammonium concentrations in the groundwater system are negligible (consistently < 0.23 mg/L. At CFO1, well DMW3 directly adjacent to the EMS has $NH_3$ is present at high concentrations in and $NO_3$ concentrations were low, but the mixing calculation were not conducted on data from this well. In samples for which the mixing calculation was conducted $NH_3$ was <10% of total-N. This is now clearly stated in Section 2.4.2 (p7 of updated manuscript).*

*Also, $NO_2$ concentrations in groundwater at CFO1, which would be expected to be elevated in the presence of nitrification (Vogel et al., 1981), was consistently < 0.5 mg/L (see p12 L14-23 and Supplementary material). The one exception is core sample DP22_6.5m, which has now been removed from the mixing calculation results (see updated Tables, Figures and text).*

*As such, we believe that it is reasonable to use a two-end member mixing model for the samples reported at these sites, where the end-members represent manure-based $NO_3$ and background (pre-agricultural) concentrations. This may not be the case at other sites where fertilizer or nitrification in groundwater are significant sources of $NO_3$ in groundwater, and this is now acknowledged in the Discussion (p14 marked-up manuscript).*

To account for the denitrification, the authors used a denitrification coefficient in equation (4). While this is conceptually OK, it does not consider that denitrification is a kinetic process (zeroth-order or first order). In other words, the denitrification coefficient used in the paper cannot reflect the kinetics of denitrification.

*In this study we assume that fractionation of $NO_3$ in groundwater during denitrification follows a Rayleigh distillation process, as described in Section 2.3.1. This approach has been used in numerous previously published studies of denitrification in groundwater (Böttcher et al., 1990; Otero et al., 2009; Xue et al., 2009) and these references are now clearly cited in the manuscript (see p6 marked up version).*

*Rates of denitrification are likely to vary, and this will be reflected in the enrichment factor (Kendall and Aravena, 2000). This leads to uncertainty in the enrichment factor, and is one of the reasons that dual-isotopic enrichment of NO3 isn't widely quantified based from isotopic enrichment. This has now been clarified in the Introduction of the manuscript and in the description of the modelling approach.*

*In this study we determined a global epsilon of -10 based on data across both sites. In the model, epsilon values were allowed to vary in accordance with a normal distribution (mean = -10, stdev = -2.5), which will reflect a range of possible reaction rates. This value of epsilon was determined based on data measured at the site, as shown in Figure 2b. The slopes corresponding ± 1 std. dev. are now also shown on this Figure.*

[Figure]

*Figure 1 (a) Cross-plot of stable isotopes of nitrate at CFO1 and CFO4 showing hypothetical nitrification trend, boundary of manure-sourced $NO_3^-$ values and linear enrichment trends associated with denitrification, (b) enrichment of $\delta^{15}N_{NO3}$ during denitrification (only samples within source region and with evidence of denitrification are shown) dashed lines represent ±1 std. dev. of enrichment factor (ε = -10) estimated from measured data.*

As to denitrification, it is unclear to what extent denitrification occurs in the aquifer. The plots in Figure 2 do not support the conclusion on denitrification, because the slopes shown in Figure 2(a) are not 0.5. For well CFO4, the slope of 0.42 is close to 0.5m, and the data listed in Table 2 and the well locations shown in Figure 1 indeed support the conclusion of denitrification. But what about well CFO1?

*We assume that the reviewer takes issue with the slope of the isotopic enrichment trend at CFO1 (0.72) as being not close enough to the general trend of 0.5 reported in some studies (Durka et al., 1994). However the value of 0.72 is not unreasonable given the range of values reported for denitrification of groundwater in the published literature of 0.47 – 0.66 (Singleton et al., 2007), 0.67 (Mengis et al., 1999), 0.77 (Fukada et al., 2003). This will has now been clarified in the description of the modelling approach.*

The authors said somewhere in the manuscript that the initial nitrate concentration can be neglected. I do not think that it is a reasonable assumption for agricultural areas.

*Presumably the reviewer is suggesting that there can be an historical legacy of nitrate in agricultural areas. This is true, and we consider "background" as not having been influenced by agricultural activity (whether this is recent or historical). This assumption that $NO_{3b}$ can be neglected underpins the simplification of the mathematics and is valid for these agricultural areas. This approach would not be suitable at sites with naturally elevated nitrate concentrations in groundwater, which is now acknowledged in the manuscript.*

*A new figure (Figure 8, see above response to reviewer 1) has been added to the revised manuscript to demonstrate that the effect of neglecting background concentrations as these sites is negligible, and discussion of the effect of neglecting background concentrations has been updated.*

[revised manuscript text omitted]

5    **Table S1. Horizontal hydraulic gradients at CFO1 at the water table.**

| Well IDs | Horizontal hydraulic gradient |
|---|---|
| DMW1 and DP10-2 | $4.63 \times 10^{-3}$ |
| DMW2 and DMW-16 | $6.06 \times 10^{-3}$ |
| DP10-2 and DMW5 | $4.39 \times 10^{-3}$ |
| DP10-2 and DMW11 | $9.74 \times 10^{-3}$ |
| DMW10 and DMW11 | $1.38 \times 10^{-2}$ |

**Table S2. Mean vertical gradients between nested water table wells and piezometers at CFO1**

| Well IDs | Vertical hydraulic gradient |
|---|---|
| DMW10 and DP11-10b | $3.34\times10^{-3}$ |
| DMW11 and DP11-11b | $-2.79\times10^{-2}$ |
| DMW12 and DP11-12b | $2.20\times10^{-3}$ |
| DMW13 and DP11-13b | $1.36\times10^{-2}$ |
| DMW14 and DP11-14b | $1.80\times10^{-3}$ |
| DMW15 and DP11-15b | $3.37\times10^{-2}$ |
| DMW16 and DP11-16b | $2.86\times10^{-2}$ |
| DP10-2 and DP10-1 | $1.78\times10^{-1}$ |

[revised manuscript text omitted]
$^{-1}$) | $f_d$ (mean ± stdev) | $NO_3$-$N_i/Cl_i$ (mean ± stdev) | $Cl_i$ (mg L$^{-1}$) min | $Cl_i$ (mg L$^{-1}$) max | $NO_3$-$N_i$ (mg L$^{-1}$) min | $NO_3$-$N_i$ (mg L$^{-1}$) max | $f_m$ min | $f_m$ max |
|---|---|---|---|---|---|---|---|---|---|---|
| **CFO1** | | | | | | | | | | |
| DMW11 | 436.1 | 17.1 | 0.17 ± 0.07 | 0.23 ± 0.10 | 436 | 667 | 98 | 150 | 0.65 | 1 |
| DMW12 | 78.0 | 2.6 | 0.23 ± 0.10 | 0.14 ± 0.06 | 78 | 1047 | 11 | 150 | 0.07 | 1 |
| DMW13 | 56.7 | 23.7 | 0.56 ± 0.22 | 0.75 ± 0.29 | 57 | 189 | 42 | 141 | 0.30 | 1 |
| DP10-2 | 74.5 | 11.8 | 0.52 ± 0.22 | 0.30 ± 0.13 | 74 | 277 | 23 | 84 | 0.27 | 1 |
| DP11-12b | 95.7 | 0.6 | 0.15 ± 0.08 | 0.04 ± 0.02 | 96 | 1300 | 4.2 | 90 | 0.07 | 1 |
| DC15-22_10m | 73.0 | 11.0 | 0.47 ± 0.21 | 0.32 ± 0.14 | 73 | 289 | 23 | 93 | 0.25 | 1 |
| DP11-13_4.3m | 28.5 | 7.0 | 0.30 ± 0.15 | 0.82 ± 0.41 | 29 | 184 | 23 | 150 | 0.15 | 1 |
| DP11-13_5.2m | 25.0 | 7.8 | 0.34 ± 0.13 | 0.91 ± 0.35 | 25 | 160 | 23 | 146 | 0.16 | 1 |
| DP11-13_7m | 72.3 | 12.0 | 0.27 ± 0.13 | 0.62 ± 0.30 | 72 | 244 | 45 | 150 | 0.30 | 1 |
| DP11-13_7.9m | 70.8 | 9.1 | 0.17 ± 0.09 | 0.76 ± 0.40 | 71 | 199 | 54 | 150 | 0.36 | 1 |
| DP11-13_8.8m | 81.7 | 11.0 | 0.32 ± 0.15 | 0.89 ± 0.42 | 82 | 323 | 39 | 150 | 0.25 | 1 |
| **CFO4** | | | | | | | | | | |
| BC4 | 163.1 | 35.1 | 0.37 ± 0.13 | 0.58 ± 0.20 | 163 | 258 | 95 | 150 | 0.63 | 1 |
| BMW2 | 595.6 | 16.5 | 0.13 ± 0.06 | 0.21 ± 0.10 | 596 | 707 | 127 | 150 | 0.84 | 1 |
| BMW5 | 131.2 | 12.9 | 0.34 ± 0.16 | 0.29 ± 0.14 | 131 | 520 | 38 | 150 | 0.25 | 1 |
| BMW6 | 156.0 | 0.4 | 0.01 ± 0.01 | 0.26 ± 0.26 | 156 | 1300 | 0.4 | 150 | 0.12 | 1 |
| BMW7 | 134.7 | 11.6 | 0.21 ± 0.11 | 0.41 ± 0.22 | 135 | 365 | 55 | 150 | 0.37 | 1 |

---

## Referee Report (RR1)

Comments to HESS-2018-31:

The paper by Bourke et al. was trying to use nitrogen isotope tracer to investigate the denitrification process in shallow aquifer at two feeding fields in Alberta, Canada. I recommend major revision on the present format. Hope these comments would be of use to you.

Abstract:

The abstract should provide a clear and concise summary of the aims of the paper together with the key results.

Page 1, Line 13: "Elevated $NO_3^-$ concentrations in groundwater" please give specific data on this in the abstract.

Lines 16-17: it should be expanded to describe "the $NO_3^-$ source and denitrification enrichment factors".

Lines 20-21: the range of $NO_3^-$ concentrations can be shown here.

Could the authors keep consistent with using the $NO_3$-N concentration or the $NO_3^-$ concentration in the whole paper? According to the description in the section RESULTS, you used "$NO_3$-N concentration in groundwater".

Introduction

The introduction provides good background to the study and places it in an international context. The specific aims of the study are reasonably clear.

Page 2: Lines 16-17: "Groundwater containing significant agriculturally derived $NO_3^-$ also typically has elevated chloride ($Cl^-$) concentrations" could use a few more details. Specifically, it is not clear what is meant by "elevated chloride concentrations" and "the characteristic enrichment of ... (Line 34)" etc. Without a detailed knowledge of the area it is difficult to assess exactly the extent of the problem or what has been done to address it. If you provide a few more details, the context will be clearer.

Page 3: "the extent of agriculturally derived $NO_3^-$ in groundwater" compared with "On-site denitrification reduced agriculturally derived $NO_3^-$ concentrations by at least half" (in Abstract) did not show the variational concentrations. "sources and initial concentrations of $NO_3^-$" (Line 25, Page 3) was not clear in the text and abstract.

Materials and methods:

Page 4, Line 28: "slug or bail tests" could you expand them and give brief introduction to make the experiment clearer for readers.

Page 5, 2.3 section: where did you complete the measurements of groundwater and pore-water hydrochemical components?

Results:

Page 8, Lines 4-19: how to obtain some specific data (e.g. mean K, vertical gradient) in this part?

Lines 25-30:"The enrichment factor of $\delta^{15}NO_3$" can be replaced by "$\varepsilon^{15}N$".

Page 10, Lines 17-18: "The NO3-N in this core sample was most likely introduced into the groundwater system by vertical infiltration or diffusion from above." What's your evidence for this description?

Line 18: "much higher Cl- concentrations" please give a range of concentration levels with mean value.

Discussion:

This section needs in-depth analysis and focus on interpreting data combined with hydrogeological conditions, such as characteristics of glacial sediments, and agricultural operations.

The thrust of this paper should be distinguished into two or three aspects, explaining the main factors controlling the denitrification processes in the groundwater systems. You should try to keep this as the main focus of the paper. Additionally, how does the paper inform our understanding of nitrate fate in groundwater caused by agricultural activity in general?

It is mainly a case study and while these are important, you need to revisit those topics and explain in the conclusions the relevance to research elsewhere.

Conclusions:

There are very study specific. Please use the conclusions to look at the broader implications for regions outside this specific area. There must be a few things in here that will inform studies in other regions that will give the paper more impact.

---

## Author Response (AR2)

Comments to HESS-2018-31:

The paper by Bourke et al. was trying to use nitrogen isotope tracer to investigate the denitrification process in shallow aquifer at two feeding fields in Alberta, Canada. I recommend major revision on the present format. Hope these comments would be of use to you.

Abstract:

The abstract should provide a clear and concise summary of the aims of the paper together with the key results.

Page 1, Line 13: "Elevated NO3- concentrations in groundwater" please give specific data on this in the abstract.

*This sentence is a general introductory statement about potential mechanisms for nitrate attenuation in groundwater and does not summarise the results of this study. It would not be appropriate to report measured data here.*

*NO CHANGE MADE*

Lines 16-17: it should be expanded to describe "the NO3- source and denitrification enrichment factors".

*It isn't entirely clear how this sentence could be expanded. This sentence describes aspects of the modelling approach and is a complete statement as written. Further detail is provided in the main body of the paper.*

*NO CHANGE MADE*

Lines 20-21: the range of NO3- concentrations can be shown here.

*Readers can refer to the full open-access manuscript for details of the range of measured values (e.g. Table 2).*

*CHANGE MADE: sentence changed and maximum measured concentration now reported in the abstract.*

Could the authors keep consistent with using the NO3-N concentration or the NO3- concentration in the whole paper? According to the description in the section RESULTS, you used "NO3-N concentration in groundwater".

*When referring to concentrations in measured data and values used in the data analysis we use $NO_3$-N, which refers to the nitrogen in the form of nitrate that was actually measured. When referring to concentrations this must be specified in full for accuracy.*

*When discussing processes in general we use the simpler $NO_3$ for the benefit of brevity and readability. Any statement made about processes controlling $NO_3$ will also clearly apply to the N in this $NO_3$ (which is $NO_3$-N).*

*We feel that this approach balances the needs for accuracy and readability.*
*NO CHANGE MADE*

Introduction

The introduction provides good background to the study and places it in an international context. The specific aims of the study are reasonably clear.

Page 2: Lines 16-17: "Groundwater containing significant agriculturally derived NO3- also typically has elevated chloride (Cl-) concentrations" could use a few more details. Specifically, it is not clear what is meant by "elevated chloride concentrations" and "the characteristic enrichment of ... (Line 34)" etc. Without a detailed knowledge of the area it is difficult to assess exactly the extent of the problem or what has been done to address it. If you provide a few more details, the context will be clearer.

*This is a general statement that acts as an introduction as to why NO3/Cl ratios can be used to investigate $NO_3$ contamination. The Cl- is not a problem in and of itself. Site specific details on agricultural activity are already presented in 2.1 Experimental sites.*
*NO CHANGE MADE*

Page 3: "the extent of agriculturally derived NO3- in groundwater" compared with "On-site denitrification reduced agriculturally derived NO3- concentrations by at least half" (in Abstract) did not show the variational concentrations.

*It is not clear what the reviewer is suggesting here, it makes no sense to report concentrations measured in this study in the Introduction.*
*NO CHANGE MADE*

"sources and initial concentrations of NO3-" (Line 25, Page 3) was not clear in the text and abstract.

*It's not entirely clear what the reviewer is asking for here. If the issue is a lack of information on site-specific agricultural practices in the introduction, this is already presented in 2.1 Experimental sites.*
*NO CHANGE MADE*

Materials and methods:

Page 4, Line 28: "slug or bail tests" could you expand them and give brief introduction to make the experiment clearer for readers.

*These are standard hydrogeological tests, and the purpose is simply to determine hydraulic conductivities (as already stated) be measuring water level changes in response to addition or removal of water from the well.*

*CHANGE MADE: description of "slug test" and "bail test" added for clarity along with a reference for the analysis approach.*

Page 5, 2.3 section: where did you complete the measurements of groundwater and pore-water hydrochemical components? *Pore water from core was analysed at USask. Groundwater wells were Analysed in Lethbridge and manure filtrate was analysed by ALS in Saskatoon.*

*CHANGE MADE: Laboratory now specified in the text.*

Results:

Page 8, Lines 4-19: how to obtain some specific data (e.g. mean K, vertical gradient) in this part?

*Site-specific ranges of values are reported here in the main text so as to give an indication of the variability in this heterogeneous groundwater system. Geometric mean K's are already presented in the text of 3.1 where the average velocity is calculated. The complete dataset of measured K and hydraulic gradients are presented in supplementary information. Readers are free to calculate means from the raw data if they wish.*

*NO CHANGE MADE*

Lines 25-30:"The enrichment factor of $\delta15NO3$" can be replaced by "$\varepsilon15N$".
*Presumably the reviewer is referring to page 9.*
*CHANGE MADE as suggested*

Page 10, Lines 17-18: "The NO3-N in this core sample was most likely introduced into the groundwater system by vertical infiltration or diffusion from above." What's your evidence for this description?

*The well directly up-gradient or this core has low NO3, but the soil above the watertable in this core has very high nitrate, therefore this high nitrate concentration at the top of the core is most likely to be top-down input, not lateral flow within the subsurface.*

*CHANGE MADE: The text has been updated and expanded to clarify this.*

Line 18: "much higher Cl- concentrations" please give a range of concentration levels with mean value.

*It seems the reviewer means page 11. Cl- concentrations for BMW2 are already reported within this sentence as a mean and standard deviation.*

*CHANGE MADE: Concentrations for BMW3 and BMW4 now also reported.*

Discussion:

This section needs in-depth analysis and focus on interpreting data combined with hydrogeological conditions, such as characteristics of glacial sediments, and agricultural operations.

*In-depth analysis of the data at each site is already presented within results.*

*NO CHANGE MADE*

The thrust of this paper should be distinguished into two or three aspects, explaining the main factors controlling the denitrification processes in the groundwater systems. You should try to keep this as the main focus of the paper.

*The focus of the paper was not on explaining factors controlling denitrification because we did not have the data to do this. Denitrification will depend on redox conditions, and because the wells were sampled using bailers with long recovery times the measured values of DO and ORP recorded are unlikely reflect conditions within the groundwater system. This was previously discussed in response to comments by Sebastian Lamontagne during the open discussion phase (AC1).*

*NO CHANGE MADE*

Additionally, how does the paper inform our understanding of nitrate fate in groundwater caused by agricultural activity in general? It is mainly a case study and while these are important, you need to revisit those topics and explain in the conclusions the relevance to research elsewhere.

*Paragraphs 1-3 of the discussion already address how the results from these sites can be used to inform our understanding of agricultural practices overlying glacial sediments in general.*

*CHANGE MADE: Subheadings within the discussion have been added to clarify the sub-topics for the reader.*

Conclusions:

There are very study specific. Please use the conclusions to look at the broader implications for regions outside this specific area. There must be a few things in here that will inform studies in other regions that will give the paper more impact.

*A number of the concluding statements were intended to apply broadly, not just at these sites.*

*CHANGE MADE: The conclusion has been modified and re-structured into two paragraphs to clarify the broad general findings.*

*A number of minor modifications have also been made throughout the manuscript to improve clarity and readability.*